# Dynamic Regret Reduces to Kernelized Static Regret

**Andrew Jacobsen**[*]
Università degli Studi di Milano
Politecnico di Milano
contact@andrew-jacobsen.com

**Alessandro Rudi**[*]
Bocconi University
alessandro.rudi@sdabocconi.it

**Francesco Orabona**
King Abdullah University of Science and Technology (KAUST)
Thuwal, 23955-6900, Kingdom of Saudi Arabia
francesco@orabona.com

**Nicolò Cesa-Bianchi**
Università degli Studi di Milano
Politecnico di Milano
nicolo.cesa-bianchi@unimi.it

## Abstract

We study dynamic regret in online convex optimization, where the objective is to achieve low cumulative loss relative to an arbitrary benchmark sequence. By observing that competing with an arbitrary sequence of comparators $u_1, \ldots, u_T$ in $\mathcal{W} \subseteq \mathbb{R}^d$ can be reframed as competing with a *fixed* comparator *function* $u : [1, T] \to \mathcal{W}$, we cast dynamic regret minimization as a *static regret* problem in a *function space*. By carefully constructing a suitable function space in the form of a Reproducing Kernel Hilbert Space (RKHS), our reduction enables us to recover the optimal $R_T(u_1, \ldots, u_T) = \mathcal{O}(\sqrt{\sum_t \|u_t - u_{t-1}\| T})$ dynamic regret guarantee in the setting of linear losses, and yields new scale-free and directionally-adaptive dynamic regret guarantees. Moreover, unlike prior dynamic-to-static reductions—which are valid only for linear losses—our reduction holds for *any* sequence of losses, allowing us to recover $\mathcal{O}(\|u\|_{\mathcal{H}}^2 + d_{\mathrm{eff}}(\lambda) \ln T)$ bounds when the losses have meaningful curvature, where $d_{\mathrm{eff}}(\lambda)$ is a measure of complexity of the RKHS. Despite working in an infinite-dimensional space, the resulting reduction leads to algorithms that are computable in practice, due to the reproducing property of RKHSs.

## 1 Introduction

This paper introduces new techniques for *Online Convex Optimization* (OCO), a framework for designing and analyzing algorithms which learn on-the-fly from a stream of data [10, 11, 18, 39, 59]. Formally, consider $T$ rounds of interaction between a learner and the environment. In each round, the learner chooses $w_t \in \mathcal{W}$ from a convex set $\mathcal{W} \subseteq \mathbb{R}^d$, the environment reveals a convex loss function $\ell_t : \mathcal{W} \to \mathbb{R}$, and the learner incurs a loss of $\ell_t(w_t)$. The classic objective in this setting is to minimize the learner's *regret* relative to any fixed benchmark $u \in \mathcal{W}$:

$$R_T(u) := \sum_{t=1}^{T} (\ell_t(w_t) - \ell_t(u)) .$$

---

[*]Equal contribution.

39th Conference on Neural Information Processing Systems (NeurIPS 2025).

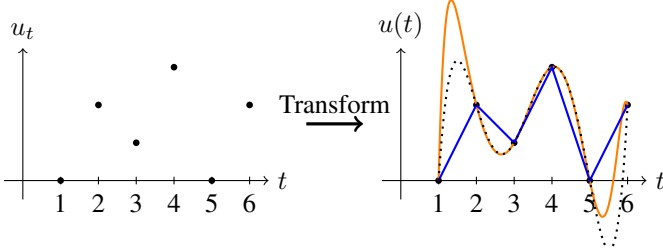

Figure 1: Transformation from a sequence of comparators to a function. Many functions may implement the transformation. In Section 4 we will see that under mild assumptions on the chosen function space $\mathcal{H}$ we can always find a function $u \in \mathcal{H}$ such that $u(t) = u_t$ for all $t$ and $\|u\|^2_{\mathcal{H}^d} = O\big( \sum_{t=2}^T \|u_t - u_{t-1}\| \big)$.

In this paper, we study the more general problem of minimizing the learner's regret relative to any *sequence* of benchmarks $u_1, \ldots, u_T \in \mathcal{W}$ [21, 22, 59]:

$$\mathrm{R}_\mathrm{T}(u_1, \ldots, u_T) := \sum_{t=1}^T (\ell_t(w_t) - \ell_t(u_t)) .$$

This objective is typically referred to as *dynamic* regret, to distinguish it from the special case where the comparator sequence is fixed $u_1 = \cdots = u_T$ (referred to as *static* regret). Intuitively, dynamic regret captures a notion of *non-stationarity* in learning problems. Problem instances where $u_1 = \cdots = u_T$ model classic problem settings, wherein there is a fixed "solution" whose performance we want to emulate, while a time-varying comparator sequence models problem settings where the learner needs to continuously adapt to a changing environment in which the solution is time-varying. The complexity of a given comparator sequence is typically characterized by its *path-length*:

$$P_T = \sum_{t=2}^T \|u_t - u_{t-1}\| .$$

Clearly, if the path-length is large there is no hope to obtain low dynamic regret. The goal is thus to obtain performance guarantees that gracefully *adapt* to the level of non-stationarity. For instance, in the setting of $G$-Lipschitz losses and a bounded domain $D = \sup_{x,y \in \mathcal{W}} \|x - y\|$, the minimax optimal dynamic regret guarantee is of the order of $\mathcal{O}(G\sqrt{(D^2 + DP_T)T})$, which scales naturally with the complexity of the benchmark sequence and recovers the optimal $\mathcal{O}(GD\sqrt{T})$ static regret guarantee when the comparator is fixed. In unbounded domains (e.g., $\mathcal{W} = \mathbb{R}^d$) these bounds would be vacuous, so the guarantee should instead be adaptive to $M := \max_t \|u_t\|$. In this case an analogous guarantee of $\widetilde{\mathcal{O}}(G\sqrt{(M^2 + MP_T)T})$ can be achieved at the expense of additional logarithmic terms. Throughout the paper we focus on the unbounded setting.

**Contributions.** In this work we introduce a new framework for reducing dynamic regret minimization to static regret minimization. Our key insight is that competing with a *sequence* $u_1, \ldots, u_T$ in $\mathcal{W}$ can be equivalently framed as competing with some fixed *function* $u(\cdot)$ such that $u(t) = u_t$ for all $t$. In this view, we effectively transform dynamic regret minimization over a domain $\mathcal{W} \subseteq \mathbb{R}^d$ into a *static* regret minimization problem over a domain of *functions*, depicted graphically in Figure 1.

The choice of the function space is crucial, as it controls the trade-offs of the resulting algorithm. To complete the construction, we carefully design a rich family of function spaces which embed the comparator sequence in a way that (1) optimizes the inherent trade-offs of the function class to achieve optimal dynamic regret guarantees and (2) ensures that the resulting algorithm is computable in practice, despite being stated as an infinite-dimensional optimization problem. Indeed, the family we design is an instance of a Reproducing Kernel Hilbert Space (RKHS), a well-studied class of functions endowed with the familiar structure of a Hilbert space. The reduction to learning in an RKHS is particularly natural in the context of online learning—the vast majority of modern online learning theory is developed for *static regret minimization in Hilbert spaces*, so our reduction enables the use of the familiar online learning toolkit while also allowing us to draw upon deep connections between dynamic regret minimization, kernel methods, and signal processing theory.

In the linear losses setting, our construction enables us to achieve the optimal dynamic regret guarantees of $\mathcal{O}(\sqrt{MP_T T})$ up to poly-logarithmic terms. Notably, the resulting algorithm is naturally *horizon independent*, and is easily extended to a *scale-free* version. These are the first algorithms

that obtain the optimal $\sqrt{P_T}$ dependence without prior knowledge of the horizon $T$ natively, without resorting to the doubling trick. Our reduction also enables us to derive new *directionally-adaptive* guarantees, which scale as $\widetilde{\mathcal{O}}\Big(\sqrt{d_{\text{eff}}(\lambda)\big(\|u\|_{\mathcal{H}^d}^2 + \sum_{t=1}^T \langle g_t, u_t\rangle^2\big)}\Big)$, where $\|u\|_{\mathcal{H}^d}^2$ and $d_{\text{eff}}(\lambda)$ are measures of the complexity of the comparator function and complexity of the function class $\mathcal{H}$ respectively.

Interestingly, because our reduction only involves viewing the comparator sequence through a different lens, it holds for *any* sequence of loss functions, contrasting prior works which are valid only for linear losses [26, 55]. We show that this allows us to account for loss *curvature* and obtain $\mathcal{O}(\lambda \|u\|_{\mathcal{H}}^2 + d_{\text{eff}}(\lambda) \log T)$ dynamic regret in the context of strongly-convex, exp-concave, and improper linear regression settings.

**Related Works.** Our work is most directly related to a recent thread of research in the linear loss setting initiated by Zhang et al. [55]. Their strategy approaches dynamic regret from a signal processing perspective, wherein the comparator sequence is stacked into a high-dimensional "signal" $\widetilde{u} = \text{vec}\,(u_1, \ldots, u_T) \in \mathbb{R}^{dT}$, and 1-dimensional static regret algorithms are employed to learn the coefficients of a basis of features which decompose that signal, leading to $\mathcal{O}(\sqrt{MP_T T})$ dynamic regret via a carefully chosen dictionary of features. Jacobsen and Orabona [26] generalize this perspective by designing static regret algorithms that are applied directly in this high-dimensional space, and derive the $\mathcal{O}(\sqrt{MP_T T})$ bound by choosing a suitable dual-norm pair in this space, such that $\|\widetilde{u}\| = \mathcal{O}(\sqrt{P_T})$. Our work further extends this perspective by interpreting the comparator sequence as samples of a *function* in an RKHS $\mathcal{H}$, and designing algorithms which obtain suitable static regret guarantees in function space. The reduction of Jacobsen and Orabona [26] can in fact be understood as a special case of our framework by choosing the discrete RKHS $\mathcal{H}$ associated with the Dirac kernel.

More broadly, the concept of dynamic regret was originally introduced by Herbster and Warmuth [21, 22]. Later, Zinkevich [59] showed that OGD naturally obtains $\mathcal{O}(P_T\sqrt{T})$ dynamic regret and Yang et al. [52] showed that $\mathcal{O}(\sqrt{DP_T T})$ can be achieved when prior knowledge of $P_T$ is available. The first to achieve the $\mathcal{O}(\sqrt{DP_T T})$ rate *without* prior knowledge of $P_T$ was Zhang et al. [53], who also proved a matching lower bound, and the analogous bound of $\mathcal{O}(\sqrt{MP_T T})$ has been achieved up to logarithmic terms in unbounded settings [23, 24, 33, 55]. There have also been several refinements to the result, replacing the $T$ factor with data-dependent quantities such as $\sum_{t=1}^T \|g_t\|^2$ or $\sum_t \sup_x |\ell_t(x) - \ell_{t-1}(x)|$ [9, 14, 19]. Going beyond linear losses, various improvements in adaptivity can be obtained when the losses are smooth or exp-concave, such as replacing the $T$ factor with $\sum_t \ell_t(u_t)$ or $\sum_t \sup_w \|\nabla \ell_t(w) - \nabla \ell_{t-1}(w)\|^2$ [56–58]. In the squared loss setting $\ell_t(w) = \frac{1}{2}(y_t - w)^2$, minimax optimal rates of $R_T(\mathring{y}_1, \ldots, \mathring{y}_T) = \mathcal{O}(C_T^{2/3} T^{1/3})$ have been obtained where $C_T = \sum_{t=2}^T |\mathring{y}_t - \mathring{y}_{t-1}|$ is the path-length of the benchmark predictions [3–5, 29, 44, 54].

## 2 Preliminaries

**Notations.** Hilbert spaces are denoted by upper case calligraphic letters. Given a Hilbert space $\mathcal{H}$, we denote the associated inner product by $\langle \cdot, \cdot \rangle_{\mathcal{H}}$. We denote $L(\mathcal{H}, \mathcal{W})$ the space of linear operators from $\mathcal{H}$ to $\mathcal{W}$. $L(\mathcal{H}, \mathcal{W})$ is itself a Hilbert space when equipped with the Hilbert-Schmidt inner product, $\langle A, B \rangle_{\text{HS}} = \text{Tr}\,(A^* B)$, where $A^* \in L(\mathcal{W}, \mathcal{H})$ is the adjoint of $A$. The subdifferential set of a function $f$ at $x$ is denoted by $\partial f(x)$. We will occasionally abuse notation and write $\nabla f(x)$ to mean an arbitrary element of $\partial f(x)$. We will denote by $[T]$ the set $\{1, 2, \ldots, T\}$. The Fourier transform of a function $Q$ is denoted $F[Q](x) = \int_{\mathbb{R}} Q(\omega)e^{-2\pi i x\omega}d\omega$ and, when clear from context, we will generally abbreviate $F[Q](x) =: \widehat{Q}(x)$.

**Reproducing Kernel Hilbert Spaces.** Let $\mathcal{H} = \{h : \mathcal{T} \to \mathbb{R}\}$ be a Hilbert space of functions a on compact set $\mathcal{T}$. The space $\mathcal{H}$ is a RKHS [1] if there exists a positive definite function $k : \mathcal{T} \times \mathcal{T} \to \mathbb{R}$ such that $k(\cdot, x) \in \mathcal{H}$ for all $x \in \mathcal{T}$, that has the *reproducing property*, i.e., we have $f(x) = \langle f, k(\cdot, x)\rangle_{\mathcal{H}}$ for all $f \in \mathcal{H}$ and $x \in \mathcal{T}$. The function $k$ is called the *kernel function* associated with $\mathcal{H}$ and the function $\phi(x) = k(\cdot, x)$ is called the *feature map*. It is known that the kernel function uniquely characterizes the RKHS $\mathcal{H}$. A kernel is *universal* if it can approximate any

---
**Algorithm 1:** Kernelized Online Learning

**Input:** Domain $\mathcal{W} \subseteq \mathbb{R}^d$, feature map $\phi : \mathcal{T} \to \mathcal{H}$, algorithm $\mathcal{A}$ defined on $L(\mathcal{H}, \mathcal{W})$
**for** $t = 1 : T$ **do**

    Receive $W_t \in L(\mathcal{H}, \mathcal{W})$ from $\mathcal{A}$ and play $w_t = W_t \phi(t) \in \mathcal{W}$

    Observe loss function $\ell_t : \mathcal{W} \to \mathbb{R}$ and incur loss $\ell_t(w_t)$

    Send $\widetilde{\ell}_t : W \mapsto \ell_t(W\phi(t))$ to $\mathcal{A}$ as the $t^{\text{th}}$ auxiliary loss

**end**

---

real-valued continuous function on $\mathcal{T}$ to arbitrary accuracy. Many of the standard kernel functions are universal, including the Gaussian RBF kernel, the Matérn kernel, and inverse multiquadratic kernel. *All kernels considered in this work are universal kernels.* For a detailed introduction to kernel methods, see, e.g., Berlinet and Thomas-Agnan [6], Paulsen and Raghupathi [42], Schölkopf and Smola [45], Wendland [51].

We will often be interested in functions taking values in $\mathcal{W} \subseteq \mathbb{R}^d$. In this case the usual RKHS machinery extends in a straight-forward way via a coordinate-wise extension. Indeed, we can represent $w : \mathcal{T} \to \mathbb{R}^d$ as a tuple $w = (w_1, \ldots, w_d)$ such that $w_i \in \mathcal{H}$ for each $i$.[2] This naturally leads to an operator-based version of the reproducing property:

$$w(t) = (w_1(t), \ldots, w_d(t)) = \left( \langle w_1, \phi(t) \rangle_{\mathcal{H}}, \ldots, \langle w_d, \phi(t) \rangle_{\mathcal{H}} \right) = W\phi(t) \in \mathcal{W},$$

where $W \in L(\mathcal{H}, \mathcal{W})$. The space $L(\mathcal{H}, \mathcal{W})$ is itself a Hilbert space when equipped with the Hilbert-Schmidt norm, and under the coordinate-wise extension above we have $\|W\|_{\text{HS}}^2 = \|w\|_{\mathcal{H}^d}^2 = \sum_{i=1}^d \|w_i\|_{\mathcal{H}}^2$. Moreover, observe that when $d = 1$ this setup simply reduces back to the usual setup, wherein $\|W\|_{\text{HS}} = \|w\|_{\mathcal{H}}$ and $w(t) = \langle w, \phi(t) \rangle_{\mathcal{H}}$.

For notational clarity we will refer to functions $w(\cdot) \in \mathcal{H}$ and their values $w(t) \in \mathcal{W}$ with lower-case letters, and their representation $W \in L(\mathcal{H}, \mathcal{W})$ using the upper-case. We will typically use the notation $w_t$ in place of $w(t)$ when referring to the evaluations of $w(t)$ at discrete time-points $t \in [T]$.

## 3 An Equivalence Between Static and Dynamic Regret

In this section, we present the main tool that we will use to develop dynamic regret guarantees for online learning. The key idea is to interpret the comparator sequence $u_1, \ldots, u_T \in \mathcal{W}$ as the evaluations of a function $u(\cdot)$ at the discrete time-points $t \in [T]$, allowing us to re-frame dynamic regret minimization as a *static regret minimization in function space*.

Note that most existing work in online learning revolves around learning in Hilbert spaces, not general function spaces, so if we hope to leverage these existing tools we should embed the comparator sequence in a *Hilbert space of functions*. In particular, our approach will be to embed the comparator sequence in a Hilbert space $\mathcal{H}$ of functions representable by a reproducing kernel $k(s, t)$ and feature map $\phi : \mathcal{T} \to \mathcal{H}$. Note that this is always possible by selecting a universal kernel on $\mathcal{T}$. Our reduction is conceptually shown in Algorithm 1 and the following theorem shows that the dynamic regret w.r.t. comparator sequence $u_1, \ldots, u_T$ in $\mathcal{W}$ is equivalent to *static* regret w.r.t. a function $u(\cdot) \in \mathcal{H}$ on the auxiliary loss sequence $\widetilde{\ell}_t : W \mapsto \ell_t(W\phi(t))$.

**Theorem 1** (Dynamic Regret via Kernelized Static Regret). *Let $\mathcal{T}$ be a compact set, $\mathcal{W} \subseteq \mathbb{R}^d$, let $\mathcal{H}$ be an RKHS with associated feature map $\phi : \mathcal{T} \to \mathcal{H}$, and for any $W \in L(\mathcal{H}, \mathcal{W})$ let $\widetilde{\ell}_t(W) = \ell_t(W\phi(t))$. Let $W_1, \ldots, W_T$ be an arbitrary sequence in $L(\mathcal{H}, \mathcal{W})$ and suppose that on each round we play $w_t = W_t\phi(t) \in \mathcal{W}$. Then, for any comparator sequence $u_1, \ldots, u_T$ in $\mathcal{W}$ and $U \in L(\mathcal{H}, \mathcal{W})$ satisfying $u_t = U\phi(t)$ for all $t$,*

$$\text{R}_{\text{T}}(u_1, \ldots, u_T) = \sum_{t=1}^T (\ell_t(w_t) - \ell_t(u_t)) = \sum_{t=1}^T (\widetilde{\ell}_t(W_t) - \widetilde{\ell}_t(U)) =: \widetilde{\text{R}}_T(U) .$$

Note that the reduction holds for *any* operator $U \in L(\mathcal{H}, \mathcal{W})$ which interpolates the comparator sequence, $u_t = U\phi(t) \ \forall t$. Hence, we can always let $u(\cdot) \in \mathcal{H}$ be the minimum norm function in $\mathcal{H}$ which interpolates these points, and take $U$ to be its representation in $L(\mathcal{H}, \mathcal{W})$. In fact, since $k$ is a

---

[2]This connection can be made more formally via Riesz representation theorem, see Appendix A for details.

universal kernel, we can assume that $u(\cdot) \in \mathcal{H}$ approximates *any* continuous function on $\mathcal{T} \subseteq [1, T]$ to arbitrary accuracy, so the assumption that $u(\cdot)$ lives in an RKHS $\mathcal{H}$ does not actually restrict the functions that we can compare against in a significant way. We will see a concrete example of an RKHS $\mathcal{H}$ which reconstructs arbitrary comparator sequences in later sections (e.g., Theorem 4).

Our reduction bears a strong resemblance to the reduction recently proposed by Jacobsen and Orabona [26], which works by embedding the comparator sequence in $\mathbb{R}^{dT}$ by simply "stacking" the comparator sequence into one long vector, $\widetilde{u} = \text{vec}(u_1, \ldots, u_T) \in \mathbb{R}^{dT}$. In fact, we show in Appendix B that our framework *precisely* recovers the reduction in Jacobsen and Orabona [26] by choosing the discrete RKHS $\mathcal{H}$ associated with the Dirac kernel. However, notice in particular that this means that their reduction is inherently tied to *finite*-dimensional features, whereas ours enables *infinite*-dimensional features. As we will see in Section 4, this distinction is key to obtaining the optimal path-length dependencies in a horizon-independent manner. Moreover, note that the regret equality in Jacobsen and Orabona [26] holds *only* in the context of linear losses, $\ell_t(w) \mapsto \langle g_t, w \rangle$, whereas in our framework the regret equality holds for any sequence of losses $\ell_1, \ldots, \ell_T$. We will see that this distinction is important in Section 5—for example, our reduction allows us to preserve the curvature of the losses needed to obtain $\mathcal{O}(\|u\|_{\mathcal{H}}^2 + d_{\text{eff}}(\lambda) \ln T)$ bounds when the original losses $\ell_t$ are strongly convex or exp-concave, where $d_{\text{eff}}(\lambda)$ is a measure of complexity of the RKHS.

## 4 Linear Losses

We first consider the setting of online linear optimization. In this setting, on round $t$ the learner receives linear loss $\ell_t(w) = \langle g_t, w \rangle_{\mathcal{W}}$, so recalling the reduction in the previous section defines the auxiliary loss as $\widetilde{\ell}_t \colon W \in L(\mathcal{H}, \mathcal{W}) \mapsto \ell_t(W\phi(t)) \in \mathbb{R}$, we have

$$\widetilde{\ell}_t(W) = \langle g_t, W\phi(t) \rangle_{\mathcal{W}} = \langle g_t \otimes \phi(t), W \rangle_{\text{HS}} = \langle G_t, W \rangle_{\text{HS}},$$

where $G_t = g_t \otimes \phi(t) \in L(\mathcal{H}, \mathcal{W})$ is the rank one operator such that $(g_t \otimes \phi(t))(h) = \langle \phi(t), h \rangle_{\mathcal{H}} g_t$ for any $h \in \mathcal{H}$. As such, it is important that the base algorithm facilitates an application of the kernel trick to avoid explicitly evaluating the feature map $\phi(t)$, which may be infinite-dimensional in general. To help make things concrete and provide intuitions, the following example shows that many of the common algorithms based on Follow the Regularized Leader (FTRL) with a radially-symmetric regularizer $w \mapsto \Psi_t(\|w\|)$ are amenable to the kernel trick. This class captures many of the fundamental regularizers in online learning, such as quadratic regularizers and the "linearithmic" [41] regularizers $\Psi_t(\|w\|) \approx \|w\| \sqrt{t \log(\|w\|/\alpha + 1)}$ associated with the comparator-adaptive regret guarantees that the key result of this section (Proposition 1) will be derived from.

**Example 1.** *(Kernel Trick for Kernelized FTRL) Let $g_1, \ldots, g_T$ be a sequence in $\mathcal{W}$ and let $G_t = g_t \otimes \phi(t) \in L(\mathcal{H}, \mathcal{W})$ for all $t$. Let $\theta_t = -\sum_{s=1}^{t-1} G_s$, $V_t = \sum_{s=1}^{t-1} \|G_t\|_{\text{HS}}^2$, let $\Psi_t(\cdot; V_t)$ be a convex function with differentiable Fenchel conjugate $\Psi_t^*$, and consider the following FTRL update:*

$$W_t = \underset{W \in L(\mathcal{H}, \mathcal{W})}{\arg\min} \langle \theta_t, W \rangle + \Psi_t(\|W\|_{\text{HS}}; V_t) = \nabla_\theta \Psi_t^*(\|\theta_t\|_{\text{HS}}; V_t) = \frac{\theta_t}{\|\theta_t\|_{\text{HS}}} (\Psi_t^*)'(\|\theta_t\|_{\text{HS}}; V_t).$$

*Then, $V_t = \sum_{s=1}^{t-1} \|g_s\|_{\mathcal{W}}^2 k(t, t)$ (Lemma 10), $\|\theta_t\|_{\text{HS}}^2 = \sum_{s,s'=1}^{t-1} k(s, s') \langle g_s, g_{s'} \rangle_{\mathcal{W}}$ (Lemma 9), and on round $t$, Algorithm 1 plays*

$$w_t = W_t \phi(t) = -\frac{(\Psi_t^*)'(\|\theta_t\|_{\text{HS}}; V_t)}{\|\theta_t\|_{\text{HS}}} \sum_{s=1}^{t-1} k(s, t) g_s.$$

The example shows that many common instances of FTRL can be kernelized without explicit computation of the feature map. The example also demonstrates an important consideration when applying static regret decompositions of this nature: the update described above would require $\mathcal{O}(t)$ time and memory to implement in general, while existing algorithms for dynamic regret can often be implemented using $\mathcal{O}(\ln T)$ computation and memory [23, 53, 55, 58]. Luckily, there is already a deep and well-developed literature on efficient approximations for kernel methods that can be leveraged to translate the algorithms developed from the kernelized OCO point-of-view into more practically implementable algorithms [see, e.g., 31, 47]. Since these extensions are already well-understood and since implementing these details would not yield any new insights in the current paper, we will not consider them further here, focusing instead on the theoretical development.

Now that we have seen how to translate an algorithm's updates to the kernelized setting, we turn now to how to translate its static regret guarantees into dynamic regret guarantees. The following result shows that an algorithm's kernelized static regret guarantee translates in a straight-forward way to a dynamic regret guarantee in the original problem. The proof is immediate by applying Theorem 1 and computing $\|G_t\|_{\mathrm{HS}} = \|g_t \otimes \phi(t)\|_{\mathrm{HS}} = \|g_t\|_{\mathcal{W}} \sqrt{k(t,t)}$ by Lemma 10.

**Theorem 2.** *Let $\mathcal{A}$ be an online learning algorithm defined on Hilbert space $\mathcal{V}$. Suppose that for any sequence of convex loss functions $h_1, \ldots, h_T$ on $\mathcal{V}$, $\mathcal{A}$ obtains a bound on the static regret of the form $\widetilde{\mathrm{R}}_T(U) \leq B_T\big(\|U\|_{\mathcal{V}}, \|\nabla h_1(W_1)\|_{\mathcal{V}}, \ldots, \|\nabla h_T(W_T)\|_{\mathcal{V}}\big)$ for any comparator $U \in \mathcal{V}$ and some function $B_T : \mathbb{R}_{\geq 0}^{T+1} \to \mathbb{R}$, where $\nabla h_t(W_t) \in \partial h_t(W_t)$ for all $t$. If we apply $\mathcal{A}$ in $\mathcal{V} = L(\mathcal{H}, \mathcal{W})$ with $\|\cdot\|_{\mathcal{V}} = \|\cdot\|_{\mathrm{HS}}$, then for any sequence $u_1, \ldots, u_T$ in $\mathcal{W}$ and $U \in L(\mathcal{H}, \mathcal{W})$ satisfying $u_t = U\phi(t)$ for all $t$, Algorithm 1 with $\mathcal{A}$ guarantees*

$$\mathrm{R}_{\mathrm{T}}(u_1, \ldots, u_T) \leq B_T\left(\|U\|_{\mathrm{HS}}, \|g_1\|_{\mathcal{W}} \sqrt{k(1,1)}, \ldots, \|g_T\|_{\mathcal{W}} \sqrt{k(T,T)}\right),$$

*where $g_t \in \partial \ell_t(w_t)$ for all $t$, and $k(\cdot, \cdot)$ is the* reproducing kernel *associated to the space $\mathcal{H}$.*

The value of the lemma is that it enables us to immediately translate static regret guarantees from OLO to guarantees in our RKHS formulation of dynamic regret, wherein the complexity of the comparator sequence is measured by the RKHS norm $\|U\|_{\mathrm{HS}} = \|u\|_{\mathcal{H}^d}$. For instance, if we simply apply the standard (sub)gradient descent guarantee to Theorem 2 we get

$$\mathrm{R}_{\mathrm{T}}(u_1, \ldots, u_T) = \widetilde{\mathrm{R}}_T(U) \leq \frac{\|u\|_{\mathcal{H}^d}^2}{2\eta} + \frac{\eta}{2} \sum_{t=1}^{T} \|g_t\|_{\mathcal{W}}^2 \, k(t,t) .$$

Optimally tuning $\eta$ yields $\mathrm{R}_{\mathrm{T}}(u_1, \ldots, u_T) \leq \|u\|_{\mathcal{H}^d} \sqrt{\sum_{t=1}^{T} \|g_t\|_{\mathcal{W}}^2 \, k(t,t)}$, so achieving the optimal $\mathcal{O}\left(\sqrt{P_T T}\right)$ has effectively been reduced to the problem of *designing a kernel* such that[3] $\|u\|_{\mathcal{H}^d} = \sqrt{\sum_{i=1}^{d} \|u_i\|_{\mathcal{H}}^2} = \widetilde{\mathcal{O}}(\sqrt{P_T})$ while controlling $k(t,t)$. We will see in Section 4.1 that this can be accomplished by using a carefully chosen translation-invariant kernel.

In the above argument, the optimal choice of $\eta$ would require prior knowledge of $\|u\|_{\mathcal{H}^d}$ and cannot be chosen in general. Luckily, there are static regret algorithms which can *adapt* to the comparator norm automatically to obtain the optimal trade-off up to logarithmic terms [16, 23, 35, 36, 40]. For our purposes we will refer to an algorithm $\mathcal{A}$ defined on Hilbert space $\mathcal{V}$ as *parameter-free* if for any sequence $G$-Lipschitz loss functions $h_1, \ldots, h_T$ and any $U \in \mathcal{V}$, $\mathcal{A}$ guarantees

$$\widetilde{\mathrm{R}}_T(U) = \widetilde{\mathcal{O}}\left(\|U\|_{\mathcal{V}} \sqrt{\sum_{t=1}^{T} \|\nabla h_t(W_t)\|_{\mathcal{V}}^2}\right) . \tag{1}$$

There are many existing algorithms which satisfy this property; we provide a concrete example and its updates in our framework for completeness in Appendix C.2. Using such an algorithm in Algorithm 1 immediately yields the following regret guarantee.

**Proposition 1.** *Let $\mathcal{A}$ be a static regret algorithm for Hilbert spaces satisfying Equation (1). For any $G > 0$, any sequence of $G$-Lipschitz losses $\ell_1, \ldots, \ell_T$, and any sequence $u_1, \ldots, u_T$ in $\mathcal{W}$, and $U \in L(\mathcal{H}, \mathcal{W})$ satisfying $u_t = U\phi(t)$, Algorithm 1 applied with $\mathcal{A}$ guarantees*

$$\mathrm{R}_{\mathrm{T}}(u_1, \ldots, u_T) = \widetilde{\mathrm{R}}_T(U) = \widetilde{\mathcal{O}}\left(\|U\|_{\mathrm{HS}} \sqrt{\sum_{t=1}^{T} \|g_t\|_{\mathcal{W}}^2 \, k(t,t)}\right), \tag{2}$$

*where $g_t \in \partial \ell_t(w_t)$ for all $t$.*

## 4.1  Controlling the Trade-offs Induced by $\mathcal{H}$

Proposition 1 demonstrates a clear trade-off between the RKHS norm $\|u\|_{\mathcal{H}}$ and the associated kernel $k(t,t)$ induced by the choice of function space: smaller the RKHS norms correspond to larger function spaces, hence higher values of $k(t,t)$. In order to obtain the optimal $\mathcal{O}(\sqrt{P_T T})$ dynamic regret, we need to design a kernel such that $\|U\|_{\mathrm{HS}} = \|u\|_{\mathcal{H}^d} = \mathcal{O}(\sqrt{P_T})$ and $k(t,t)$ is

---

[3]Here and in the following the $\widetilde{\mathcal{O}}$ notation will hide polylogarithmic factors.

controlled for all $t$. Throughout this section, we will assume for simplicity that $d = 1$ but note that the extension to $d > 1$ is straightforward via the coordinate-wise extension in Section 2.

Recall that a *translation invariant* kernel over $\mathbb{R}$ is characterized by the Fourier transform of its *spectral density* $Q$ [51], where $Q$ is a real non-negative integrable function. In particular, a translation invariant kernel and its associated norm are

$$k(t, t') = \widehat{Q}(t - t') = \int_{\mathbb{R}} Q(\omega) e^{-2\pi i \omega (t - t')} d\omega, \quad \|f\|_{\mathcal{H}}^2 = \int_{\mathbb{R}} \frac{|\widehat{f}(\omega)|^2}{Q(\omega)} d\omega, \tag{3}$$

where we use the short-hand notation $\widehat{g} = F[g]$ to denote the Fourier transorm of a function $g(\cdot)$. The intuition behind focusing on translation-invariant kernels is that the associated norm provides a natural connection to the $\sqrt{MP_T}$ dependencies we would like like to achieve. Indeed, observe that with spectral density $Q(\omega) \approx 1/\omega$, we would have via Parseval's identity and the fact that $F[f'(x)](\omega) = 2\pi i \omega \widehat{f}(\omega)$ that

$$\|u\|_{\mathcal{H}}^2 \approx \int_{\mathbb{R}} \omega \widehat{u}(\omega) \overline{\widehat{u}(\omega)} d\omega \leq \int_{\mathbb{R}} |\nabla u(t)| |u(t)| dt \leq \sup_t |u(t)| \, \|\nabla u\|_{L^1}, \tag{4}$$

which is the continuous-time analogue of $MP_T$. The key challenge is to choose an integrable $Q(\omega)$ which suitably trades off the comparator norm $\|u\|_{\mathcal{H}}^2$ and the magnitude of the associated kernel entries $k(t, t)$. Unfortunately, these trade-offs are non-trivial using standard translation-invariant kernels, as shown in the following example.

**Example 2** (Existing kernels lead to sub-optimal trade-offs [51])**.** *At first glance, the spline kernel seems like a natural candidate since it has $\|u\|_{\mathcal{H}}^2 = \|\nabla u\|_{L^2}^2 = \mathcal{O}(\sum_t \|u_t - u_{t-1}\|^2)$ (Theorem 8). However, the spline kernel also has $k(t, t) = t$, leading to a suboptimal rate in Proposition 1. On the other hand, for the classical translation invariant kernels such as the Gaussian or the Matern kernels, we have $k(t, t) = \mathcal{O}(1)$ but $\|u\|_{\mathcal{H}}^2 = \|u\|_{L^2}^2 + \sum_{n \geq 1} c_n \|\nabla^n u\|^2$ for $c_n$ positive and summable. In this case, note that $k(t, t)$ has the good rate but $\|u\|_{\mathcal{H}}^2 \geq \|u\|_{L^2}^2$ and $\|u\|_{L^2}^2 = c^2 T$ already for constant comparators on $[0, T]$, $u(t) = c1_{[0,T]}(t)$, $c > 0$, precluding the optimal rate.*

Given the above, we next turn our attention to designing a new kernel that will achieve the desired trade-offs. Since we need to find a delicate balance in the trade-off of $\|u\|_{\mathcal{H}}$ and $k(t, t)$ to achieve optimal rates, in the first part of the section we first derive a result that identifies general sufficient conditions to bound the RKHS norm of a translation invariant kernel in terms of the *continuous path-length* $\|\nabla u\|_{L^1} = \int |\nabla u(t)| dt$ (Theorem 3). Then, in Proposition 2, we design an explicit kernel satisfying such conditions, leading to a trade-off of $\mathcal{O}(\sqrt{\|\nabla u\|_{L^1} T})$. Finally, in Theorem 4 we show that under mild conditions (which are satisfied by the kernel in the Proposition 2), it is always possible to find an $u(\cdot) \in \mathcal{H}$ such that $u(t) = u_t$ for all $t$ and $\|\nabla u\|_{L^1} = \mathcal{O}(\sum_t \|u_t - u_{t-1}\|_{\mathcal{W}})$, so achieving dynamic regret scaling with $\|u\|_{\mathcal{H}} = \mathcal{O}(\|\nabla u\|_{L^1})$ recovers the usual path-length. The proof of the following theorem can be found in Appendix E.1

**Theorem 3.** *Let $Q : \mathbb{R} \to \mathbb{R}_+$ be an integrable strictly positive even function on $\mathbb{R} \setminus \{0\}$ and such that $R(x) := 2\pi/(x(1 + (x/2\pi)^{2m}) Q(x))$ is also integrable for some $m \in \mathbb{N}$, $m \geq 1$. Let $k$ be defined in terms of $Q$ as in Eq. (3). Then $k$ is a translation invariant universal kernel with $k(t, t) \leq \|Q\|_{L^1}$ for all $t \in \mathbb{R}$. The RKHS $\mathcal{H}$ associated to $k$ contains the space of finitely supported functions with bounded derivatives up to order $2m$, and moreover, for any $T > 0$ and any $2m$-times differentiable function $f$ that is supported on $[0, T + 1]$,*

$$\|f\|_{\mathcal{H}}^2 \leq c(T) \|\nabla f\|_{L^1} \|f - \nabla^{2m} f\|_{L^{\infty}},$$

*where $c(T) := \|F[R]\|_{L^1([-T-1, T+1])}$. If $R$ is monotonically decreasing on $(0, \infty)$, then,*

$$c(T) \leq \inf_{\alpha > 0} 2\pi (T+1)^2 \int_0^{\alpha} R(x) x \, dx + \frac{2}{\pi} \int_{\alpha}^{\infty} \frac{R(x)}{x} dx, \quad \forall T > 0$$

With this in hand, the following proposition provides an example of spectral density $Q$ which will leads to the desired dependency $\|u\|_{\mathcal{H}}^2 = \mathcal{O}(\|\nabla u\|_{L^1})$, up to poly-logarithmic terms. Proof can be found in Appendix E.4.

**Proposition 2.** *Let* $Q : \mathbb{R} \to \mathbb{R}_+$ *be defined as*

$$Q(\omega) = \frac{1/4 \, \log \log \pi}{|\omega| \, (1 + |\omega|^2/4\pi^2)^{\frac{1}{4}} \, \log(\pi + |\omega|^{-\frac{1}{2}}) \, \log^2 \log(\pi + |\omega|^{-\frac{1}{2}})} \, .$$

*Then we can apply Theorem 3 with* $m = 1$*: the function* $k$ *defined in terms of* $Q$ *as in Eq. (3) is a translation invariant kernel with* $k(t,t) \leq 8\pi^2$*,* $\forall t \in \mathbb{R}$*; the associated RKHS norm satisfies*

$$\|f\|_{\mathcal{H}}^2 \ \leq \ c^2 \, \|\nabla f\|_{L^1} \, \|f - \nabla^2 f\|_{L^\infty} \, (\ln(1+T) \ln \ln(1+T))^2,$$

*for any* $f$ *that is* 2*-times differentiable and supported in* $[0, T+1]$*, where* $T > 2$ *and* $c \leq (2\pi e)^2$*.*

A notable property of the kernel characterized by Proposition 2 is that it is *horizon independent*, requiring no upper bound on $T$ to control $\|f\|_{\mathcal{H}}^2$ and $k(t,t)$. This is a non-trivial property to guarantee using existing methods without resorting to the doubling trick, which is well-known to perform poorly in practice. The intuitions behind the choice of $Q(\omega)$ follow from the discussion above: we would like to set $Q(\omega) \approx 1/|\omega|$ so that $\|u\|_{\mathcal{H}}$ relates to the path-length via Equation (4), but this would not be a valid choice because $Q(\omega) = 1/|\omega|$ is not integrable. Proposition 2 adds a small bit of additional regularization to ensure that $Q(\omega)$ is integrable while remaining close to $1/|\omega|$. We provide additional intuition on the choice of regularization in Appendix D.

A subtlety that we have glossed over thus far is that the *continuous path-length*, $\|\nabla u\|_{L^1} = \int \|\nabla u(t)\|_{\mathcal{W}} \, dt$, does not necessarily compare favorably to the classic *discrete path-length* $P_T = \sum_t \|u_t - u_{t-1}\|_{\mathcal{W}}$ since the function may vary wildly between the interpolated points. The next theorem shows that we can always find a function such that $\|\nabla u\|_{L^1} = \mathcal{O}(P_T)$. Proof can be found in Appendix E.2.

**Theorem 4.** *Let* $v_1, \ldots, v_T \in \mathbb{R}^d$ *and let* $\mathcal{H}$ *be the RKHS associated to kernel* $k$ *contain finitely supported functions with bounded derivatives up to order* $2m$*, with* $m \in \mathbb{N}$*,* $m \geq 1$*. Then there exists a function* $u \in \mathcal{H}$ *supported on* $[0, T+1]$*, such that* $u(t) = v_t$ *for all* $t \in [T]$ *and*

$$\|\nabla u\|_{L^1} \leq C\|v_1\|_{\mathcal{W}} + C\sum_{t=2}^{T} \|v_t - v_{t-1}\|_{\mathcal{W}}, \qquad \left\|u - \nabla^{2m} u\right\|_{L^\infty} \leq C' \max_t \|v_t\|_{\mathcal{W}} \, .$$

*with* $C, C'$ *depending only on* $m$ *and given in explicitly in the proof.*

The theorem demonstrates that the continuous path-length can be bound by the usual discrete path-length under mild assumptions on the RKHS that are satisfied by the translation invariant kernel with spectral density $Q$ chosen according to Theorem 3. Based on this observation, we immediately see that the RKHS characterized by the kernel in Proposition 2 satisfies the condition of the theorem, and has RKHS norm satisfying $\|u\|_{\mathcal{H}}^2 = \widetilde{\mathcal{O}}(\|\nabla u\|_{L^1} \|u - \nabla^2 u\|_{L^\infty}) = \widetilde{\mathcal{O}}(M^2 + M \sum_{t=2}^{T} \|u_t - u_{t-1}\|_{\mathcal{W}})$ where $M = \max_t \|u_t\|_{\mathcal{W}}$.

**Optimal Path-length Dependencies.** Applying our reduction Proposition 1 with the translation invariant kernel characterized by Proposition 2, followed by Theorem 4 to bound $\|\nabla u\|_{L^1} = \mathcal{O}(\sqrt{M^2 + MP_T})$ immediately yields the following dynamic regret guarantee for OLO.

**Theorem 5.** *Let* $G > 0$ *and apply the algorithm characterized in Proposition 1 with the kernel with spectral density described by Proposition 2. Then for any* $T > 3$*, and any sequence* $g_1, \ldots, g_T$ *satisfying* $\|g_t\|_{\mathcal{W}} \leq G$ *and sequence* $u_1, \ldots, u_T$ *in* $\mathcal{W} \subseteq \mathbb{R}^d$*, the dynamic regret is bounded as*

$$R_T(u_1, \ldots, u_T) = \widetilde{\mathcal{O}}\left(\sqrt{(M^2 + MP_T) \sum_{t=1}^{T} \|g_t\|_{\mathcal{W}}^2}\right),$$

*where* $M = \max_t \|u_t\|_{\mathcal{W}}$ *and* $P_T = \sum_{t=2}^{T} \|u_t - u_{t-1}\|_{\mathcal{W}}$*.*

As observed in Section 4, the kernel that produces this result is horizon independent, so the algorithm described above requires no prior knowledge of $T$. This is in fact the first dynamic regret algorithm we are aware of that achieves the optimal $\sqrt{P_T}$ dependence in the absence of prior knowledge of $T$ without resorting to a doubling trick. Likewise, in Appendix C.3 we show that these guarantees extend immediately to *scale-free* guarantees using the gradient-clipping argument of [12]. These are the first scale-free dynamic regret guarantees that we are aware of that achieve the optimal $\sqrt{P_T}$ dependencies.

# 5 Curved Losses

An advantage of our reduction over the dynamic-to-static reduction of Jacobsen and Orabona [26] is that, by preserving the curvature of the losses, our reduction allows us to apply (quasi)second-order methods like Online Newton Step (ONS) [20].

**Exp-concave Losses** The following proposition shows that exp-concave losses retain the crucial property required to apply ONS under our reduction (proof in Appendix F).

**Proposition 3.** *Let $\ell_t : \mathcal{W} \to \mathbb{R}$ be a $\beta$-exp-concave function, let $\mathcal{H}$ be an RKHS with feature map $\phi(t) \in \mathcal{H}$, and define $\widetilde{\ell}_t(W) = \ell_t(W\phi(t))$ for $W \in L(\mathcal{H}, \mathcal{W})$. Then for any $X, Y \in L(\mathcal{H}, \mathcal{W})$,*

$$\widetilde{\ell}_t(X) - \widetilde{\ell}_t(Y) \le \left\langle \nabla \widetilde{\ell}_t(X), X - Y \right\rangle_{\mathrm{HS}} - \frac{\beta}{2} \left\langle \nabla \widetilde{\ell}(X), X - Y \right\rangle_{\mathrm{HS}}^2 .$$

Note that this is precisely the curvature assumption that is required to run Kernelized ONS (KONS) [7, 8]. Hence, applying our reduction Theorem 1 with KONS to the loss sequence $\widetilde{\ell}_1, \ldots, \widetilde{\ell}_T$ leads immediately to the following dynamic regret guarantee, adapted from Calandriello et al. [8, Theorem 1].

**Theorem 6.** *Let $\ell_1, \ldots, \ell_T$ be a sequence of $\beta$-exp-concave losses. For any sequence $u_1, \ldots, u_T \in \mathcal{W}$ and $U \in L(\mathcal{H}, \mathcal{W})$ satisfying $u_t = U\phi(t)$ for all $t$, Algorithm 1 applied with KONS guarantees*

$$R_T(u_1, \ldots, u_T) = \mathcal{O}\left( \lambda \|U\|_{\mathrm{HS}}^2 + d_{\mathrm{eff}}\left( \frac{\lambda}{\beta G^2 k_{\max}} \right) \frac{\ln\left(2\beta G^2 k_{\max} T\right)}{\beta} \right),$$

*where $G \ge \|\nabla \ell_t(w)\|$ for all $w \in W$, $k_{\max} = \max_t k(t,t)$, $d_{\mathrm{eff}}(\lambda) = \mathrm{Tr}\left(K_T (K_T + \lambda I)^{-1}\right)$, and $K_T = (\langle \nabla \ell_i(w_i), \nabla \ell_j(w_j) \rangle_{\mathcal{W}} k(i,j))_{i,j=1}^T \in \mathbb{R}^{T \times T}$.*

In the previous section, we observed a direct trade-off between the complexity of the comparator—measured in terms of $\|u\|_{\mathcal{H}}$—and a term measuring the complexity of the RKHS, $\max_t k(t,t)$. Here we again see a trade-off in measures of complexity, but now the complexity of the RKHS is characterized by the *effective dimension* $d_{\mathrm{eff}}(\lambda)$. Loosely speaking, the effective dimension represents the number of "non-negligable directions" spanned by the features $\phi(1), \ldots, \phi(T)$, characterized by the number of eigenvectors of $K_T$ associated with non-negligable eigenvalues relative to $\lambda$.

**Strongly-convex Losses** Interestingly, for strongly-convex losses it can be shown that an analogous curvature condition to Proposition 3 holds under our reduction as well, leading to an analogous result to Theorem 6. Indeed, the main difference is that in the strongly-convex setting, one uses the feature covariance $\lambda I + \sum_{s=1}^t \phi(t) \otimes \phi(t)$ to define a weighted norm while KONS the covariance matrix of the product kernel associated with features $\widetilde{\phi}(t) = g_t \otimes \phi(t)$. Applying a similar argument then leads to a guarantee which is analogous to Theorem 6 (see Appendix F.1 for more details).

**Online Linear Regression** Similar results also apply in the context of online regression. In that setting, at the start of round $t$ the learner first observes a *context* $x_t \in \mathcal{T}$, then predicts a $\widehat{y}_t \in \mathbb{R}$, and incurs a loss $\ell_t(\widehat{y}) = \frac{1}{2}(y_t - \widehat{y})^2$. In this setting, our reduction recovers *kernelized online regression*, by letting $\widehat{y}_t = \langle f, \phi(x_t) \rangle$ where $f \in L(\mathcal{H}, \mathbb{R})$ and $\phi(x_t) \in \mathcal{H}$ is the feature map associated with $\mathcal{H}$. Applying the Kernelized Vovk-Azoury-Warmuth forecaster [2, 27, 50] guarantees regret of the same form as above. The result follows from Jézéquel et al. [27, Proposition 1 and Proposition 2].

**Proposition 4.** *Let $\mathcal{W} = \mathbb{R}$ and for all $t$ let $\ell_t(\widehat{y}) = \frac{1}{2}(y_t - \widehat{y})^2$. Then for any sequence $(x_1, y_1), \ldots, (x_T, y_T)$ in $\mathcal{T} \times \mathcal{W}$ and any benchmark sequence $\mathring{y}_1, \ldots, \mathring{y}_T$ in $\mathbb{R}$ and $u \in \mathcal{H}$ satisfying $\mathring{y}_t = \langle u, \phi(x_t) \rangle$ for all $t$, the Kernelized VAW Forecaster guarantees*

$$\sum_{t=1}^T (\ell_t(\widehat{y}_t) - \ell_t(\mathring{y}_t)) \le \lambda \|u\|_{\mathcal{H}}^2 + d_{\mathrm{eff}}(\lambda) y_{\max}^2 \log\left( e + \frac{eT k_{\max}^2}{\lambda} \right),$$

*where $k_{\max} = \max_t k(t,t)$ and $y_{\max}^2 = \max_t y_t^2$.*

It is known that the dependence on $d_{\mathrm{eff}}(\lambda)$ for kernel ridge regression is optimal [30], demonstrating that these trade-offs are unimprovable in the context of dynamic regret as well.

In each of the results above, the main trade-off is between the comparator norm $\lambda \|u\|_{\mathcal{H}}^2$ and the effective dimension $d_{\text{eff}}(\lambda)$. As an illustrative example, the following shows that the linear spline kernel can achieve non-trivial *squared* path-length guarantees, which were recently shown to be unattainable in the OLO setting [26].

**Example 3.** *The linear spline kernel $k(s,t) = \min(s,t)$ has well-known RKHS norm of $\|u\|_{\mathcal{H}}^2 = \|\nabla u\|_{L^2}^2 = \int |\nabla u(t)|^2 dt$. Moreover, in [Appendix F.2](#) we show that we can bound $\|\nabla u\|_{L^2} \le \mathcal{O}(\sqrt{\sum_t |\mathring{y}_t - \mathring{y}_{t-1}|_{\mathcal{W}}^2}) := C_T'$ and that the effective dimension is $d_{\text{eff}}(\lambda) = \mathcal{O}(T/\sqrt{\lambda})$ ([Theorems 8](#) and [9](#) respectively). Optimally tuning $\lambda$ leads to $R_T(\mathring{y}_1, \ldots, \mathring{y}_T) = \widetilde{\mathcal{O}}(T^{2/3}(C_T')^{2/3})$ which matches the minimax optimal rate for forecasting in the class of discrete Sobolev sequences of bounded variation [3, 44]. Note that $\lambda$ can be tuned without data-dependent prior knowledge using mixture-of-experts and a simple clipping argument [25, 34].*

In the special case of the 1-dimensional squared loss $\ell_t(y) = \frac{1}{2}(y_t - y)^2$, it is possible to achieve $R_T(\mathring{y}_1, \ldots, \mathring{y}_T) = \widetilde{\mathcal{O}}(T^{1/3} C_T^{2/3})$ where $C_T = \sum_t |\mathring{y}_t - \mathring{y}_{t-1}|$ is the (unsquared) path-length of the benchmark *predictions*, and this bound is minimax optimal among the class of discrete TV-bounded sequences, which is more general than the Sobolev class in the example above [3, 54]. Designing a kernel with a suitable effective dimension to achieve this trade-off has proven non-trivial and is left as a direction for future work.

## 6 Directional Adaptivity

An exciting benefit of reducing to static regret is that we can leverage more "exotic" static regret guarantees to uncover new and interesting trade-offs in dynamic regret, essentially for free. For example, in recent years there has been an interest in algorithms which adapt to the *directional* covariance between the comparator and the losses [13, 15, 16, 37, 49], to guarantee

$$R_T(u) = \widetilde{\mathcal{O}}\Big(\sqrt{d \sum_{t=1}^T \langle g_t, u \rangle_{\mathcal{W}}^2}\Big).$$

These bounds recover the usual $\widetilde{\mathcal{O}}\big(\|u\|_{\mathcal{W}} \sqrt{\sum_{t=1}^T \|g_t\|_{\mathcal{W},*}^2}\big)$ bounds in the worst case, but could be significantly smaller if the comparator tends to be orthogonal to the losses. Passing from dynamic regret to static regret via [Theorem 1](#), the following proposition shows that guarantees of this form translate into dynamic regret guarantees which naturally decouple the comparator variability $\|U\|_{\text{HS}}$ from the a *per-round* directional variance penalty $\sum_{t=1}^T \langle g_t, u_t \rangle_{\mathcal{W}}^2$. The full statement and proof of this result can be found in [Appendix G](#).

**Proposition 5.** *Let $\ell_1, \ldots, \ell_T$ be an arbitrary sequence of $G$-Lipschitz convex loss functions over $\mathcal{W}$. There exists an algorithm such that for any sequence of $u_1, \ldots, u_T$ in $\mathcal{W}$ and $U \in L(\mathcal{H}, \mathcal{W})$ satisfying $u_t = U\phi(t)$ for all $t$, the dynamic regret $R_T(u_1, \ldots, u_T)$ is bounded by*

$$\widetilde{\mathcal{O}}\Big(L_k d_{\text{eff}}(\lambda) + \sqrt{d_{\text{eff}}(\lambda) \Big[(\lambda + L_k^2) \|U\|_{\text{HS}}^2 + \sum_{t=1}^T \langle g_t, u_t \rangle_{\mathcal{W}}^2\Big] \ln\Big(e + \frac{e\lambda_{\max}(K_T)}{\lambda}\Big)}\Big),$$

*where $g_t \in \partial \ell_t(w_t)$, $L_k^2 = G^2 \max_t k(t,t)$, $K_T = (\langle g_t, g_s \rangle_{\mathcal{W}} k(t,s))_{t,s \in [T]}$, and $d_{\text{eff}}(\lambda) = \text{Tr}(K_T(\lambda I + K_T)^{-1})$.*

## 7 Discussion

In this paper we developed a general reduction from dynamic regret to static regret based on embedding the comparator sequence as a function in an RKHS. We showed that the optimal $\sqrt{P_T}$ path-length dependence of can be obtained via a carefully designed translation-invariant kernel. We also developed new scale-free and directionally-adaptive guarantees for online linear optimization and $\|u\|_{\mathcal{H}}^2 + d_{\text{eff}}(\lambda) \ln T$ bounds for losses with curvature.

There are many promising directions for future work. As noted in [Section 4](#), if implemented naively, the algorithms described here could be prohibitively expensive to run in practice. Future work should study how to best leverage kernel approximation techniques or sparse dictionary methods to achieve the standard $\mathcal{O}(d \ln T)$ per-round computation without ruining the desired regret bounds. We also anticipate many interesting directions for future work by investigating the rich intersections between online learning, kernel methods, and signal processing that our reduction brings to light.

## Acknowledgments and Disclosure of Funding

AJ and NCB acknowledge the financial support from the FAIR project, funded by the NextGenerationEU program within the PNRR-PE-AI scheme (M4C2, investment 1.3, line on Artificial Intelligence), the MUR PRIN grant 2022EKNE5K (Learning in Markets and Society), funded by the NextGenerationEU program within the PNRR scheme (M4C2, investment 1.1), the EU Horizon CL4-2022-HUMAN-02 research and innovation action under grant agreement 101120237, project ELIAS (European Lighthouse of AI for Sustainability), and the One Health Action Hub, University Task Force for the resilience of territorial ecosystems, funded by Università degli Studi di Milano (PSR 2021-GSA-Linea 6). AR acknowledges support from the European Research Council (grant REAL 947908).

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

## A  Additional Functional Analysis Background

In this section we briefly recall some additional definitions and background from functional analysis, which will be useful for understanding the proofs of our results but were not relevant for the main text.

**Additional Notations.**   The *dual space* of a normed space $\mathcal{H}$ is the space of bounded linear functionals $\mathcal{H}^* = L(\mathcal{H}, \mathbb{R})$, and the associated norm is the *dual norm* $\|g\|_{\mathcal{H},*} = \sup_{\|h\|_{\mathcal{H}}=1} g(h)$ for any $g \in \mathcal{H}^*$. An operator $\mathcal{T} : \mathcal{H} \to \mathcal{W}$ is *Hilbert-Schmidt* if for any orthonormal basis $\{h_i\}_i$ of $\mathcal{H}$ we have $\|T\|_{\mathrm{HS}}^2 := \sum_i \|Th_i\|_{\mathcal{W}}^2 < \infty$. The space $L(\mathcal{H}, \mathcal{W})$ is itself a Hilbert space with inner product $\langle A, B \rangle_{\mathrm{HS}} = \sum_i \langle Ah_i, Bh_i \rangle_{\mathcal{W}}$.

**A Brief Review of Reproducing Kernel Hilbert Spaces.**   A linear functional $\varphi \in L(\mathcal{H}, \mathbb{R})$ is *bounded* if there exists a constant $M$ such that $|\varphi(x)| \leq M \|x\|_{\mathcal{H}}$ for all $h \in \mathcal{H}$. A *reproducing kernel Hilbert space* (RKHS) is a Hilbert space $\mathcal{H}$ of functions $h : X \to \mathbb{R}$ for which the *evaluation functional* $\delta_x : h \mapsto h(x)$ is bounded for all $x \in X$. For any such space, Riesz representation theorem tells us that for any $x \in X$ there is a unique $k_x \in \mathcal{H}$ such that $\delta_x(h) = \langle h, k_x \rangle_{\mathcal{H}}$ for all $h \in \mathcal{H}$. The function $k(x, x') = k_x(x')$ is called the *reproducing kernel* associated with $\mathcal{H}$. The reproducing kernel is often expressed in terms of the *feature map* $\phi(x) = k_x \in \mathcal{H}$ as $k(x, x') = \langle \phi(x), \phi(x') \rangle_{\mathcal{H}}$.

We will often be interested in functions taking values in $\mathcal{W} \subseteq \mathbb{R}^d$. In this case the preceding discussion can be extended in a straightforward way by considering a coordinate-wise extension. In particular, observe that in this setting an operator $W \in L(\mathcal{H}, \mathcal{W})$ can be represented as a tuple $(W_1, \ldots, W_d)$ such that $W_i \in L(\mathcal{H}, \mathbb{R}) = \mathcal{H}^*$ for each $i \in [d]$. Riesz representation theorem then tells us that there is a $w_i \in \mathcal{H}$ such that $W_i(h) = \langle w_i, h \rangle_{\mathcal{H}}$ for any $h \in \mathcal{H}$, so using the reproducing property we have $W_i(\phi(t)) = \langle w_i, \phi(t) \rangle_{\mathcal{H}} = w_i(t)$. Hence, each $W \in L(\mathcal{H}, \mathcal{W})$ is identified by a tuple $(W_1, \ldots, W_d) \in \mathcal{H}^d$ and we can write

$$w(t) = \big( \langle w_1, \phi(t) \rangle_{\mathcal{H}}, \ldots, \langle w_d, \phi(t) \rangle_{\mathcal{H}} \big) = \big( W_1(\phi(t)), \ldots, W_d(\phi(t)) \big) = W\phi(t) \in \mathcal{W}.$$

Note the that space $L(\mathcal{H}, \mathcal{W})$ is itself a Hilbert space when equipped with the Hilbert-Schmidt norm, which in the coordinate-wise extension above can be expressed as $\|W\|_{\mathrm{HS}}^2 = \|w\|_{\mathcal{H}^d}^2 = \sum_{i=1}^d \|w_i\|_{\mathcal{H}}^2$. This can be seen by definition of the Hilbert-Schmidt norm: let $\{h_i\}_i$ be an orthonormal basis of $\mathcal{H}$, then

$$\begin{aligned}
\|W\|_{\mathrm{HS}}^2 &= \sum_i \|Wh_i\|_{\mathcal{W}}^2 = \sum_i \|(W_1(h_i), \ldots, W_d(h_i))\|_{\mathcal{W}}^2 \\
&= \sum_i \|(\langle w_1, h_i \rangle_{\mathcal{H}}, \ldots, \langle w_d, h_i \rangle_{\mathcal{H}})\|_{\mathcal{W}}^2 \\
&= \sum_i \sum_{j=1}^d \langle w_j, h_i \rangle_{\mathcal{H}}^2 = \sum_{j=1}^d \sum_i \langle w_j, h_i \rangle_{\mathcal{H}}^2 \\
&= \sum_{j=1}^d \|w_j\|_{\mathcal{H}}^2 := \|w\|_{\mathcal{H}^d}^2,
\end{aligned}$$

where the last line uses Parseval's identity.

## B  Recovering Jacobsen and Orabona [26]

In this section we demonstrate that the reduction in Jacobsen and Orabona [26] is equivalent to the special case of our framework. Note that we assume linear losses in this section because the reduction of Jacobsen and Orabona [26] is only defined for linear losses (or by linearizing the losses $\ell_t$ via convexity).

Let $e_t \in \mathbb{R}^T$ be the $t^{\mathrm{th}}$ standard basis vector and consider Algorithm 1 with $\mathcal{H} = \mathbb{R}^T$, $\langle A, B \rangle_{\mathrm{HS}} = \mathrm{Tr}\left(A^\top B\right)$, kernel feature map $\phi(t) = e_t \in \mathbb{R}^T$, and linear losses $W \mapsto \langle G_t, W \rangle_{\mathrm{HS}}$ for $G_t =$

$g_t \otimes e_t = g_t e_t^\top \in \mathbb{R}^{d \times T}$. Then for any sequence $u_1, \ldots, u_T$ in $\mathbb{R}^d$, let $U = (u_1 \quad \ldots \quad u_T) \in \mathbb{R}^{d \times T}$ and observe that we can write $u_t = U\phi(t)$. Moreover,

$$\sum_{t=1}^T \langle g_t, u_t \rangle = \sum_{t=1}^T \langle g_t, U\phi(t) \rangle = \sum_{t=1}^T \mathrm{Tr} \left( \phi(t) g_t^\top U \right) = \sum_{t=1}^T \mathrm{Tr} \left( \left( g_t \phi(t)^\top \right)^\top U \right)$$

$$= \sum_{t=1}^T \left\langle g_t \phi(t)^\top, U \right\rangle_{\mathrm{HS}} = \sum_{t=1}^T \langle G_t, U \rangle_{\mathrm{HS}} .$$

Similarly, suppose $\mathcal{A}$ is an online learning algorithm and let $W_t = \left( w_t^{(1)} \quad \ldots \quad w_t^{(T)} \right) \in \mathbb{R}^{d \times T}$ denote its output on round $t$. Suppose on round $t$ we play $w_t = W_t \phi(t) = w_t^{(t)}$. Then

$$\sum_{t=1}^T \langle g_t, w_t \rangle = \sum_{t=1}^T \langle g_t, W_t \phi(t) \rangle = \mathrm{Tr} \left( \phi_t g_t^\top W_t \right) = \left\langle g_t \phi(t)^\top, W_t \right\rangle_{\mathrm{HS}} = \sum_{t=1}^T \langle G_t, W_t \rangle_{\mathrm{HS}} .$$

Thus,

$$\mathrm{R}_{\mathrm{T}}(u_1, \ldots, u_T) = \sum_{t=1}^T \langle g_t, w_t - u_t \rangle = \sum_{t=1}^T \langle G_t, W_t - U \rangle_{\mathrm{HS}} = \widetilde{\mathrm{R}}_T^{\mathcal{A}}(U).$$

To see why this is precisely equivalent to the reduction of Jacobsen and Orabona [26], observe that their reduction is simply phrased in terms of the "flattened" versions of each of the above quantities, and can be interpreted as working in the finite-dimensnional RKHS over $\widetilde{\mathcal{H}} = \mathbb{R}^{dT}$ with $\langle x, y \rangle_{\widetilde{\mathcal{H}}} = \langle x, y \rangle$ being the canonical inner product on $\mathbb{R}^{dT}$. In particular, they instead define

$$\widetilde{u} = \mathrm{vec}\,(U) = \begin{pmatrix} u_1 \\ \vdots \\ u_T \end{pmatrix} \in \mathbb{R}^{dT}, \; \widetilde{w}_t = \mathrm{vec}\,(W_t) = \begin{pmatrix} w_t^{(1)} \\ \vdots \\ w_t^{(T)} \end{pmatrix} \in \mathbb{R}^{dT}, \; \widetilde{g}_t = \mathrm{vec}\,(G_t) = \begin{pmatrix} \mathbf{0} \\ \vdots \\ g_t \\ \mathbf{0} \\ \vdots \end{pmatrix} \in \mathbb{R}^{dT},$$

and run an algorithm $\widetilde{\mathcal{A}}$ defined on $\widetilde{\mathcal{H}} = \mathbb{R}^{dT}$ against the losses $\widetilde{g}_t$. As shown in their Proposition 1, under this setup it holds that $\sum_{t=1}^T \langle g_t, w_t - u_t \rangle = \sum_{t=1}^T \langle \widetilde{g}_t, \widetilde{w}_t - \widetilde{u} \rangle$, from which it immediately follows that

$$R_T(u_1, \ldots, u_T) = \overbrace{\sum_{t=1}^T \langle G_t, W_t - U \rangle_{\mathrm{HS}} = \widetilde{\mathrm{R}}_T^{\mathcal{A}}(U)}^{\text{Theorem 1}}$$

$$= \underbrace{\sum_{t=1}^T \langle \widetilde{g}_t, \widetilde{w}_t - \widetilde{u} \rangle_{\widetilde{\mathcal{H}}} = \widetilde{\mathrm{R}}_T^{\widetilde{\mathcal{A}}}(\widetilde{u})}_{\text{Jacobsen and Orabona [26, Proposition 1]}}.$$

# C Proofs for Section 4 (Linear Losses)

## C.1 Proof of Theorem 2

**Theorem 2.** *Let $\mathcal{A}$ be an online learning algorithm defined on Hilbert space $\mathcal{V}$. Suppose that for any sequence of convex loss functions $h_1, \ldots, h_T$ on $\mathcal{V}$, $\mathcal{A}$ obtains a bound on the static regret of the form $\widetilde{R}_T(U) \le B_T\left(\|U\|_{\mathcal{V}}, \|\nabla h_1(W_1)\|_{\mathcal{V}}, \ldots, \|\nabla h_T(W_T)\|_{\mathcal{V}}\right)$ for any comparator $U \in \mathcal{V}$ and some function $B_T : \mathbb{R}^{T+1}_{\ge 0} \to \mathbb{R}$, where $\nabla h_t(W_t) \in \partial h_t(W_t)$ for all $t$. If we apply $\mathcal{A}$ in $\mathcal{V} = L(\mathcal{H}, \mathcal{W})$ with $\|\cdot\|_{\mathcal{V}} = \|\cdot\|_{\mathrm{HS}}$, then for any sequence $u_1, \ldots, u_T$ in $\mathcal{W}$ and $U \in L(\mathcal{H}, \mathcal{W})$ satisfying $u_t = U\phi(t)$ for all $t$, Algorithm 1 with $\mathcal{A}$ guarantees*

$$R_T(u_1, \ldots, u_T) \le B_T\left(\|U\|_{\mathrm{HS}}, \|g_1\|_{\mathcal{W}}\sqrt{k(1,1)}, \ldots, \|g_T\|_{\mathcal{W}}\sqrt{k(T,T)}\right),$$

*where $g_t \in \partial \ell_t(w_t)$ for all $t$, and $k(\cdot, \cdot)$ is the* reproducing kernel *associated to the space $\mathcal{H}$.*

*Proof.* We note that $L(\mathcal{H}, \mathcal{W})$ is a separable Hilbert space for separable $\mathcal{H}$ and $\mathcal{W}$. Moreover, $\widetilde{\ell}_t$ is differentiable, with derivative

$$\nabla \widetilde{\ell}_t(W) = \nabla \ell_t(W\phi(t)) \otimes \phi(t) \in L(\mathcal{H}, \mathcal{W}),$$

where $\otimes$ is the tensor product. Then applying algorithm $\mathcal{A}$ to the loss sequence $(\widetilde{\ell}_t)_t$, we obtain

$$\widetilde{R}_T(U) \le \phi\left(\|U\|_{L(\mathcal{H}, \mathcal{W})}, \|\nabla \ell_1(w_1) \otimes \phi(1)\|_{L(\mathcal{H}, \mathcal{W})}, \ldots, \|\nabla \ell_t(w_T) \otimes \phi(T)\|_{L(\mathcal{H}, \mathcal{W})}\right).$$

The proof is concluded by noting that $R_T(u_1, \ldots, u_T) = \widetilde{R}_T(U)$ by Theorem 1, and that

$$\|\nabla \ell_t(w_t) \otimes \phi(t)\|_{L(\mathcal{H}, \mathcal{W})} = \|\nabla \ell_t(w_t)\|_{\mathcal{W}} \|\phi(t)\|_{\mathcal{H}},$$

since $u \otimes v \in L(\mathcal{U}, \mathcal{V})$ is a rank one operator and so $\|u \otimes v\|_{L(\mathcal{U}, \mathcal{V})} = \|u\|_{\mathcal{U}}\|v\|_{\mathcal{V}}$, for any $u, v \in \mathcal{U}, \mathcal{V}$ and $\mathcal{U}, \mathcal{V}$ Hilbert spaces. Finally, note that $\|\phi(t)\|^2_{\mathcal{H}} = \langle \phi(t), \phi(t) \rangle = k(t, t)$. $\square$

**Remark 1.** *Note that the result of Theorem 2 applies more generally to algorithms that use dual-weighted-norm pairs $(\|\cdot\|_M, \|\cdot\|_{M^{-1}})$, since this amounts to transforming the decision space $W \mapsto M^{\frac{1}{2}}W$, the losses $G_t \mapsto M^{-\frac{1}{2}}G_t$, and preserving the original inner product structure. We expect the theorem should also generalize to arbitrary dual-norm pairs on $\mathcal{W}$, but this will require some additional care to interpret the norm $\|\cdot\|_{\mathcal{W}^* \otimes \mathcal{H}}$.*

## C.2 A Concrete Example of Proposition 1

---

**Algorithm 2:** Kernelized Instance of Jacobsen and Cutkosky [23, Algorithm 4]

---

**Input:** Lipschitz bound $G \ge \|g_t\|_{\mathcal{W}}$ for all $t$, Value $\epsilon > 0$

**Initialize:** $w_1 = \mathbf{0}$, $G_0 = G \max_t \sqrt{k(t, t)}$, $V_1 = 4G_0^2$, $S_1 = 0$

**Define:** $\Psi(S, V) = \begin{cases} \frac{\epsilon G_0}{\sqrt{V} \log^2(V/G_0^2)} \left[\exp\left(\frac{S^2}{36V}\right) - 1\right] & \text{if } S \le \frac{6V}{G_0} \\ \frac{\epsilon G_0}{\sqrt{V} \log^2(V/G_0^2)} \left[\exp\left(\frac{S}{3G_0} - \frac{6V}{G_0}\right) - 1\right] & \text{otherwise} \end{cases}$

**for** $t = 1 : T$ **do**

    Play $w_t$, receive subgradient $g_t$

    Set $V_{t+1} = V_t + \|g_t\|^2_{\mathcal{W}} k(t, t)$

    Set $S^2_{t+1} = S^2_t + k(t, t)\|g_t\|^2_{\mathcal{W}} + 2\sum_{s=1}^{t-1} k(s, t)\langle g_t, g_s \rangle_{\mathcal{W}}$

    Update $w_{t+1} = \frac{-\sum_{s=1}^{t} k(s, t+1)g_s}{S_{t+1}} \Psi(S_{t+1}, V_{t+1})$

**end**

---

There are many examples of algorithms which would produce the static regret guarantee stated in Proposition 1. In this section, we briefly provide an example which attains the result of the stated form, and provide the full regret guarantee and update in our framework.

Let us consider the algorithm characterized by Jacobsen and Cutkosky [23, Theorem 1] in an unconstrained setting. Their algorithm can be understood as a particular instance of FTRL, and so we can develop its kernelized version using the same reasoning as Example 1. Indeed, applying their algorithm in the space $L(\mathcal{H}, \mathcal{W})$ with inner product $\langle \cdot, \cdot \rangle_{\mathrm{HS}}$ against losses $G_t = g_t \otimes \phi(t)$ leads to updates of the form

$$W_{t+1} = \frac{-\sum_{s=1}^t G_s}{\left\| \sum_{s=1}^t G_s \right\|_{\mathrm{HS}}} \Psi \left( \left\| \sum_{s=1}^t G_s \right\|_{\mathrm{HS}}, V_{t+1} \right),$$

where $V_{t+1} = 4G_0^2 + \sum_{s=1}^t \|G_s\|_{\mathrm{HS}}^2$ and $\Psi(S, V)$ defined in Algorithm 2. Moreover, we have $V_{t+1} = 4G_0^2 + \sum_{s=1}^t \|g_s\|_{\mathcal{W}}^2 k(s,s)$ and $\left\| \sum_{s=1}^t G_s \right\|_{\mathrm{HS}} = \sum_{i,j}^t \langle g_i, g_j \rangle k(i,j)$ via Lemma 10 and Lemma 9 respectively, and so in the context of our reduction Algorithm 1 the updates are

$$w_{t+1} = W_{t+1} \phi(t+1)$$

$$= \frac{-\sum_{s=1}^t G_s \phi(t+1)}{\left\| \sum_{s=1}^t G_s \right\|_{\mathrm{HS}}} \Psi \left( \left\| \sum_{s=1}^t G_s \right\|_{\mathrm{HS}}, V_{t+1} \right)$$

$$= \frac{-\sum_{s=1}^t k(s,t) g_s}{\sqrt{\sum_{i,j=1}^t k(i,j) \langle g_i, g_j \rangle_{\mathcal{W}}}} \Psi \left( \sqrt{\sum_{i,j=1}^t k(i,j) \langle g_i, g_j \rangle_{\mathcal{W}}}, 4G_0^2 + \sum_{s=1}^t k(s,s) \|g_s\|_{\mathcal{W}}^2 \right),$$

leading to the procedure described in Algorithm 2. Notice that, as mentioned in Section 4, this naive implementation requires $\mathcal{O}(t)$ time and memory to update on round $t$ due to having to re-weight the sum $\sum_{s=1}^t k(s,t) g_s$ and compute $\sum_{s=1}^{t-1} k(s,t) \langle g_t, g_s \rangle_{\mathcal{W}}$, so in practice one would ideally implement additional measures to reduce the complexity, such as implementing Nystrom projections or choosing a suitably sparse kernel.

Now applying Algorithm 2 with Theorem 2, we immediately get the following regret guarantee from Jacobsen and Cutkosky [23, Theorem 1].

**Proposition 6.** *Let $\ell_1, \ldots, \ell_T$ be G-Lipschitz convex loss functions and let $g_t \in \partial \ell_t(w_t)$ for all $t$. For any $u_1, \ldots, u_T$ in $\mathcal{W}$ and $U \in L(\mathcal{H}, \mathcal{W})$ satisfying $u_t = U\phi(t)$ for all $t$, Algorithm 2 guarantees*

$$R_T(u_1, \ldots, u_T) = \widetilde{R}_T(U)$$

$$\leq 4G_0 \epsilon + 6 \|U\|_{\mathrm{HS}} \max \left\{ \sqrt{V_{T+1} \ln \left( \frac{\|U\|_{\mathrm{HS}}}{\alpha_{T+1}} + 1 \right)}, G_0 \ln \left( \frac{\|U\|_{\mathrm{HS}}}{\alpha_{T+1}} + 1 \right) \right\},$$

*where $V_{T+1} = 4G_0^2 + \sum_{t=1}^T \|g_t\|_{\mathcal{W}}^2 k(t,t)$ and $\alpha_{T+1} = \frac{\epsilon G_0}{\sqrt{V_{T+1}} \log^2(V_{T+1}/G_0^2)}$.*

### C.3 Scale-free Guarantees

Now that we have seen how to obtain the optimal path-length dependencies on Lipschitz losses, we can extend these guarantees to be *scale-free* by simply changing the base algorithm. In particular, there are algorithms which are adaptive to both the comparator norm and the effective Lipschitz constant, $L_T = \max_{t \in [T]} \|\nabla \ell_t(w_t)\|_{\mathcal{W}}$. Algorithms which scale with $L_T$ rather than a given upper bound $G \geq L_T$ are referred to as *scale-free*. We first consider the setting in which the domain is constrained $\mathcal{W} = \{ w \in \mathbb{R}^d : \|w\|_{\mathcal{W}} \leq D \}$.[4]

**Proposition 7.** *There exists an algorithm which guarantees that for any sequence $u_1, \ldots, u_T$ in $\mathcal{W} = \{ w \in \mathbb{R}^d : \|w\|_{\mathcal{W}} \leq D \}$,*

$$R_T(u_1, \ldots, u_T) = \widetilde{O} \left( L_T(\max_t \|u_t\|_{\mathcal{W}} + D) + \|U\|_{\mathrm{HS}} \sqrt{L_T^2 + \sum_{t=1}^T \|g_t\|_{\mathcal{W}}^2 k(t,t)} \right),$$

*where $L_t = \max_t \|g_t\|_{\mathcal{W}}$.*

---

[4]More generally, this assumption amounts to assuming prior knowledge on a bound $D \geq \|u_t\|_{\mathcal{W}}$ for all $t$, which the learner can leverage by projecting to the same set, regardless of any boundedness of the underlying problem's domain.

The result follows by constraining $\|w_t\|_{\mathcal{W}} \leq D$ and applying the gradient-clipping argument of Cutkosky [12].[5] Indeed, if we've constrained our iterates to satisfy $\|w_t\|_{\mathcal{W}} \leq D$, then we can replace the gradients $g_t$ the the clipped gradients $\widehat{g}_t = g_t \min\left\{1, \frac{\max_{s<t}\|g_s\|_{\mathcal{W}}}{\|g_t\|_{\mathcal{W}}}\right\}$ to get

$$R_T(u_1,\ldots,u_T) = \sum_{t=1}^{T} \langle \widehat{g}_t, w_t - u_t \rangle_{\mathcal{W}} + \sum_{t=1}^{T} \langle g_t - \widehat{g}_t, w_t - u_t \rangle_{\mathcal{W}}$$

$$\leq \widehat{R}_T(u_1,\ldots,u_T) + \max_t \|g_t\|_{\mathcal{W}} \left( D + \max_t \|u_t\|_{\mathcal{W}} \right)$$

following the same telescoping argument as Cutkosky [12]. With this in hand, we can simply apply our reduction Algorithm 1 with the losses $\widehat{G}_t = \widehat{g}_t \otimes \phi(t)$, which we now have an *a priori* bound on at the start of round $t$: $\left\|\widehat{G}_t\right\|_{\mathrm{HS}} \leq \max_{s<t}\|g_s\|_{\mathcal{W}}\sqrt{k(t,t)} := \widehat{L}_t \leq L_t$. Hence, even without prior knowledge of a $G \geq \|g_t\|_{\mathcal{W}}$ we can obtain $\widehat{R}_T(u_1,\ldots,u_T) \leq \widetilde{O}\left(\|U\|_{\mathrm{HS}}\sqrt{\max_t\|g_t\|_{\mathcal{W}}^2 k(t,t) + \sum_{t=1}^{T}\|g_t\|_{\mathcal{W}}^2 k(t,t)}\right)$.

To see why this is difficult using existing techniques, note that nearly all existing algorithms which achieve the optimal $\sqrt{P_T}$ dependence do so by designing an algorithm which guarantees dynamic regret of the form

$$\mathrm{R}_T(u_1,\ldots,u_T) = \widetilde{O}\left(\frac{P_T}{\eta} + \eta G^2 T\right),$$

from which $G\sqrt{P_T T}$ regret is obtained by tuning $\eta$.[6] The tuning step is done by running several instances of the base algorithm in parallel for each $\eta$ in some set $\mathcal{S} = \left\{2^i/G\sqrt{T} \wedge 1/G : i = 0, 1, \ldots\right\}$, and combining the outputs—typically using a mixture-of-experts algorithm like Hedge. Note however that the set $\mathcal{S}$ requires prior knowledge of the Lipschitz constant $G$. There is no straightforward way to adapt to this argument without resorting to unsatisfying doubling strategies, which are well-known to perform poorly in practice. Instead, using our framework we avoid these issues entirely by simply applying a scale-free static regret guarantee to get the a $L_T \|U\|_{\mathrm{HS}} \sqrt{T}$ dependence, and then designing a kernel which ensures $\|U\|_{\mathrm{HS}} \leq \sqrt{M P_T}$.

More generally, when the domain $\mathcal{W}$ is not uniformly bounded, it is still possible to achieve a scale-free bound at the expense of an $L_T \max_t \|u_t\|_{\mathcal{W}}^3$ penalty, again using the same argument as [12]. One simply starts by replacing the comparator sequence $u_1, \ldots, u_T$ with a new one satisfying $\widehat{u}_t = \Pi_{\mathcal{W}_t} u_t$, where $\mathcal{W}_t = \left\{w \in \mathcal{W} : \|w\|_{\mathcal{W}} \leq \sqrt{\sum_{s=1}^{t-1}\|g_s\|_{\mathcal{W}}}\right\}$. Then one can show that

$$R_T(u_1,\ldots,u_T) = \sum_{t=1}^{T} \langle g_t, w_t - u_t \rangle_{\mathcal{W}} = \sum_{t=1}^{T} \langle g_t, w_t - \widehat{u}_t \rangle_{\mathcal{W}} + \sum_{t=1}^{T} \langle g_t, \widehat{u}_t - u_t \rangle_{\mathcal{W}}$$

$$\leq O\left(R_T(\widehat{u}_1,\ldots,\widehat{u}_T) + G\max_t\|u_t\|_{\mathcal{W}}^3\right).$$

Applying the same clipping argument as above and observing that $\widehat{P}_T = \sum_t \|\widehat{u}_t - \widehat{u}_{t-1}\|_{\mathcal{W}} \leq O(P_T + \max_t \|u_t\|_{\mathcal{W}})$ we get the following result.

**Proposition 8.** *There exists an algorithm such that for any $u_1,\ldots,u_T$ in $\mathcal{W} \subseteq \mathbb{R}^d$,*

$$R_T(u_1,\ldots,u_T) = \widetilde{O}\left(L_T \max_t\|u_t\|_{\mathcal{W}}^3 + \|U\|_{\mathrm{HS}}\sqrt{\sum_{t=1}^{T}\|g_t\|_{\mathcal{W}}^2 k(t,t)}\right).$$

*where $L_T = \max_t \|g_t\|_{\mathcal{W}}\sqrt{k(t,t)}$.*

---

[5]Note that constraining the final outputs $w_t$ is straight-forward in our framework; one can simply apply a standard unconstrained-to-constrained reduction in $\mathcal{W}$ [14, 16] prior to applying our dynamic-to-static reduction.

[6]The exception being Zhang et al. [55], which uses a similar high-dimensional embedding as [26], but neither works obtain the optimal $\sqrt{P_T}$ while being scale-free.

# D   Intuitions on the Spectral Density $Q(\omega)$

In this section we provide some additional high-level intuitions that motivate the choice of $Q(\omega)$ in Proposition 2.

Recall that we would like to choose $Q(\omega) \approx 1/|\omega|$, since this choice yields a continuous-time analogue of the $MP_T$ dependence that we want to achieve: $\|u\|_{\mathcal{H}}^2 \leq \max_t |u(t)| \, \|\nabla u\|_{L^1}$. The key issue is that this choice of $Q(\omega)$ is not integrable, diverging as $\omega \to 0$ and $\omega \to \infty$. We fix these issues by adding a small amount of additional regularization, setting

$$Q(\omega) \propto \frac{R_0(\omega)R_\infty(\omega)}{|\omega|} \, ,$$

where $R_\infty(\omega) \approx 1/(1 + |\omega|^2)^{1/4}$ is a *tapering function* that ensures integrability in the asymptotic regime, and $R_0(\omega) \approx 1/\log(1 + 1/\sqrt{|\omega|}) \log^2(\log(1 + 1/\sqrt{|\omega|}))$ ensures that $Q$ is well-behaved near zero. It is clear that $R_\infty(\omega)$ ensures integrability in the asymptotic regime since when $\omega$ is large we have $Q(\omega) \approx R_\infty(\omega)/|\omega| \leq 1/|\omega|^2$, which is integrable away from zero. On the other hand, near zero $R_\infty(\omega) \approx 1$ and we have

$$Q(\omega) \approx \frac{R_0(\omega)}{|\omega|} \approx \frac{1}{|\omega| \log(|\omega|^{-1/2}) \log^2 \log(|\omega|^{-1/2})} \, ,$$

which after a change of variables $t = \log(\omega^{-1/2})$ integrates near zero as

$$\int_{-\epsilon}^{\epsilon} Q(\omega)d\omega = 2\int_0^{\epsilon} Q(\omega)d\omega \approx \int_{\log(1/\epsilon)}^{\infty} \frac{1}{t \log^2(t)} dt = \mathcal{O}(1)$$

for an appropriately chosen $\epsilon$.

# E   Proofs for Section 4.1 (Controlling the Trade-offs Induced by $\mathcal{H}$)

## E.1   Proof of Theorem 3

**Theorem 3.** *Let $Q : \mathbb{R} \to \mathbb{R}_+$ be an integrable strictly positive even function on $\mathbb{R} \setminus \{0\}$ and such that $R(x) := 2\pi/(x(1 + (x/2\pi)^{2m})Q(x))$ is also integrable for some $m \in \mathbb{N}$, $m \geq 1$. Let $k$ be defined in terms of $Q$ as in Eq. (3). Then $k$ is a translation invariant universal kernel with $k(t,t) \leq \|Q\|_{L^1}$ for all $t \in \mathbb{R}$. The RKHS $\mathcal{H}$ associated to $k$ contains the space of finitely supported functions with bounded derivatives up to order $2m$, and moreover, for any $T > 0$ and any $2m$-times differentiable function $f$ that is supported on $[0, T+1]$,*

$$\|f\|_{\mathcal{H}}^2 \leq c(T) \, \|\nabla f\|_{L^1} \, \|f - \nabla^{2m} f\|_{L^\infty},$$

*where $c(T) := \|F[R]\|_{L^1([-T-1,T+1])}$. If $R$ is monotonically decreasing on $(0,\infty)$, then,*

$$c(T) \leq \inf_{\alpha > 0} 2\pi(T+1)^2 \int_0^\alpha R(x)x dx + \frac{2}{\pi} \int_\alpha^\infty \frac{R(x)}{x} dx \, , \qquad \forall T > 0$$

*Proof.* Consider a function $f$ that is supported on $[0,\tau]$ for some $\tau > 0$. To bound the norm above in terms of the $L^1$ norm we use the fact that for any $u, v, w \in L^2(\mathbb{R}) \oplus L^1(\mathbb{R})$ we have $F[\nabla^k u] = (2\pi i)^k \hat{u}$, for $k \in \mathbb{N}$, $F[u \star v] = \hat{u}\hat{v}$, where $\star$ is the convolution operator, and that by the Plancherel theorem we have $\int_{\mathbb{R}} \hat{u}(\omega)\hat{v}(\omega)dt = \int_{\mathbb{R}} u(t)v(t)dt$ and so, in particular, $\int_{\mathbb{R}} \hat{u}(\omega)\hat{v}(\omega)\hat{w}(\omega)dt = \int_{\mathbb{R}} u(t)(v \star w)(t)dt$. Moreover, note that, by construction $R$ is an integrable real odd function, so its Fourier transform $\hat{R}$ is an odd purely imaginary function. So, we have

$$\begin{aligned}
\|f\|_{\mathcal{H}}^2 &:= \int_{\mathbb{R}} \frac{|\hat{f}(\omega)|^2}{Q(\omega)} = i\int_{\mathbb{R}} \frac{\overline{\omega \hat{f}(\omega)}}{2\pi i} \left(1 + \frac{\omega^{2m}}{(2\pi)^{2m}}\right) \hat{f}(\omega) \, R(\omega)d\omega \\
&= i\int_{\mathbb{R}} \overline{\nabla f(-t)} \, ((f - \nabla^{2m} f) \star \hat{R})(t)dt \\
&= i\int_{-\tau}^0 \overline{\nabla f(-t)} \, ((f - \nabla^{2m} f) \star \hat{R})(t)dt \\
&\leq \|\nabla f\|_{L^1} \|(f - \nabla^{2m} f) \star \hat{R}\|_{L^\infty([-\tau,0])},
\end{aligned}$$

where in the last two steps we used the fact that $f$ is bounded in $[0, \tau]$ and the Hölder inequality. Since also $\nabla^m f$ is supported on $[0, \tau]$, we have

$$\|(f - \nabla^{2m} f) \star \widehat{R}\|_{L^\infty([-\tau,0])} = \sup_{t' \in [-\tau, 0]} \int_0^\tau |f(t) - \nabla^{2m} f(t)||\widehat{R}(t' - t)|dt$$

$$\leq \|f - \nabla^{2m} f\|_{L^\infty(\mathbb{R})} \|\widehat{R}\|_{L^1([-\tau,\tau])} .$$

The last step is finding a bound that is easy to compute for $c(\tau) := \|\widehat{R}\|_{L^1([-\tau,\tau])}$. We start noting that since $R$ is odd

$$\widehat{R}(t) := \int_{\mathbb{R}} R(x) e^{2\pi i x t} dx = 2i \int_0^\infty R(x) \sin(2\pi i x t) dx,$$

and, in particular also $\widehat{R}$ is an odd function. The characterization we propose for $c(\tau)$ is a direct consequence of the fact that the sine transform of $R$ is non-negative (also known as Polya criterion, which we recall in Lemma 3 and is applicable since $R$ is positive and decreasing on $(0, \infty)$). Now, by expanding the definition of $\widehat{R}$ and using the non-negativity of the sine transform, we have

$$\int_{-\tau}^\tau |\widehat{R}(t)|dt = 2 \int_0^\tau |\widehat{R}(t)|dt = 2 \int_0^\tau \left| \int_0^\infty R(x) \sin(2\pi x t)dx \right| dt$$

$$= 2 \int_0^\tau \int_0^\infty R(x) \sin(2\pi x t)dxdt = 2 \int_0^\infty R(x) \left( \int_0^\tau \sin(2\pi x t)dt \right) dx$$

$$= 2 \int_0^\infty R(x) \frac{\sin(\tau \pi x)^2}{\pi x} dx.$$

To conclude, let $\alpha > 0$, since $|\sin(z)| \leq \min(|z|, 1)$ for any $z \in \mathbb{R}$

$$\int_0^\infty R(x) \frac{\sin(\tau \pi x)^2}{\pi x} dx = \int_0^\alpha R(x) \frac{\sin^2(\tau \pi x)}{\pi x} dx + \int_\alpha^\infty R(x) \frac{\sin^2(\tau \pi x)}{\pi x} dx$$

$$\leq \pi \tau^2 \int_0^\alpha R(x) x dx + \frac{1}{\pi} \int_\alpha^\infty \frac{R(x)}{x} dx.$$

The stated result then follows by choosing $\tau = T + 1$. $\qquad \square$

### E.2 Proof of Theorem 4

Before proving Theorem 4 we need two auxiliary results

**Lemma 1.** *Given $m \in \mathbb{N}$ with $m \geq 1$ and $T > 0$ there exists a function $b_T$ that is $2m$-times differentiable and that is identically equal to 0 on $\mathbb{R} \setminus (0, T + 1)$ and that is identically equal to 1 on the interval $[1, T]$. Moreover, for any $2m$-times differentiable function $f$, with derivatives in $L^p(S)$ for $p \in [1, \infty]$ and interval $S \subseteq \mathbb{R}$, we have*

$$\|\nabla^k (f b_T)\|_{L^p(S)} \leq \frac{(8(2m + 3/2))^{k+1}}{\pi^{2m+3/2}} \max_{0 \leq h \leq k} \|\nabla^h f\|_{L^p(S \cap [0, T+1])}.$$

*Proof.* Consider the function

$$B(x) = \frac{2\Gamma(3/2 + 2m)}{\sqrt{\pi}\Gamma(1 + 2m)} (1 - 4x^2)_+^{2m},$$

where $\Gamma$ is the gamma function. Then $B$ is supported on $(-1/2, 1/2)$, integrates to 1, and is $2m$-times differentiable everywhere. Its Fourier transform (see [46], Thm. 4.15) is

$$\widehat{B}(\omega) = \Gamma(3/2 + 2m) J_{\frac{1}{2} + 2m}(\pi|\omega|) \left( \frac{2}{\pi|\omega|} \right)^{2m+1/2},$$

where $J_\nu$ is the Bessel J function of order $\nu$ [46]. Since $J_\nu$ is analytic on $[0, \infty)$ for $\nu > 0$ and $J_\nu(z) = \mathcal{O}(|z|^\nu)$ when $|z| \to 0$ and also $J_\nu(z) = \mathcal{O}(z^{-1/2})$ for $z \to \infty$, then $\widehat{B}$ is in $L^1 \cap L^\infty$ and analytic. We build $b_T$ as follows

$$b_T(t) = \int_0^t B(x - 1/2) - B(x - T - 1/2)dx.$$

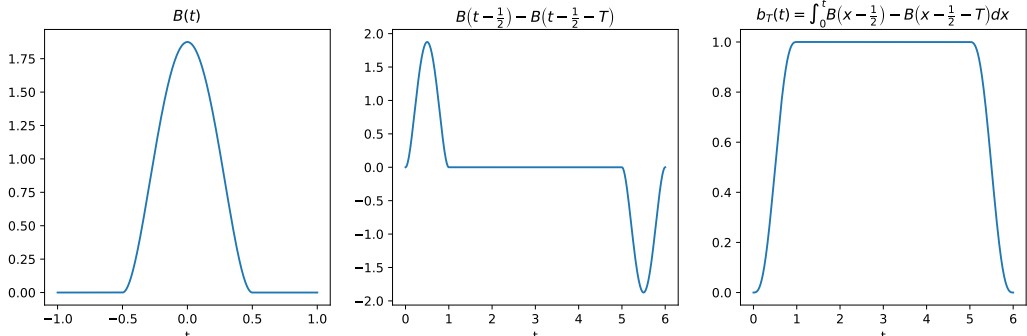

Figure 2: Plots demonstrating the functions used in the construction of $b_T(t)$ for $m = 1$ and $T = 5$. The function $B(x)$ is a simple bump function, designed such that $\int_{\mathbb{R}} B(x)dx = 1$, shown on the left. The center demonstrates how we can combine translations of $B(x)$ to get a function with two bumps which will eventually cancel out when integrated over $[0, T + 1]$, leading to the function $b_T$ shown on the right.

This function by construction is $2m$-times differentiable everywhere, moreover it is identically equal to $0$ on $\mathbb{R} \setminus (0, T+1)$ and identically equal to $1$ on $[0, T]$. To help with intuitions, we show an example of the functions $B(x)$, $B(x - \frac{1}{2}) - B(x - \frac{1}{2} - T)$, and $b_T(t)$ for $m = 1$ and $T = 5$ in Figure 2. The Fourier transform of $b_T$ is

$$\widehat{b}_T(\omega) = \frac{1}{2\pi i \omega} \widehat{B}(\omega) e^{-\pi i \omega} (1 - e^{-2\pi i T \omega}).$$

Now we define $u(t) := f(t)b_T(t)$. By construction, $u$ is equal to $f_T$ on $[1, T]$ since $b_T$ is identically $1$ on this interval, let $Z := S \cap [0, T + 1]$, we have

$$\|\nabla^k u\|_{L^p(S)} = \|\nabla^k u\|_{L^p(Z)} = \left\| \sum_{h=0}^{k} \binom{k}{h} \nabla^h f \nabla^{k-h} b_T \right\|_{L^p(Z)}$$

$$\leq \max_{0 \leq j \leq k} \|\nabla^h f\|_{L^p(Z)} \sum_{h=0}^{l} \binom{k}{h} \|\nabla^j b_T\|_{L^\infty(\mathbb{R})},$$

Now, given the Fourier transform of $\widehat{b}_T$,

$$\|\nabla^h b_T\|_{L^\infty} \leq (2\pi)^h \|\omega^h \widehat{b}_T\|_{L^1} \leq \Gamma(3/2 + 2m) \frac{2^{2m+h-1/2}}{\pi^{2m-h+3/2}} \|\omega^{-2m-3/2+h} J_{2m+1/2}(\pi|\omega|)\|_{L^1}.$$

Using the fact that $|J_\nu(z)| \leq \min(z^\nu 2^{-\nu}/\Gamma(1 + \nu), \nu^{-1/3})$ (see [38], Eq. 10.14.2, 10.14.4) for any $z \geq 0$, for any $\alpha > 0$, we have

$$\int_{\mathbb{R}} \frac{|J_{2m+1/2}(\pi|\omega|)|}{|\omega|^{-2m-3/2+h}} d\omega = 2 \int_0^\alpha \frac{|J_{2m+1/2}(\pi x)|}{x^{2m+3/2-h}} dx + 2 \int_\alpha^\infty \frac{|J_{2m+1/2}(\pi x)|}{x^{2m+3/2-h}} dx$$

$$\leq 2 \int_0^\alpha \frac{x^{2m+1/2} 2^{-2m-1/2}}{x^{2m+3/2-h} \Gamma(2m + 3/2)} dx + 2 \int_\alpha^\infty \frac{(2m + 1/2)^{-1/3}}{x^{2m+3/2-h}} dx$$

$$= \frac{2^{-2m+1/2} \alpha^h}{h \Gamma(2m + 3/2)} + \frac{2\alpha^{-(2m-h+1/2)}}{(2m - h + 1/2)(2m + 1/2)^{1/3}}.$$

Optimizing in $\alpha$, we obtain

$$\|\omega^{-2m-3/2+h} J_{2m+1/2}(\pi|\omega|)\|_{L^1} \leq 2^{-2m+5/2+h} \Gamma(2m + 3/2)^{-\frac{2m+1/2-h}{2m+1/2}},$$

leading to

$$\|\nabla^h b_T\|_{L^\infty} \leq \frac{2^{2h+2}}{\pi^{2m-h+3/2}} \Gamma(2m + 3/2)^{\frac{h}{2m+1/2}} \leq \frac{2(8m + 6)^{h+1/2}}{\pi^{2m-h+3/2}},$$

this leads to

$$\|\nabla^k u\|_{L^p(S)} \le \max_{0 \le j \le k} \|\nabla^h f\|_{L^p(S)} \sum_{h=0}^l \binom{k}{h} \|\nabla^j b_T\|_{L^\infty(\mathbb{R})}$$

$$\le \frac{2(8m+6)^{1/2}}{\pi^{2m+3/2}} \left(1 + \frac{8m+6}{\pi}\right)^k \max_{0 \le j \le k} \|\nabla^h f\|_{L^p(Z)}.$$

To conclude, note that

$$\frac{2(8m+6)^{1/2}}{\pi^{2m+3/2}} \left(1 + \frac{8m+6}{\pi}\right)^k \le \frac{8^{k+1}(2m+3/2)^{k+1}}{\pi^{2m+3/2}}.$$

□

**Lemma 2.** *Let $f$ be such that its Fourier transform $\widehat{f}$ and its weak derivative $\nabla \widehat{f}$ are both $L^2$, then*

$$\|f\|_{L^1(\mathbb{R})} \le \frac{2}{\sqrt{\pi}} \|\widehat{f}\|_{L^2(\mathbb{R})}^{1/2} \|\nabla \widehat{f}\|_{L^2(\mathbb{R})}^{1/2}.$$

*Proof.* For any $t > 0$,

$$\|f\|_{L^1(\mathbb{R})} \le \|f\|_{L^1([-t,t])} + \|f\|_{L^1(\mathbb{R} \setminus [-t,t])}.$$

Now by the Hölder inequality we have

$$\|f\|_{L^1([-t,t])} = \|f \cdot 1\|_{L^1([-t,t])} \le \|f\|_{L^2([-t,t])} \|1\|_{L^2([-t,t])} = \sqrt{2t} \|f\|_{L^2([-t,t])}$$

where $1(x)$ is the constant function 1. Similarly,

$$\|f\|_{L^1(\mathbb{R} \setminus [-t,t])} = \|fx \cdot 1/x\|_{L^1(\mathbb{R} \setminus [-t,t])} \le \|fx\|_{L^2(\mathbb{R} \setminus [-t,t])} \|1/x\|_{L^2(\mathbb{R} \setminus [-t,t])} = \sqrt{2/t} \|fx\|_{L^2(\mathbb{R} \setminus [-t,t])},$$

since $\|1/x\|_{L^2(\mathbb{R} \setminus [-t,t])}^2 = 2 \int_t^\infty 1/x^2 dx = 2/t$. This leads to

$$\|f\|_{L^1(\mathbb{R})} \le \sqrt{2}(\sqrt{t} \|f\|_{L^2} + 1/\sqrt{t} \|fx\|_{L^2}).$$

Optimizing on $t$ we obtain,

$$\|f\|_{L^1} \le 2\sqrt{2} \|f\|_{L^2}^{1/2} \|fx\|_{L^2}^{1/2}.$$

The first case is concluded by applying the Plancherel theorem for which $\|f\|_{L^2} = \|\widehat{f}\|_{L^2}$ and $\|fx\|_{L^2} = \|F[fx]\|_{L^2}$ and by the fact that $F[f(x)x](\omega) = i/(2\pi)\nabla \widehat{f}(\omega)$. □

**Theorem 4.** *Let $v_1, \ldots, v_T \in \mathbb{R}^d$ and let $\mathcal{H}$ be the RKHS associated to kernel $k$ contain finitely supported functions with bounded derivatives up to order $2m$, with $m \in \mathbb{N}$, $m \ge 1$. Then there exists a function $u \in \mathcal{H}$ supported on $[0, T+1]$, such that $u(t) = v_t$ for all $t \in [T]$ and*

$$\|\nabla u\|_{L^1} \le C\|v_1\|_{\mathcal{W}} + C \sum_{t=2}^T \|v_t - v_{t-1}\|_{\mathcal{W}}, \qquad \left\|u - \nabla^{2m} u\right\|_{L^\infty} \le C' \max_t \|v_t\|_{\mathcal{W}}.$$

*with $C, C'$ depending only on $m$ and given in explicitly in the proof.*

*Proof.* We will build $u$ as follows

$$u(t) = f(t)b_T(t),$$

where $b_T$ is a $m+1$-times differentiable function that is supported on $[0, T+1]$ and that is equal to 1 on $[1, T]$, while $f$ is a function that interpolates $v_t$, i.e. $f(t) = v_t$ for $t \in \{1, \ldots, T\}$. We build $f$ as follows. Consider the following function, that is a product of two sinc functions

$$S(x) := \frac{\sin(\pi x)}{\pi x} \frac{2\sin(\pi x/2)}{\pi x},$$

$S$ satisfies $S(0) = 1$ and $S(t) = 0$ on $t \in \mathbb{Z} \setminus \{0\}$. Now we can build $f$ as follows

$$f(t) = \sum_{\ell=0} v_\ell S(t - \ell).$$

By construction, for all $t \in \{1, \ldots, T\}$ the following holds

$$f(t) = \sum_{\ell=0} v_\ell S(t - \ell) = \sum_{\ell=0} v_\ell \delta_{t=\ell} = v_t.$$

Note that, by construction $f$ is a band-limited function with band $[-3/4, 3/4]$ that interpolates the given points.

**Step 1. Bounding $\|\nabla f\|_{L^1}$.** Now we bound pointwise $|\nabla f(x)|$ with $|v(x) - v(x-1)|$. The Fourier transform $\widehat{f}$ of $u$ is equal to

$$\widehat{f}(\omega) = \sum_{\ell=1}^{T} v_\ell e^{2\pi i \ell \omega} \widehat{S}(\omega) = g(\omega)\widehat{S}(\omega),$$

where $g(\omega) = \sum_{\ell=1}^{T} v_\ell e^{2\pi i \ell \omega}$ and where $\widehat{S}$ is Fourier Transform of $S$. Note that $S$ is band-limited, i.e., $\widehat{S}$ is equal to 0 on $\mathbb{R} \setminus [-3/4, 3/4]$. In particular,

$$\widehat{S}(\omega) = |\omega - 3/4| - |\omega - 1/4| - |\omega + 1/4| + |\omega + 3/4|, \quad \forall \omega \in \mathbb{R}.$$

Now passing by the Fourier transform of $\nabla f$, we obtain

$$F[\nabla f](\omega) = g(\omega)\widehat{S}(\omega)\omega = \widehat{L}(\omega)\,\widehat{M}(\omega), \quad \widehat{L}(\omega) := g(\omega)\frac{1 - e^{2\pi i \omega}}{1 + \omega^2}, \quad \widehat{M}(\omega) := \widehat{S}(\omega)\frac{\omega(1 + \omega^2)}{1 - e^{2\pi i \omega}}.$$

Note that $\widehat{M}(\omega)$ is bounded, continuous and supported in $[-3/4, 3/4]$, since $\widehat{S}$ is supported in $[-3/4, 3/4]$ and $\frac{\omega(1+\omega^2)}{1-e^{2\pi i \omega}}$ is bounded and analytic on such interval. So we have

$$\nabla f(x) = F^{-1}[F[\nabla f]](x) = F^{-1}\left[\widehat{L}(\omega)\widehat{M}(\omega)\right] = (L \star M)(x),$$

where $\star$ is the convolution operator and $M = F^{-1}[\widehat{M}]$, $L = F^{-1}[\widehat{L}]$. Now note that

$$\widehat{L}(\omega) := g(\omega)\frac{1 - e^{2\pi i \omega}}{1 + \omega^2} = \sum_{\ell=1}^{T} v_\ell \frac{e^{2\pi i \ell \omega}}{1 + \omega^2} - \sum_{\ell=1}^{T} v_\ell \frac{e^{2\pi i(\ell+1)\omega}}{1 + \omega^2}$$

$$= \sum_{\ell=2}^{T}(v_\ell - v_{\ell-1})\frac{e^{2\pi i \ell \omega}}{1 + \omega^2} + v_1 \frac{e^{2\pi i \omega}}{1 + \omega^2}$$

Since the inverse Fourier transform of $e^{2\pi a \omega}/(1 + \omega^2)$ is $e^{-|x-a|}$ for any $a \in \mathbb{R}$, we have

$$L(x) := F^{-1}\left[g(\omega)\frac{1 - e^{2\pi i \omega}}{1 + \omega^2}\right](x) = \pi \sum_{\ell=2}^{T}(v_\ell - v_{\ell-1})e^{-2\pi|x-\ell|} + v_1 \pi e^{-2\pi|x-1|}.$$

By Young's inequality for the convolution, we have that $\|f \star g\|_{L^1} \leq \|f\|_{L^1}\|g\|_{L^1}$ for any integrable functions $f, g$, so in our case

$$\|\nabla f\|_{L^1} = \|L \star M\|_{L^1} \leq \|L\|_{L^1}\|M\|_{L^1}$$

$$\leq \pi \|M\|_{L^1} \sum_{\ell=2}^{T} \|v_\ell - v_{\ell-1}\|\|e^{-2\pi|x-\ell|}\|_{L^1} + \pi\|M\|_{L^1}\|v_1\|\|e^{-2\pi|x-1|}\|_{L^1}.$$

To conclude we have $\|e^{-2\pi|x-\ell|}\|_{L^1} = \|e^{-2\pi|x|}\|_{L^1} = 1/\pi$ and we need to bound the $L^1$ norm of $M$. Note that $\widehat{M}(\omega)$ admits a weak derivative, since it is the product of a bounded analytic function on the support and $\widehat{S}$ that admits a weak derivative that is the following

$$\nabla \widehat{S}(\omega) = \text{sign}(\omega - 3/4) - \text{sign}(\omega - 1/4) - \text{sign}(\omega + 1/4) + \text{sign}(\omega + 3/4).$$

In particular, we have for every $\omega \in \mathbb{R}$

$$|\widehat{M}(\omega)| = \left|\widehat{S}(\omega)\frac{\omega(1 + \omega^2)}{1 - e^{2\pi i \omega}}\right| \leq 1_{[-3/4, 3/4]}(\omega)$$

$$\left|\nabla \widehat{M}(\omega)\right| = \left|\nabla \widehat{S}(\omega)\frac{\omega(1+\omega^2)}{1-e^{2\pi i\omega}} + \widehat{S}(\omega)\frac{\partial}{\partial\omega}\frac{\omega(1+\omega^2)}{1-e^{2\pi i\omega}}\right| \le 2 \times 1_{[-3/4,3/4]}(\omega)$$

Using Lemma 2 we can bound $\|M\|_{L^1}$ as follows,

$$\|M\|_{L^1} \le \frac{2}{\sqrt{\pi}}\|\widehat{M}\|_{L^2}^{1/2}\|\partial\widehat{M}\|_{L^2}^{1/2} \le 2,$$

obtaining

$$\|\nabla f\|_{L^1(\mathbb{R})} \le 2\|v_1\|_{\mathcal{W}} + 2\sum_{t=2}^{T}\|v_t - v_{t-1}\|_{\mathcal{W}}. \tag{5}$$

**Step 2. Bounding $\|\nabla^k f\|_{L^\infty}$.** Let $k \in \mathbb{N}$ and $k \ge 1$. Since $\widehat{Z}(\omega) := \widehat{S}(\omega/4)$ is equal to 1 on $[-1,1]$ and it is supported on $[-3,3]$, we have that $\widehat{f}(\omega) = \widehat{Z}(\omega)\widehat{f}(\omega)$. So, by using the properties of the Fourier transform of convolutions, we have that for all $t \in \mathbb{R}$,

$$f(t) = F^{-1}[\widehat{f}](t) = F^{-1}[\widehat{Z}\widehat{f}](t) = (F^{-1}[\widehat{Z}] \star F^{-1}[f]) = (Z \star f)(t),$$

where $Z(t) = 4S(4t)$ for all $t \in \mathbb{R}$ is the inverse Fourier transform of $\widehat{Z}$. So we have

$$\|\nabla^k f\|_{L^\infty} = \|\nabla^k(Z \star f)\|_{L^\infty} = \|(\nabla^k Z) \star f\|_{L^\infty} \le \|\nabla^k Z\|_{L^1}\|f\|_{L^\infty}.$$

Now, we have

$$\|f\|_{L^\infty} = \sup_{t\in\mathbb{R}}\left|\sum_{\ell=1}^{T} v_\ell S(t-\ell)\right| \le \left(\max_{\ell\in[T]}\|v_\ell\|_{\mathcal{W}}\right)\sup_{t\in\mathbb{R}}\sum_{\ell=1}^{T}|S(t-\ell)|. \tag{6}$$

Since $|S(t)| \le 1/(1+2t^2)$ for any $t \in \mathbb{R}$, we have

$$\sup_{t\in\mathbb{R}}\sum_{\ell=1}^{T}|S(t-\ell)| \le \sup_{t\in\mathbb{R}}\sum_{\ell\in\mathbb{Z}}\frac{1}{1+2(t-\ell)^2} = \sum_{\ell\in\mathbb{Z}}\frac{1}{1+2\ell^2} \le 4.$$

To conclude this section, using Lemma 2 and since $F[\nabla^k Z] = \omega^k Z/(2\pi i)^k$

$$\|\nabla^k Z\|_{L^1} \le \frac{2\sqrt{2}}{(2\pi)^k}\|\omega^k\widehat{Z}\|_{L^2}^{1/2}\|\nabla(\omega^k\widehat{Z})\|_{L^2}^{1/2} \le \frac{3^{2+k}+4k}{(2\pi)^k(1+k^{5/2})} \le 3.$$

So, for any $k \in \mathbb{N}$ (including 0, since we have Eq. (6))

$$\|\nabla^k f\|_{L^\infty} \le \|\nabla^k Z\|_{L^1}\|f\|_{L^\infty} \le 12\max_{\ell}\|v_\ell\|_{\mathcal{W}} \tag{7}$$

**Step 3. Building $b_T$ and computing the final norms.** Lemma 1 constructs a function $b_T$ that is $2m$-times differentiable and that is identically equal to 0 on $\mathbb{R} \setminus (0, T+1)$ and that is identically equal to 1 on $[1,T]$, moreover it proves that for any $2m$-times differentiable function that has the derivatives $L^p(S)$ integrable for some $p \in [0,\infty]$ and some interval $S \subseteq \mathbb{R}$, we have

$$\|\nabla^k(fb_T)\|_{L^p(S)} \le C_{k,m}\max_{0\le h\le k}\|\nabla^h f\|_{L^p(S\cap[0,T+1])}.$$

Define $u(t) := f(t)b_T(t)$. By construction $u$ is equal to $f_T$ on $[0,T]$ since $b_T$ is identically 1 on this interval, and so in particular

$$u(\ell) = f(\ell)b_T(\ell) = f(\ell) = v_\ell, \quad \forall \ell \in \{1,\dots,T\},$$

moreover $u(t) = f(t)b_T(t) = 0$ for $t \in \mathbb{R} \setminus (0, T+1)$. By applying the result above, together with Eq. (7)

$$\|u - \nabla^{2m}u\|_{L^\infty} \le \|u\|_{L^\infty} + \|\nabla^{2m}u\|_{L^\infty} \le 12(1+C_{2m,m})\max_{\ell}\|v_\ell\|_{\mathcal{W}}.$$

Applying the same lemma, with Eq. (5) we have

$$\|\nabla u\|_{L^1(\mathbb{R})} \le 2C_{1,m}\|v_1\|_{\mathcal{W}} + 2C_{1,m}\sum_{t=2}^{T}\|v_t - v_{t-1}\|_{\mathcal{W}}.$$

This completes the proof. $\qquad\square$

### E.3 Proof of Lemma 3

Here, we recall a classical result about the positivity of the sine transform of a positive decreasing function (see e.g. [48] Eq. 4).

**Lemma 3.** *Let $R$ be an integrable, positive and strictly decreasing on $(0, \infty)$. Then, for any $t > 0$ we have*

$$\int_0^\infty R(x) \sin(2\pi x t) dx \geq 0.$$

*Proof.* Since $\sin(2\pi(z + 1/2)) = -\sin(2\pi z)$ for each $z \in [j, j + 1/2]$ and $j \in \mathbb{N}$, we have

$$\int_0^\infty R(x) \sin(2\pi x t) dx = \sum_{j=0}^\infty \int_{\frac{j}{t}}^{\frac{j+1}{t}} R(x) \sin(2\pi x t) dx$$

$$= \frac{1}{t} \sum_{j=0}^\infty \int_0^1 R\left(\frac{j + \theta}{t}\right) \sin(2\pi\theta) d\theta$$

$$= \frac{1}{t} \sum_{j=0}^\infty \int_0^{1/2} \left[R(\frac{j + \theta}{t}) - R(\frac{j + 1/2 + \theta}{t})\right] \sin(2\pi\theta) d\theta$$

$$\geq 0,$$

where the last step is due to the fact that $R$ is decreasing, so $R((j+\theta)/t) - R((j+\theta)/t + 1/2t) \geq 0$ for any $j \in \mathbb{N}, \theta \in [0, 1/2]$ and that $\sin(2\pi\theta) \geq 0$ on the integration interval $\theta \in [0, 1/2]$. □

### E.4 Proof of Proposition 2

**Proposition 2.** *Let $Q : \mathbb{R} \to \mathbb{R}_+$ be defined as*

$$Q(\omega) = \frac{1/4 \, \log\log\pi}{|\omega| \, (1 + |\omega|^2/4\pi^2)^{\frac{1}{4}} \, \log(\pi + |\omega|^{-\frac{1}{2}}) \, \log^2\log(\pi + |\omega|^{-\frac{1}{2}})}.$$

*Then we can apply Theorem 3 with $m = 1$: the function $k$ defined in terms of $Q$ as in Eq. (3) is a translation invariant kernel with $k(t, t) \leq 8\pi^2$, $\forall t \in \mathbb{R}$; the associated RKHS norm satisfies*

$$\|f\|_{\mathcal{H}}^2 \leq c^2 \|\nabla f\|_{L^1} \|f - \nabla^2 f\|_{L^\infty} (\ln(1 + T) \ln\ln(1 + T))^2,$$

*for any $f$ that is 2-times differentiable and supported in $[0, T + 1]$, where $T > 2$ and $c \leq (2\pi e)^2$.*

*Proof.* **Step 1. Characterization of $Q$.** The function $Q$ is even, strictly positive and analytic on $\mathbb{R} \setminus \{0\}$. To study its integrability define the auxiliary function $S : [0, \infty) \to [0, \infty)$ as

$$S(z) = \frac{\log\log\pi}{2 \log\log(\pi + 1/z^s)}.$$

$S$ is concave, strictly increasing, on $(0, \infty)$ and with $S(0) = 0$ and $\lim_{z \to \infty} S(z) = 1/2$. So its derivative $S'$ corresponding to

$$S'(z) = \frac{s/2 \, \log\log\pi}{z \, (1 + \pi z^s) \, \log(\pi + z^{-s}) \, \log^2\log(\pi + z^{-s})},$$

is positive and strictly decreasing on $(0, \infty)$ and since $0 < s < 2m$, the function $(\cdot)^{s/2m}$ is concave, and we have $(1 + (|\omega|/2\pi)^{2m})^{s/2m} \geq 2^{\frac{s}{2m} - 1}(1 + (|\omega|/2\pi)^s)$, so

$$L := \sup_{\omega \in \mathbb{R}} \frac{Q(\omega)}{S'(|\omega|)} = \sup_{\omega \in \mathbb{R}} \frac{1 + \pi|\omega|^s}{(1 + (|\omega|/2\pi)^{2m})^{s/2m}} \leq 2^{1 - \frac{s}{2m}} \sup_{\omega \in \mathbb{R}} \frac{1 + \pi|\omega|^s}{1 + (|\omega|/2\pi)^s} = 2^{\frac{2m(1+s)-s}{2m}} \pi^{1+s}. \tag{8}$$

So $0 < Q(\omega) \leq LS'(|\omega|)$ for any $\omega \in \mathbb{R}$, and since $S'$ is positive and integrable, then $Q$ is integrable too and we have

$$\int_{\mathbb{R}} Q(\omega) d\omega \leq 2L \int_0^\infty S'(z) dz = 2L(\lim_{z \to \infty} S(z) - S(0)) = L.$$

To conclude this first step, since $Q$ is integrable it admits a Fourier transform $\widehat{Q}$, and since it is also positive, $k(t, t') := \widehat{Q}(t - t')$ for $t, t' \in \mathbb{R}$ is a translation invariant kernel. In particular, $k(t, t) = \widehat{Q}(0) = \int_{\mathbb{R}} Q(\omega) d\omega \leq L$, for any $t \in \mathbb{R}$.

**Step 2. Characterization of $R$ and explicit bound for $\|F[R]\|_{L^1[-T,T]}$.** The second condition to apply Theorem 3, concerns the function $R$ defined as

$$R(x) := \frac{2\pi}{x(1 + (x/2\pi)^{2m})Q(x)}.$$

Note that $R$ is an odd function, since $x$ is odd, while both $1 + (\frac{x}{2\pi})^{2m}$ and $Q(x)$ are even. Moreover, $R$ is analytic on $\mathbb{R} \setminus \{0\}$ since $Q(x)$ is analytic on the same domain and $x(1 + (x/2\pi)^{2m})$ is analytic on the whole axis. Expanding the definition of $Q$ in $R$, we obtain

$$R(\omega) = \frac{C_0 \log(\pi + |\omega|^{-s}) \log^2(\log(\pi + |\omega|^{-s}))}{(1 + (\omega/2\pi)^{2m})^{\frac{2m-s}{2m}}},$$

where $C_0 = 4\pi/(s \log \log \pi)$. From which we observe that $R$ is also positive and strictly decreasing on $(0, \infty)$ since, $\log(\pi + |x|^{-s})$, $\log(\log(\pi + |x|^{-s}))^2$ and $1/(1 + (\omega/2\pi)^{2m})^{\frac{2m-s}{2m}})$ are strictly positive and strictly decreasing on $(0, \infty)$. So we can apply the bound on $\|F[R]\|_{L^1([-T,T])}$ in Theorem 3, obtaining, for any $\alpha > 0$

$$c(T) := \|F[R]\|_{L^1([-T,T])} \leq 2\pi T^2 \int_0^\alpha R(x) x dx + \frac{2}{\pi} \int_\alpha^\infty \frac{R(x)}{x} dx.$$

To bound such integrals, we first simplify $R$. Let $\beta, \gamma > 0$, since the following functions are bounded, non-negative, with a unique critical point that is a maximum, by equating their derivative to zero we obtain

$$\sup_{z > \pi} \frac{\log^2(\log(z))}{\log^{1+\gamma}(z)} = (2/\gamma)^2 e^{-2}, \qquad \sup_{z > \pi} \frac{\log^{1+\gamma}(z)}{z^\beta} = (\frac{1+\gamma}{\beta})^{1+\gamma} e^{-1-\gamma},$$

so we have for any $x > 0$,

$$\begin{aligned}
R(x) &= \frac{C_0 \log(\pi + x^{-s}) \log(\log(\pi + x^{-s}))^2}{(1 + (x/2\pi)^{2m})^{\frac{2m-s}{2m}}} \\
&= \frac{\log^2(\log(\pi + x^{-s}))}{\log^\gamma(\pi + x^{-s})} \frac{\log^{1+\gamma}(\pi + x^{-s})}{(\pi + x^{-s})^\beta} \frac{C_0(\pi + x^{-s})^\beta}{(1 + (x/2\pi)^{2m})^{\frac{2m-s}{2m}}} \\
&\leq C_1(\beta, \gamma) \frac{(\pi + x^{-s})^\beta}{(1 + (x/2\pi)^{2m})^{\frac{2m-s}{2m}}}.
\end{aligned}$$

where $C_1(\beta, \gamma) = (2/\gamma)^2 (\frac{1+\gamma}{\beta})^{1+\gamma} e^{-3-\gamma} C_0 \leq 16 C_0/(e^3 \gamma^2 \beta^{1+\gamma}) \leq \gamma^{-2} \beta^{-1-\gamma} C_0$. Now we can control the integral of interest by using the bound above. First, we will split it in two regions of interest. For the first term, letting $\beta < 1$,

$$\begin{aligned}
T^2 \pi \int_0^\alpha R(x) x dx &\leq T^2 \pi C_1(\beta, \gamma) \int_0^\alpha \frac{\pi^\beta + x^{-s\beta}}{(1 + (x/2\pi)^{2m})^{\frac{2m-s}{2m}}} x dx \\
&\leq T^2 \pi C_1(\beta, \gamma) \int_0^\alpha (\pi^\beta + x^{-s\beta}) x dx \\
&= C_1(\beta, \gamma) T^2 \alpha^2 \left( \frac{\pi^{1+\beta}}{2} + \frac{\pi \alpha^{-\beta s}}{2 - \beta s} \right)
\end{aligned}$$

For the second term, we have

$$\begin{aligned}
\frac{1}{\pi} \int_\alpha^\infty \frac{R(x)}{x} dx &\leq \frac{C_1(\beta, \gamma)}{\pi} \int_\alpha^\infty \frac{(\pi + x^{-s})^\beta}{x(1 + (x/2\pi)^{2m})^{\frac{2m-s}{2m}}} dx \\
&\leq \frac{C_1(\beta, \gamma)}{\pi} \int_\alpha^\infty \frac{\pi^\beta + \alpha^{-\beta s}}{x(1 + x^{2m-s}/(2\pi)^{2m-s})} dx \\
&\leq \frac{C_1(\beta, \gamma)}{\pi} \int_\alpha^{2\pi} \frac{\pi^\beta + \alpha^{-\beta s}}{x} dx + \frac{C_1(\beta, \gamma)}{\pi} \int_{2\pi}^\infty \frac{\pi^\beta + \alpha^{-\beta s}}{x^{2m+1-s}/(2\pi)^{2m-s}} dx \\
&= \frac{C_1(\beta, \gamma)}{\pi} (\pi^\beta + \alpha^{-s\beta}) \log(\frac{2\pi}{\alpha}) + \frac{C_1(\beta, \gamma)}{\pi} (\pi^\beta + \alpha^{-\beta s})/(2m - s).
\end{aligned}$$

So $c(T)$ is bounded by

$$c(T) \leq \frac{2\pi T^2 \alpha^2}{\gamma^2 \beta^{1+\gamma}} \left( \frac{\pi^\beta}{2} + \frac{\alpha^{-\beta s}}{2 - \beta s} \right) + \frac{2(\pi^\beta + \alpha^{-s\beta}) \log(\frac{2\pi}{\alpha})}{\pi \gamma^2 \beta^{1+\gamma}} + \frac{2\pi^\beta + 2\alpha^{-\beta s}}{\pi \gamma^2 \beta^{1+\gamma}(2m - s)}.$$

By choosing $\alpha = 1/T$, $\beta = 1/\log(T)$, $\gamma = 1/\log(\log(T))$, we have

$$T\alpha = 1, \quad \alpha^{-\beta} = T^{1/\log(T)} = e, \quad \beta^{1+\gamma} = (\log(T))^{1/\log(\log(T))} = e,$$

and, since $s \geq 1$ (by assumption), $2m - s \geq 1$ (by definition of $m$) and $T > 3$ (by assumption), we have

$$c(T) \leq C_0 \log(T) \log^2 \log(T) \left( \frac{2\pi^2 e}{2} + \frac{2\pi e^{1+s}}{2 - s} + \frac{2(\pi + e^s) \log(2\pi T)}{\pi} + \frac{2\pi + e^s}{\pi(2m - s)} \right)$$

$$\leq C_0 \log^2(T) \ \log^2 \log(T) \ \left( \frac{2\pi^2 e}{2} + 2\pi e^2 + \frac{2(\pi + 2e) \log(2\pi)}{\pi} + \frac{2e}{\pi} + 2 \right)$$

$$\leq (4\pi^2 e^2)^2 \log^2(T) \ \log^2 \log(T). \qquad \square$$

## F  Proofs for Section 5 (Curved Losses)

**Proposition 3.** *Let $\ell_t : \mathcal{W} \to \mathbb{R}$ be a $\beta$-exp-concave function, let $\mathcal{H}$ be an RKHS with feature map $\phi(t) \in \mathcal{H}$, and define $\widetilde{\ell}_t(W) = \ell_t(W\phi(t))$ for $W \in L(\mathcal{H}, \mathcal{W})$. Then for any $X, Y \in L(\mathcal{H}, \mathcal{W})$,*

$$\widetilde{\ell}_t(X) - \widetilde{\ell}_t(Y) \leq \left\langle \nabla \widetilde{\ell}_t(X), X - Y \right\rangle_{\mathrm{HS}} - \frac{\beta}{2} \left\langle \nabla \widetilde{\ell}(X), X - Y \right\rangle_{\mathrm{HS}}^2.$$

*Proof.* Let $x = X\phi(t)$ and $y = Y\phi(t)$. By definition and $\beta$-exp-concavity of $\ell_t$, we have

$$\widetilde{\ell}_t(X) - \widetilde{\ell}_t(Y) = \ell_t(x) - \ell_t(y) \leq \langle \nabla \ell_t(x), x - y \rangle - \frac{\beta}{2} \langle \nabla \ell_t(x), x - y \rangle_{\mathcal{W}}^2$$

$$= \langle \nabla \ell_t(x), (X - Y)\phi(t) \rangle_{\mathcal{W}} - \frac{\beta}{2} \langle \nabla \ell_t(x), (X - Y)\phi(t) \rangle_{\mathcal{W}}^2$$

$$= \langle \nabla \ell_t(x) \otimes \phi(t), (X - Y) \rangle_{\mathrm{HS}} - \frac{\beta}{2} \langle \nabla \ell_t(x) \otimes \phi(t), (X - Y) \rangle_{\mathrm{HS}}^2.$$

Observing that $\nabla \widetilde{\ell}(X) = \nabla \ell_t(x) \otimes \phi(t) \in L(\mathcal{H}, \mathcal{W})$ completes the proof. $\qquad \square$

### F.1  Strongly-convex Losses

In this section we show how to apply our static-to-dynamic reduction in the context of strongly-convex losses. Interestingly, the algorithm ends up being essentially the same as the Kernelized-ONS algorithm of [27], but with a weighted norm defined in terms of the feature covariance operator, $\Sigma_t = \lambda I + \beta \sum_{s=1}^t \phi(s) \otimes \phi(s)$. The following lemma shows how to connect the instantaneous regret on round $t$ to the kernelized linear losses $g_t \otimes \phi(t)$ and is analogous to Proposition 3.

**Proposition 9.** *Let $\ell_t : \mathcal{W} \to \mathbb{R}$ be a $\beta$-strongly-convex function, let $\mathcal{H}$ be an RKHS with associated feature map $\phi(t) \in \mathcal{H}$, and define $\widetilde{\ell}_t(W\phi(t))$ for $W \in L(\mathcal{H}, \mathcal{W})$. Then for any $X, Y \in L(\mathcal{H}, \mathcal{W})$,*

$$\widetilde{\ell}_t(X) - \widetilde{\ell}_t(Y) \leq \left\langle \widetilde{\ell}_t(X), X - Y \right\rangle_{\mathrm{HS}} - \frac{\beta}{2} \langle (X - Y)(\phi(t) \otimes \phi(t)), X - Y \rangle_{\mathrm{HS}},$$

*where $\phi(t) \otimes \phi(t) : \mathcal{H} \to \mathcal{H}$ is the operator with action $(\phi(t) \otimes \phi(t))h = \langle \phi(t), h \rangle_{\mathcal{H}} \phi(t)$.*

*Proof.* Let $x = X\phi(t)$ and $y = Y\phi(t)$, and observe that by $\beta$-strong-convexity of $\ell_t$ in $\mathcal{W}$ we have

$$\widetilde{\ell}_t(X) - \widetilde{\ell}_t(Y) = \ell_t(x) - \ell_t(y) \leq \langle \nabla \ell_t(x), x - y \rangle_{\mathcal{W}} - \frac{\beta}{2} \|x - y\|_{\mathcal{W}}^2$$

$$= \langle \nabla \ell_t(x), (X - Y)\phi(t) \rangle_{\mathcal{W}} - \frac{\beta}{2} \langle (X - Y)\phi(t), (X - Y)\phi(t) \rangle_{\mathcal{W}}$$

$$\overset{(\star)}{=} \langle \nabla \ell_t(x) \otimes \phi(t), X - Y \rangle_{\mathrm{HS}} - \frac{\beta}{2} \langle (X - Y)\phi(t) \otimes \phi(t), (X - Y) \rangle_{\mathcal{W}}$$

$$= \left\langle \nabla \widetilde{\ell}_t(x), X - Y \right\rangle_{\mathrm{HS}} - \frac{\beta}{2} \langle (X - Y)\phi(t) \otimes \phi(t), (X - Y) \rangle_{\mathcal{W}},$$

where $(\star)$ uses Lemma 8 and the last line observes that $\nabla \ell_t(x) \otimes \phi(t) = \nabla \widetilde{\ell}_t(X)$. $\qquad \square$

Using this result it is straight-forward to see that the usual ONS arguments work in this setting. For instance, by running mirror descent with regularizer $\psi_t(W) = \frac{1}{2}\langle W\Sigma_t, W\rangle_{\mathrm{HS}}$ where $\Sigma_t = \lambda I + \beta \sum_{s=1}^{t} \phi(t) \otimes \phi(t)$, we have the following regret guarantee.

**Theorem 7.** *(K-ONS for Strongly-convex Losses) Let $\ell_1, \ldots, \ell_T$ be a sequence of $\beta$-strongly convex losses. Let $\lambda > 0$ and for all $t$, define $\Sigma_t = \lambda I + \beta \sum_{s=1}^{t} \phi(s) \otimes \phi(s)$ and $\|W\|_{\Sigma_t}^2 = \langle W\Sigma_t, W\rangle_{\mathrm{HS}}$ for $W \in L(\mathcal{H}, \mathcal{W})$. Suppose that on each round $\mathcal{A}$ updates*

$$W_{t+1} = \underset{W \in L(\mathcal{H},\mathcal{W})}{\arg\min} \ \langle G_t, W\rangle_{\mathrm{HS}} + \frac{1}{2}\|W - W_t\|_{\Sigma_t}^2 ,$$

*starting from $W_1 = \mathbf{0} \in L(\mathcal{H}, \mathcal{W})$. Then for any $u_1, \ldots, u_T$ in $\mathcal{W}$ and $U \in L(\mathcal{H}, \mathcal{W})$ satisfying $u(t) = U\phi(t)$ for all $t$, Algorithm 1 applied with $\mathcal{A}$ guarantees*

$$R_T(u_1, \ldots, u_T) \leq \frac{\lambda}{2}\|U\|_{\mathrm{HS}}^2 + \frac{G^2}{2\beta}d_{\mathrm{eff}}\left(\lambda/\beta\right)\ln\left(e + \frac{e\beta\lambda_{\max}(K_T)}{\lambda}\right) ,$$

*where $K_T = (\langle \phi(s), \phi(t)\rangle_{\mathcal{H}})_{s,t\in[T]}$ and $d_{\mathrm{eff}}(\lambda) = \mathrm{Tr}\left(K_T(\lambda I + K_T)^{-1}\right)$.*

*Proof.* Applying Theorem 1 followed by Proposition 9, we have

$$R_T(u_1, \ldots, u_T) = \widetilde{R}_T(U) = \sum_{t=1}^{T} \widetilde{\ell}_t(W_t) - \widetilde{\ell}_t(U)$$

$$\leq \sum_{t=1}^{T} \langle G_t, W_t - U\rangle_{\mathrm{HS}} - \frac{\beta}{2}\langle (W_t - U)(\phi(t) \otimes \phi(t)), W_t - U\rangle_{\mathrm{HS}}$$

$$\overset{(a)}{\leq} \sum_{t=1}^{T} \frac{1}{2}\|U - W_t\|_{\Sigma_t}^2 - \frac{1}{2}\|U - W_{t+1}\|_{\Sigma_t}^2 - \frac{\beta}{2}\langle (W_t - U)(\phi(t) \otimes \phi(t)), W_t - U\rangle_{\mathrm{HS}}$$

$$+ \sum_{t=1}^{T} \langle G_t, W_t - W_{t+1}\rangle_{\mathrm{HS}} - \frac{1}{2}\|W_{t+1} - W_t\|_{\Sigma_t}^2$$

$$\overset{(b)}{\leq} \sum_{t=1}^{T} \frac{1}{2}\|U - W_t\|_{\Sigma_{t-1}}^2 - \frac{1}{2}\|U - W_{t+1}\|_{\Sigma_t}^2 + \sum_{t=1}^{T} \frac{1}{2}\|g_t\|_{\mathcal{W}}^2 \|\phi(t)\|_{\Sigma_t^{-1}}^2$$

$$\leq \frac{\lambda}{2}\|U\|_{\mathrm{HS}}^2 + \sum_{t=1}^{T} \frac{G^2}{2}\|\phi(t)\|_{\Sigma_t^{-1}}^2$$

$$\overset{(c)}{\leq} \frac{\lambda}{2}\|U\|_{\mathrm{HS}}^2 + \frac{G^2}{2\beta}d_{\mathrm{eff}}\left(\lambda/\beta\right)\ln\left(e + \frac{e\beta\lambda_{\max}(K_T)}{\lambda}\right),$$

where $(a)$ applies the standard bound for online mirror descent, $(b)$ observes that

$$\frac{1}{2}\|X - Y\|_{\Sigma_t}^2 - \frac{\beta}{2}\langle (X - Y)(\phi(t) \otimes \phi(t)), (X - Y)\rangle_{\mathrm{HS}} = \frac{1}{2}\langle (X - Y)\Sigma_{t-1}, X - Y\rangle_{\mathrm{HS}}$$

$$= \frac{1}{2}\|X - Y\|_{\Sigma_{t-1}}^2$$

and uses Fenchel-Young inequality to bound

$$\langle G_t, W_t - W_{t+1}\rangle_{\mathrm{HS}} - \frac{1}{2}\|W_{t+1} - W_t\|_{\Sigma_t}^2 \leq \frac{1}{2}\left\|G_t\Sigma_t^{-\frac{1}{2}}\right\|_{\mathrm{HS}}^2 = \frac{1}{2}\|g_t\|_{\mathcal{W}}^2 \|\phi(t)\|_{\Sigma_t^{-1}}^2$$

and $(c)$ uses a mild generalization of the usual log-determinant lemma (Lemmas 6 and 7) and defines defines $K_T = (\langle \phi(s), \phi(t)\rangle_{\mathcal{H}})_{s,t\in[T]}$. $\square$

Note that in the static regret setting, it is possible to avoid the dependence on the comparator norm entirely and pay only the logarithmic penalty—we do not expect such an improvement to be possible here since it would violate known $\Omega(P_T)$ lower bounds for strongly-convex losses [52].

## F.2 Additional Details for Example 3

In this section we provide some extra details showing that for the RKHS $\mathcal{H}$ associated with kernel $k(t,s) = \min(s,t)$, we can bound $\|u\|_{\mathcal{H}}^2 = \|\nabla u\|_{L^2}^2 = \mathcal{O}\left(\sqrt{\sum_t \|u_t - u_{t-1}\|_{\mathcal{W}}^2}\right)$ (Theorem 8) and $d_{\text{eff}}(\lambda) = \mathcal{O}(T/\sqrt{\lambda})$ (Theorem 9). We begin with the bound on the continuous squared path-lenth $\|\nabla u\|_{L^2}^2$.

**Theorem 8.** *Let $\mathcal{H}$ be the RKHS associated to the kernel $k(s,t) = \min(s,t)$ on $[0,T]$. Then for any $v_1, \ldots, v_T \in \mathbb{R}^d$ there exists a function $u \in \mathcal{H}$ such that $u(t) = v_t$ for all $t \in [T]$ and*

$$\|u\|_{\mathcal{H}}^2 = \|\nabla u\|_{L^2}^2 \leq C \sum_{t=2}^T \|v_t - v_{t-1}\|_{\mathcal{W}}^2 + C\|v_1\|_{\mathcal{W}}^2,$$

*with $C \leq \frac{5}{4}$.*

*Proof.* We assume without loss of generality that $v_t \in \mathbb{R}$ since the result extends immediately to $\mathbb{R}^d$ via the coordinate-wise extension mentioned in Section 2. For brevity we will define $v_t = 0$ for $t \notin \{1, \ldots, T\}$ so that we can write $\sum_{t=1}^T |v_t - v_{t-1}|^2 + |v_1|^2 = \sum_t |v_t - v_{t-1}|^2$.

Note that the RKHS associated with kernel $k(s,t) = \min(s,t)$ is $\mathcal{H} = \{f \in L^2 : f' \in L^2, f(0) = 0\}$, with associated norm $\|f\|_{\mathcal{H}} = \|\nabla f\|_{L^2} = \int |\nabla f(x)|^2 dx$ (see, e.g., Example 23 of Berlinet and Thomas-Agnan [6] with $m = 1$). Now suppose we define

$$u(t) = \sum_{i=1}^T v_i \text{sinc}(t - i)$$

where $\text{sinc}(x) = \sin(\pi x)/\pi x$. Then $u$ and $u'$ are square integrable and $u(0) = 0$, so $u \in \mathcal{H}$. Moreover, the norm associated with $\mathcal{H}$ is $\|f\|_{\mathcal{H}}^2 = \int |\nabla f(x)|^2 dx = \|\nabla f\|_{L^2}^2$, so we need only show that the constructed function $u(t)$ has $\|\nabla u\|_{L^2}^2 \leq \mathcal{O}(\sum_t |v_t - v_{t-1}|^2)$.

Denote $v(t) = \sum_i v_i \delta(t - i) = v_t$ and observe that we can write $u(t) = \sum_{i=1}^T v(i)\text{sinc}(t - i) = (v \star \text{sinc})(t)$, so using the fact that the Fourier transform of sinc is the rectangle function $1_{[-\frac{1}{2}, \frac{1}{2}]}(\omega) = \mathbb{I}\{\omega \in [-\frac{1}{2}, \frac{1}{2}]\}$ (see, e.g., Kammler [28]), we have $\widehat{u}(\omega) = \widehat{v \star \text{sinc}}(\omega) = \widehat{v}(\omega) 1_{[-\frac{1}{2}, \frac{1}{2}]}(\omega)$. Thus,

$$\|\nabla u\|_{L^2}^2 = \int_{\mathbb{R}} |\nabla u(t)|^2 dt = \int_{\mathbb{R}} \omega^2 |\widehat{u}(\omega)|^2 d\omega = \int_{-\frac{1}{2}}^{\frac{1}{2}} \omega^2 |\widehat{v}(\omega)|^2 d\omega$$

via Parseval's identity. We proceed by relating $\widehat{v}(\omega)$ to the DFT of the difference sequence, $\Delta v(t) = v_t - v_{t-1}$ and then applying Parseval's inequality for sequences to get $\int |\widehat{\Delta v}(\omega)|^2 \leq \sum_t |\Delta v(t)|^2 = \sum_t |v_t - v_{t-1}|^2$.

Observe that the DFT of the difference sequence is

$$\widehat{\Delta v}(\omega) = \sum_t (v_t - v_{t-1})e^{-2\pi i \omega t} = (1 - e^{-2\pi i \omega}) \sum_t v_t e^{-\pi i \omega t} = (1 - e^{-2\pi i \omega})\widehat{v}(\omega).$$

Thus,

$$\|\nabla u\|_{L^2}^2 = \int_{-\frac{1}{2}}^{\frac{1}{2}} \omega^2 |\widehat{v}(\omega)|^2 d\omega$$

$$= \int_{-\frac{1}{2}}^{\frac{1}{2}} \frac{\omega^2}{|1 - e^{-\pi i \omega}|^2} |\widehat{\Delta v}(\omega)|^2 d\omega$$

Now observe that using the identity $1 - \cos(x) = 2\sin^2(x/2)$ we have

$$|1 - e^{2\pi i \omega}|^2 = (1 - e^{-\pi i \omega})(1 - e^{\pi i \omega}) = 2 - e^{-\pi i \omega} - e^{\pi i \omega}$$
$$= 2(1 - \cos(\omega)) = 4\sin^2(\omega/2),$$

then, using $\omega^2 / \sin^2(\omega/2) \leq 5$ on $[-\frac{1}{2}, \frac{1}{2}]$ we have

$$\|\nabla u\|_{L^2}^2 = \int_{-\frac{1}{2}}^{\frac{1}{2}} \frac{\omega^2}{4 \sin^2(\omega/2)} |\widehat{\Delta v}(\omega)|^2 d\omega$$

$$\leq \frac{5}{4} \int_{-\frac{1}{2}}^{\frac{1}{2}} |\widehat{\Delta v}(\omega)|^2 d\omega = \frac{5}{4} \sum_t |\Delta v(t)|^2$$

$$= \frac{5}{4} \sum_t |v_t - v_{t-1}|^2$$

where the last line applies Parseval's identity for sequences.

$\square$

Next, the following theorem shows that the effective dimension of the linear spline kernel is indeed $\mathcal{O}(T/\sqrt{\lambda})$.

**Theorem 9.** *Let $K_T \in \mathbb{R}^{T \times T}$ be the matrix with entries $[K_T]_{ij} = \min(i,j)$. Then*

$$d_{\text{eff}}(\lambda) := \text{Tr}\left(K_T(\lambda I + K_T)^{-1}\right) \leq \frac{\pi T}{2\sqrt{\lambda}}.$$

*Proof.* By Lemma 12, the inverse of a matrix $K_T$ with entries $[K_T]_{ij} = \min(i,j)$ is the tri-diagonal matrix of the form

$$K_T^{-1} = \begin{pmatrix} 2 & -1 & 0 & 0 & \ldots & 0 \\ -1 & 2 & -1 & 0 & \ldots & 0 \\ 0 & -1 & 2 & -1 & \ldots & 0 \\ 0 & 0 & -1 & 2 & \ldots & 0 \\ \vdots & & & & \ddots & \\ 0 & 0 & 0 & 0 & \ldots & 1 \end{pmatrix}. \tag{9}$$

The eigenvalues of matrices of this form are well-known [17, 32, 43] and have a closed form expression:

$$\lambda_k(K_T^{-1}) = 2\left(1 - \cos\left(\frac{2k\pi}{(2T+1)}\right)\right) = 4\sin^2\left(\frac{k\pi}{2T+1}\right),$$

where the second equality uses the identity $1 - \cos(x) = 2\sin^2(x/2)$. Moreover, using the fact that $\sin(x)$ is concave on $[0, \pi/2]$ we can bound $\sin(x) \geq \frac{2}{\pi}x$, so the eigenvalues of $K_T^{-1}$ can be bounded as

$$\lambda_k(K_T^{-1}) = 4\sin^2\left(\frac{k\pi}{2T+1}\right) \geq 4 \cdot \frac{4}{\pi^2} \cdot \frac{\pi^2 k^2}{(2T+1)^2} \geq \frac{k^2}{T^2}$$

Thus, via direct calculation of the effective dimension $d_{\text{eff}}(\lambda) = \text{Tr}\left(K_T(\lambda I + K_T)^{-1}\right) = \sum_{k=1}^{T} \frac{\lambda_k(K_T)}{\lambda_k(K_T) + \lambda}$, we have

$$d_{\text{eff}}(\lambda) = \sum_{k=1}^{T} \frac{\lambda_k(K_T)}{\lambda_k(K_T) + \lambda} = \sum_{k=1}^{T} \frac{1}{1 + \lambda/\lambda_k(K_T)} = \sum_{k=1}^{T} \frac{1}{1 + \lambda \lambda_k(K_T^{-1})}$$

$$\leq \sum_{k=1}^{T} \frac{1}{1 + \lambda k^2/T^2} \leq \int_0^T \frac{1}{1 + \frac{\lambda}{T^2}x^2} dx \overset{(a)}{=} \frac{T}{\sqrt{\lambda}} \int_0^{\sqrt{\lambda}} \frac{1}{1 + u^2} du$$

$$\overset{(b)}{=} \frac{T}{\sqrt{\lambda}} \arctan(x)\Big|_{x=0}^{\sqrt{\lambda}} \overset{(c)}{\leq} \frac{\pi T}{2\sqrt{\lambda}}$$

where $(a)$ makes a change of variables $x = T/\sqrt{\lambda} u$, $(b)$ uses the fact that $\int_a^b \frac{1}{1+u^2} du = \arctan(x)\Big|_a^b$ and $(c)$ uses $|\arctan(x)| \leq \pi/2$ for all $x$. $\square$

# G   Proofs for Section 6 (Directional Adaptivity)

In this section we provide the full statement and proof of Proposition 5.

**Proposition 10.** *Let $\ell_1, \ldots, \ell_T$ be a sequence of $G$-Lipschitz losses and for all $t$, let $g_t \in \partial \ell_t(w_t)$ and define $G_t = g_t \otimes \phi(t) \in L(\mathcal{H}, \mathcal{W})$, $G_0 = G\sqrt{\max_t k(t,t)}$ and $\Sigma_t = (\lambda + G_0^2)I + \sum_{s=1}^{t-1} G_t \otimes G_t.$[7] Let $(\|\cdot\|_t, \|\cdot\|_{t,*})$ be a dual-norm pair characterized by $\|W\|_t = \sqrt{\langle W, \Sigma_t W \rangle_{\mathrm{HS}}}$. Let $\epsilon > 0$, and for all $t$ let $V_t = 4G_0^2 + \sum_{s=1}^{t-1} \|G_t\|_{t,*}^2$, $\alpha_t = \frac{\epsilon G_0}{\sqrt{V_t} \log^2(V_t/G_0^2)}$ and set $\psi_t(W) = k \int_0^{\|W\|_t} \min_{\eta \leq 1/G_0} \left[ \frac{\ln(x/\alpha_t + 1)}{\eta} + \eta V_t \right] dx.$*

*Suppose on each round we set $W_t = \arg\min_{W \in L(\mathcal{H}, \mathcal{W})} \left\langle \sum_{s=1}^{t-1} G_s, W \right\rangle + \psi_t(W)$ and we play $w_t = W_t \phi(t)$. Then for $u_1, \ldots, u_T$ in $\mathcal{W}$ and $U \in L(\mathcal{H}, \mathcal{W})$ satisfying $u_t = U\phi(t)$ for all $t$,*

$$\widetilde{\mathrm{R}}_T(U) = \widetilde{\mathcal{O}}\left( \epsilon G_0 + \sqrt{d_{\mathrm{eff}}(\lambda) \left[ (\lambda + G_0^2) \|U\|_{\mathrm{HS}}^2 + \sum_{t=1}^T \langle g_t, u_t \rangle^2 \right] \ln\left( e + \frac{e\lambda_{\max}(K_T)}{\lambda} \right)} \right)$$

*where $K_T = (\langle g_t, g_s \rangle k(t,s))_{t,s \in [T]}$, and $d_{\mathrm{eff}}(\lambda) = \mathrm{Tr}\left( K_T(\lambda I + K_T)^{-1} \right)$*

*Proof.* The result follows as a special case of Theorem 10 with sequence of non-decreasing norms characterized by $\|W\|_t = \sqrt{\langle W, \Sigma_t W \rangle_{\mathcal{W}}}$ and Lipschitz constant $G_0 = G\sqrt{\max_t k(t,t)} \geq \left\| \nabla \widetilde{\ell}_t(W_t) \right\|$ for all $t$. First, observe that

$$\langle G_t, U \rangle_{\mathrm{HS}}^2 = \langle g_t \otimes \phi(t), U \rangle_{\mathrm{HS}}^2 = \langle g_t, U\phi(t) \rangle_{\mathcal{W}}^2 = \langle g_t, u_t \rangle_{\mathcal{W}}^2,$$

hence from the static regret guarantee of Theorem 10, we get

$$\widetilde{\mathrm{R}}_T(U) = \widetilde{\mathcal{O}}\left( \epsilon G_0 + \|U\|_{T+1} \sqrt{\sum_{t=1}^T \|G_t\|_{t,*}^2} \right)$$

$$= \widetilde{\mathcal{O}}\left( \epsilon G_0 + \sqrt{\left( (\lambda + G_0^2) \|U\|_{\mathrm{HS}}^2 + \sum_{t=1}^T \langle G_t, U \rangle_{\mathrm{HS}}^2 \right) \sum_{t=1}^T \|G_t\|_{t,*}^2} \right)$$

$$= \widetilde{\mathcal{O}}\left( \epsilon G_0 + \sqrt{\left( (\lambda + G_0^2) \|U\|_{\mathrm{HS}}^2 + \sum_{t=1}^T \langle g_t, u_t \rangle^2 \right) \sum_{t=1}^T \|G_t\|_{t,*}^2} \right).$$

Moreover, observing that

$$\|G_t\|_{t,*}^2 = \left\langle G_t, \left( (\lambda + G_0^2)I + \sum_{s=1}^{t-1} G_s \otimes G_s \right)^{-1} G_t \right\rangle \leq \left\langle G_t, \left( \lambda I + \sum_{s=1}^t G_s \otimes G_s \right)^{-1} G_t \right\rangle,$$

we have via Lemma 7 that

$$\sum_{t=1}^T \|G_t\|_{t,*}^2 \leq d_{\mathrm{eff}}(\lambda) \ln\left( e + \frac{e\lambda_{\max}(K_T)}{\lambda} \right),$$

where $K_T$ is the gram matrix with entries $[K_T]_{ij} = \langle g_i, g_j \rangle k(i,j)$ and $d_{\mathrm{eff}}(\lambda) = \mathrm{Tr}\left( K_T(\lambda I + K_T)^{-1} \right) = \sum_{k=1}^T \frac{\lambda_k(K_T)}{\lambda + \lambda_k(K_T)}$.

Hence the dynamic regret $\mathrm{R}_T(u_1, \ldots, u_T) = \widetilde{\mathrm{R}}_T(U)$ can be bound above by

$$\widetilde{\mathcal{O}}\left( \epsilon G_0 + \sqrt{d_{\mathrm{eff}}(\lambda) \left[ (\lambda + G_0^2) \|U\|_{\mathrm{HS}}^2 + \sum_{t=1}^T \langle g_t, u_t \rangle^2 \right] \ln\left( 1 + \frac{\lambda_{\max}(K_T)}{\lambda} \right)} \right).$$

$\square$

---

[7]Here, the tensor product $G_t \otimes G_t$ is the map such that for $V \in L(\mathcal{H}, \mathcal{W})$, $(G_t \otimes G_t)(V) = \langle G_t, V \rangle_{\mathrm{HS}} G_t \in L(\mathcal{H}, \mathcal{W})$. Note that $\Sigma_t$ is a self-adjoint operator.

## G.1 Directional Adaptivity via Varying-norms

For completeness we provide a mild generalization of the static regret algorithm of [23] to leverage an arbitrary sequence of increasing norms. A similar technique has been used to get full-matrix parameter-free rates by [13].

The analysis remains mostly the same as Jacobsen and Cutkosky [23], but their analysis of the stability term bounds $-D_{\psi_t}(w_{t+1}|w_t)$ via a lemma that assumes that $\psi_t(w) = \Psi_t(\|w\|_2)$ for $w \in \mathbb{R}^d$. To obtain a full-matrix version of their result, we would instead like to have $\Psi_t(\|w\|_M)$, where $\|\cdot\|_M$ is a weighted norm w.r.t. to the inner product $\langle \cdot, \cdot \rangle_{\mathcal{W}}$ on an arbitrary Hilbert space $\mathcal{W}$. In what follows, we drop the dependence on $\mathcal{W}$ for brevity and simply write $\langle \cdot, \cdot \rangle$.

We first state and prove the main result of this section. The proof will rely on a few technical lemmas, which we state and prove at the end of the section in [Appendix G.1.1](#).

**Theorem 10.** *Let $\mathcal{W}$ be a Hilbert space and let $\langle \cdot, \cdot \rangle$ denote the associated inner product, let $\|\cdot\|_1, \ldots, \|\cdot\|_{T+1}$ be an arbitrary sequence of non-decreasing norms on $\mathcal{W}$, and let $\|\cdot\|_0 := \sqrt{\langle \cdot, \cdot \rangle} \le \|\cdot\|_t$ for all $t$. Let $\ell_1, \ldots, \ell_T$ be convex functions over $\mathcal{W}$ satisfying $\|g_t\|_{t,*} \le G$ for all $t$ and $g_t \in \partial \ell_t(w_t)$. Let $\epsilon, \lambda > 0$, $V_t = 4G^2 + \sum_{s=1}^{t-1} \|g_s\|_{s,*}^2$, $\alpha_t = \frac{\epsilon G}{\sqrt{V_t} \log^2(V_t/G^2)}$, and set $\psi_t(w) = 3 \int_0^{\|w\|_t} \min_{\eta \le \frac{1}{G}} \left[ \frac{\ln(x/\alpha_t + 1)}{\eta} + \eta V_t \right] dx$, and on each round update $w_{t+1} = \arg\min_{w \in \mathcal{W}} \left\langle \sum_{s=1}^t g_s, w \right\rangle + \psi_{t+1}(w)$. Then for all $u \in \mathcal{W}$,*

$$R_T(u) = \widehat{\mathcal{O}}\left( G\epsilon + \|u\| \left[ \sqrt{V_T \ln\left( \frac{\|u\|\sqrt{V_T}}{\epsilon G} + 1 \right)} \vee G \ln\left( \frac{\|u\|\sqrt{V_T}}{\epsilon G} + 1 \right) \right] \right)$$

*where $\widehat{\mathcal{O}}(\cdot)$ hides constant and $\log(\log)$ factors (but not $\log$ factors).*

*Proof.* Begin by applying the standard FTRL regret template (see, e.g., Orabona [39, Lemma 7.1]):

$$R_T(u) = \sum_{t=1}^T \langle g_t, w_t - u \rangle \le \psi_{T+1}(u) + \sum_{t=1}^T F_t(w_t) - F_{t+1}(w_{t+1}) + \langle g_t, w_t \rangle,$$

where $F_t(w) = \left\langle \sum_{s=1}^{t-1} g_s, w \right\rangle + \psi_t(w)$. Observe that the summation can be written as

$$\sum_{t=1}^T F_t(w_t) - F_{t+1}(w_{t+1}) + \langle g_t, w_t \rangle$$

$$= \sum_{t=1}^T \langle g_t, w_t - w_{t+1} \rangle + F_t(w_t) - F_t(w_{t+1}) + (\psi_t - \psi_{t+1})(w_{t+1})$$

$$\stackrel{(a)}{=} \sum_{t=1}^T \langle g_t, w_t - w_{t+1} \rangle + \langle \nabla F_t(w_t), w_t - w_{t+1} \rangle - D_{F_t}(w_{t+1}|w_t) + (\psi_t - \psi_{t+1})(w_{t+1})$$

$$\stackrel{(b)}{\le} \sum_{t=1}^T \langle g_t, w_t - w_{t+1} \rangle - D_{F_t}(w_{t+1}|w_t) + (\psi_t - \psi_{t+1})(w_{t+1})$$

$$\stackrel{(c)}{=} \sum_{t=1}^T \|g_t\|_{t,*} \|w_t - w_{t+1}\|_t - D_{\psi_t}(w_{t+1}|w_t) - (\psi_{t+1} - \psi_t)(w_{t+1}),$$

where $(a)$ uses the definition of Bregman divergence to write $f(a) - f(b) = \langle \nabla f(a), a - b \rangle - D_f(b|a)$, $(b)$ uses the fact that $w_t = \arg\min_{w \in \mathcal{W}} F_t(w)$, hence $\langle \nabla F_t(w_t), w_t - w_{t+1} \rangle \le 0$ by the first-order optimality condition, and $(c)$ uses the fact that Bregman divergences are invariant to linear terms, so from the definition of $F_t$ we have $D_{F_t}(\cdot|\cdot) = D_{\psi_t}(\cdot|\cdot)$. Moreover, since $\|\cdot\|_1, \ldots, \|\cdot\|_T$ is a non-decreasing sequence of norms, we can bound the terms

$$(\psi_{t+1} - \psi_t)(w) = \Psi_{t+1}(\|w\|_{t+1}) - \Psi_t(\|w\|_t) \ge \underbrace{\Psi_{t+1}(\|w\|_t) - \Psi_t(\|w\|_t)}_{=: \Delta_t^\Psi(\|w\|_t)},$$

so overall the regret is bounded by

$$R_T(u) \le \psi_{T+1}(u) + \sum_{t=1}^{T} \underbrace{\|g_t\|_{t,*} \|w_t - w_{t+1}\|_t - D_{\psi_t}(w_{t+1}|w_t) - \Delta_t^{\Psi}(\|w_{t+1}\|_t)}_{=:\delta_t}.$$

From here, the rest of the proof follows using the same arguments as [23], but using our Lemma 5 to bound $D_{\psi_t}(w_{t+1}|w_t) \ge \frac{1}{2} \|w_t - w_{t+1}\|^2 \Psi_t(\|\widetilde{w}\|_t)$ instead of their Lemma 7. $\qquad\square$

### G.1.1 A Stability Lemma for Weighted Norms

In this section generalize the stability lemma of Jacobsen and Cutkosky [23] to weighted norms $\|x\|_M = \sqrt{\langle x, Mx \rangle}$. This is the main technical detail needed for the proof of Theorem 10 that is not covered by the proof of their static regret algorithm. Throughout this section we assume the domain $\mathcal{W}$ is a Hilbert space with associated inner product $\langle \cdot, \cdot \rangle$. The following helper lemma follows via a straight-forward but somewhat tedious computation.

**Lemma 4.** *Let $g : \mathcal{W} \to \mathbb{R}$ be a convex function and let $f(x) = \sqrt{g(x)}$. Then for $x \in \mathcal{W}$ s.t. $g(x) > 0$ we have*

$$\nabla f(x) = \frac{\nabla g(x)}{2f(x)} \qquad \text{and} \qquad \nabla^2 f(x) = \frac{\nabla^2 g(x)}{2f(x)} - \frac{\nabla g(x) \otimes \nabla g(x)}{4f(x)^3},$$

*where $\otimes$ denotes the tensor product.*

Using this, we have the following Hessian bounds for elliptically-symmetric functions:

**Lemma 5.** *Let $M \in L(\mathcal{H}, \mathcal{H})$ be a positive definite linear operator and assume $M$ is self-adjoint w.r.t. $\langle \cdot, \cdot \rangle$. Let $\|x\|_M = \sqrt{\langle x, Mx \rangle}$ be the weighted norm induced by $M$ and let $\psi(w) = \Psi(\|w\|_M)$ for some convex function $\Psi : \mathbb{R} \to \mathbb{R}$. Then for any $w \in \mathcal{W}$ bounded away from $0$ and any $u \in \mathcal{W}$,*

$$\langle u, \nabla^2 \psi(w) u \rangle \ge \min \left\{ \Psi''(\|w\|_M), \frac{\Psi'(\|w\|_M)}{\|w\|_M} \right\} \|u\|_M^2$$

*Moreover, if $\Psi'(\cdot)$ is concave and non-negative, then for any $w \in \mathcal{W}$ bounded away from $0$ and $u \in \mathcal{W}$,*

$$\langle u, \nabla^2 \psi(w) u \rangle \ge \Psi''(\|w\|_M) \|u\|_M^2 .$$

*Proof.* The proof follows a similar argument to Orabona and Pál [41, Lemma 23]. Let us first compute the gradients of $f(x) = \|x\|_M = \sqrt{\langle x, Mx \rangle}$. Let $g(x) = \langle x, Mx \rangle$ and observe that if $M$ is self-adjoint w.r.t. $\langle \cdot, \cdot \rangle$, we have $\nabla g(x) = 2Mx$ and $\nabla^2 g(x) = 2M$. Hence, applying Lemma 4 we have

$$\nabla f(w) = \frac{2Mw}{2 \|w\|_M} = \frac{Mw}{\|w\|_M}$$

$$\nabla^2 f(w) = \frac{2M}{2 \|w\|_M} - \frac{4Mw \otimes Mw}{4 \|w\|_M^3} = \frac{M}{\|w\|_M} - \frac{Mw \otimes Mw}{\|w\|_M^3}.$$

Using this, we have

$$\nabla \psi(w) = \nabla \Psi(\|w\|_M) = \nabla \|w\|_M \Psi'(\|w\|_M) = \frac{Mw}{\|w\|_M} \Psi'(\|w\|_M),$$

and

$$\begin{aligned}
\nabla^2 \psi(w) &= \nabla \left( \nabla \|w\|_M \Psi_t'(\|w\|_M) \right) \\
&= \nabla^2 \|w\|_M \Psi'(\|w\|_M) + \Psi''(\|w\|_M) \left( \nabla \|w\|_M \otimes \nabla \|w\|_M \right) \\
&= \Psi_t'(\|w\|_M) \left( \frac{M}{\|w\|_M} - \frac{(Mw \otimes Mw)}{\|w\|_M^3} \right) + \Psi''(\|w\|_M) \frac{(Mw \otimes Mw)}{\|w\|_M^2} \\
&= \underbrace{\left( \frac{\Psi''(\|w\|_M)}{\|w\|_M^2} - \frac{\Psi'(\|w\|_M)}{\|w\|_M^3} \right)}_{=:\beta} (Mw \otimes Mw) + \underbrace{\frac{\Psi'(\|w\|_M)}{\|w\|_M}}_{=:\gamma} M \\
&= \beta(Mw \otimes Mw) + \gamma M.
\end{aligned}$$

Hence, for any $u \in \mathcal{W}$ we have

$$
\begin{aligned}
\langle u, \nabla^2 \psi(w) u \rangle &= \langle u, (\beta M w \otimes M w + \gamma M) u \rangle \\
&= \beta \langle u, M w \rangle^2 + \gamma \|u\|_M^2
\end{aligned}
$$

Now decompose $u = w + v$ for some $v$ such that $\langle M w, v \rangle = 0$; such a $v$ always exists for positive definite $M$. Then

$$
\begin{aligned}
\langle u, \nabla^2 \psi(w) u \rangle &= \beta \langle v + w, M w \rangle^2 + \gamma \|v + w\|_M^2 = \beta \|w\|_M^4 + \gamma \|w\|_M^2 + \gamma \|v\|_M^2 \\
&= \left( \frac{\Psi''(\|w\|_M)}{\|w\|_M^2} - \frac{\Psi'(\|w\|_M)}{\|w\|_M^3} \right) \|w\|_M^4 + \frac{\Psi'(\|w\|_M)}{\|w\|_M} \left( \|w\|_M^2 + \|v\|_M^2 \right) \\
&= \Psi''(\|w\|_M) \|w\|_M^2 + \frac{\Psi'(\|w\|_M)}{\|w\|_M} \|v\|_M^2 \\
&\geq \min \left\{ \Psi''(\|w\|_M), \frac{\Psi'(\|w\|_M)}{\|w\|_M} \right\} \left( \|w\|_M^2 + \|v\|_M^2 \right) \\
&= \min \left\{ \Psi''(\|w\|_M), \frac{\Psi'(\|w\|_M)}{\|w\|_M} \right\} \|u\|_M^2 ,
\end{aligned}
$$

where the last step uses the fact that $w$ and $v$ are orthogonal w.r.t. $M$.

For the second statement of the lemma, we need only show that $\Psi'(\|w\|_M)/\|w\|_M \geq \Psi''(\|w\|_M)$. This is indeed the case by concavity and non-negativity of $\Psi'(\cdot)$:

$$
\frac{\Psi'(\|w\|_M)}{\|w\|_M} \geq \frac{\Psi'(0) + \Psi''(\|w\|_M)(\|w\|_M - 0)}{\|w\|_M} \geq \Psi''(\|w\|_M) . \qquad \square
$$

# H  Supporting Lemmas

The following lemma is a straight-forward generalization of the usual log-determinant lemma (see, *e.g.*, [10, Lemma 11.11]), taking a bit of extra care to handle determinants of potentially infinite-dimensional linear operators.

**Lemma 6.** *Let $\mathcal{H}$ be a Hilbert space and for all $t$ let $v_t \in \mathcal{H}$. Suppose $v_t \otimes v_t : \mathcal{H} \to \mathcal{H}$ defines a bounded linear operator for all $t$ and suppose $A_t = A_{t-1} + v_t \otimes v_t$ for any $t \geq 1$, starting from $A_0 = I$. Then*

$$
\langle v_t, A_t^{-1} v_t \rangle = 1 - \frac{\operatorname{Det}(A_{t-1})}{\operatorname{Det}(A_t)}.
$$

*Proof.* Observe that for any $t$, we have $A_t = A_{t-1} + v_t \otimes v_t$, so re-arranging terms, factoring, and taking determinants of both sides we have

$$
\operatorname{Det}(A_t) \operatorname{Det}\left(I - A_t^{-1} v_t \otimes v_t\right) = \operatorname{Det}(A_{t-1}). \tag{10}
$$

Note that each of these determinants are well-defined in terms of the Fredholm determinant: each of the three terms above is a trace-class perturbation of the identity operator. Moreover, observe that $A_t^{-1}(v_t \otimes v_t)$ is a rank-one operator having single eigenvalue equal to $\lambda = \langle v_t, A_t^{-1} v_t \rangle$. Indeed, for any $w \in \mathcal{H}$ we have $A_t^{-1}(v_t \otimes v_t)(w) = \langle v_t, w \rangle A_t^{-1} v_t$, hence, for $w = A_t^{-1} v_t$ we have

$$
A_t^{-1}(v_t \otimes v_t)(w) = A_t^{-1}(v_t \otimes v_t)(A_t^{-1} v_t) = \langle v_t, A_t^{-1} v_t \rangle A_t^{-1} v_t = \lambda w .
$$

Therefore, from the standard rank-one perturbation identity for the determinant we have $\operatorname{Det}\left(I - A_t^{-1} v_t \otimes v_t\right) = 1 - \langle v_t, A_t^{-1} v_t \rangle$, so re-arranging Equation (10) yields

$$
\langle v_t, A_t^{-1} v_t \rangle = 1 - \frac{\operatorname{Det}(A_{t-1})}{\operatorname{Det}(A_t)} . \qquad \square
$$

**Lemma 7.** *Let $\mathcal{H}$ be a Hilbert space. For all $t$ let $G_t \in \mathcal{H}$ be a bounded linear operator and define $S_t = \lambda I + \sum_{s=1}^{t} G_s \otimes G_s$ for $\lambda > 0$. Then*

$$\sum_{t=1}^{T} \langle G_t, S_t^{-1} G_t \rangle \leq d_{\text{eff}}(\lambda) \ln \left( e + \frac{e \lambda_{\max}(K_T)}{\lambda} \right),$$

*where $K_T = (\langle g_t, g_s \rangle \, k(t,s))_{t,s \in [T]}$ and $d_{\text{eff}}(\lambda) = \text{Tr}\left( K_T(\lambda I + K_T)^{-1} \right)$.*

*Proof.* First apply Lemma 6 with $v_t = G_t/\sqrt{\lambda}$ followed by the elementary inequality $1 - x \leq \ln(1/x)$ to get

$$
\begin{aligned}
\sum_{t=1}^{T} \langle G_t, S_t^{-1} G_t \rangle &= \sum_{t=1}^{T} \frac{1}{\lambda} \left\langle G_t, (S_t/\lambda)^{-1} G_t \right\rangle \\
&= \sum_{t=1}^{T} 1 - \frac{\text{Det}\,(S_{t-1}/\lambda)}{\text{Det}\,(S_t/\lambda)} \\
&\leq \sum_{t=1}^{T} \ln \left( \frac{\text{Det}\,(S_t/\lambda)}{\text{Det}\,(S_{t-1}/\lambda)} \right) \\
&= \ln \left( \text{Det} \left( I + \frac{\sum_{t=1}^{T} G_t \otimes G_t}{\lambda} \right) \right) \\
&= \sum_{t=1}^{T} \ln \left( 1 + \frac{\lambda_t(K_T)}{\lambda} \right),
\end{aligned}
$$

where the last line uses the well-known fact that the gram matrix $K_T = (\langle g_t, g_s \rangle \, k(t,s))_{t,s \in [T]}$ and the empirical covariance operator $\sum_{t=1}^{T} G_t \otimes G_t$ have the same eigenvalues. Moreover, following Jézéquel et al. [27, Proposition 2] we can use the inequality $\ln(1 + x) \leq \frac{x}{1+x}(1 + \ln(1 + x))$ to expose a dependence on the effective dimension $d_{\text{eff}}(\lambda) = \text{Tr}\left( K_T(\lambda I + K_T)^{-1} \right)$ as follows:

$$
\begin{aligned}
\sum_{t=1}^{T} \langle G_t, S_t^{-1} G_t \rangle &\leq \sum_{t=1}^{T} \frac{\lambda_t(K_T)}{\lambda + \lambda_t(K_T)} \left[ 1 + \ln \left( 1 + \frac{\lambda_t(K_T)}{\lambda} \right) \right] \\
&\leq \left[ 1 + \ln \left( 1 + \frac{\lambda_{\max}(K_T)}{\lambda} \right) \right] \sum_{t=1}^{T} \frac{\lambda_t(K_T)}{\lambda + \lambda_t(K_t)} \\
&= \left[ 1 + \ln \left( 1 + \frac{\lambda_{\max}(K_T)}{\lambda} \right) \right] \text{Tr}\left( K_T(\lambda I + K_T)^{-1} \right) \\
&= d_{\text{eff}}(\lambda) \left[ 1 + \ln \left( 1 + \frac{\lambda_{\max}(K_T)}{\lambda} \right) \right] \\
&= d_{\text{eff}}(\lambda) \ln \left( e + \frac{e \lambda_{\max}(K_T)}{\lambda} \right) . \qquad \square
\end{aligned}
$$

**Lemma 8.** *Let $\mathcal{H}$ be a separable RKHS with associated feature map $\phi(t) \in \mathcal{H}$ and let $x \in \mathcal{H}$ satisfy $x(t) = X\phi(t)$ for some $X \in L(\mathcal{H}, \mathcal{W})$. Then*

$$\|x(t)\|_{\mathcal{W}}^2 = \langle X(\phi(t) \otimes \phi(t)), X \rangle_{\text{HS}},$$

*where $\phi(t) \otimes \phi(t) : \mathcal{H} \to \mathcal{H}$ is the linear operator with action $(\phi(t) \otimes \phi(t))h = \langle \phi(t), h \rangle_{\mathcal{H}} \, \phi(t)$.*

*Proof.* Let $h_1, h_2, \ldots$ be an orthonormal basis of $\mathcal{H}$. By definition of the Hilbert-Schmidt inner product, we have

$$
\begin{aligned}
\langle X(\phi(t) \otimes \phi(t)), X \rangle_{\mathrm{HS}} &= \sum_i \langle X(\phi(t) \otimes \phi(t)) h_i, X h_i \rangle_{\mathcal{W}} \\
&= \sum_i \langle \phi(t), h_i \rangle_{\mathcal{H}} \langle X \phi(t), X h_i \rangle_{\mathcal{W}} \\
&= \sum_i \langle \phi(t), h_i \rangle_{\mathcal{H}} \langle X^* X \phi(t), h_i \rangle_{\mathcal{H}} \\
&\overset{(\star)}{=} \langle \phi(t), X^* X \phi(t) \rangle_{\mathcal{H}} = \langle X \phi(t), X \phi(t) \rangle_{\mathcal{W}} \\
&= \langle x, x \rangle_{\mathcal{W}} = \|x\|_{\mathcal{W}}^2,
\end{aligned}
$$

where $X^* : \mathcal{W} \to \mathcal{H}$ is the adjoint of $X$ and $(\star)$ uses Parseval's identity. $\qquad\square$

**Lemma 9.** *Let $\mathcal{W} \subseteq \mathbb{R}^d$ and let $\mathcal{H}$ be an RKHS with associated feature map $\phi$. For all $t \in [T]$, let $G_t = g_t \otimes \phi(t) \in L(\mathcal{H}, \mathcal{W})$ denote the rank-one operator mapping $G_t(h) = \langle \phi(t), h \rangle g_t \in \mathcal{W}$. Then for any $t$,*

$$
\left\| \sum_{s=1}^t G_s \right\|_{\mathrm{HS}}^2 = \sum_{s,s'}^t k(s, s') \langle g_s, g_{s'} \rangle_{\mathcal{W}} .
$$

*Proof.* Let $h_1, h_2, \ldots$ be an orthonormal basis of $\mathcal{H}$. Observe that for any $h \in \mathcal{H}$, $\left( \sum_{s=1}^t G_s \right)(h) = \sum_{s=1}^t \langle \phi(s), h \rangle g_s$. Hence, by definition of the Hilbert-Schmidt norm,

$$
\begin{aligned}
\left\| \sum_{s=1}^t G_t \right\|_{\mathrm{HS}}^2 &= \sum_i \left\| \sum_{s=1}^t G_t h_i \right\|_{\mathcal{W}}^2 = \sum_i \left\langle \sum_{s=1}^t \langle \phi(s), h_i \rangle_{\mathcal{H}} g_s, \sum_{s'=1}^t \langle \phi(s'), h_i \rangle_{\mathcal{H}} g_{s'} \right\rangle_{\mathcal{W}} \\
&= \sum_i \sum_{s,s'}^t \langle \phi(s), h_i \rangle_{\mathcal{H}} \langle \phi(s'), h_i \rangle_{\mathcal{H}} \langle g_s, g'_s \rangle_{\mathcal{W}} \\
&= \sum_{s,s'} \langle g_s, g_{s'} \rangle_{\mathcal{W}} \sum_i \langle \phi(s), h_i \rangle_{\mathcal{H}} \langle \phi(s'), h_i \rangle_{\mathcal{H}} \\
&= \sum_{s,s'} \langle g_s, g_{s'} \rangle_{\mathcal{W}} k(s, s'),
\end{aligned}
$$

where the last line observes that for orthonormal basis $h_i$ we have $\sum_i \langle \phi(s), h_i \rangle_{\mathcal{H}} \langle \phi(s'), h_i \rangle_{\mathcal{H}} = \langle \phi(s), \phi(s') \rangle_{\mathcal{H}} = k(s, s')$. $\qquad\square$

The following theorem shows how to compute the norm of $G_t = g_t \otimes \phi(t)$, which is the auxiliary loss for OLO under our framework. Here we state the result in terms of $g_t \in \mathcal{W}^*$ for generality, but note that in the main text we implicitly invoke Riesz representation theorem to write $g_t \in \mathcal{W}$, $G_t \in L(\mathcal{H}, \mathcal{W})$, and $\|G_t\| = \|g_t\|_{\mathcal{W}} \sqrt{k(t,t)}$.

**Lemma 10.** *Let $\mathcal{H}$ be a RKHS with associated feature map $\phi(t)$ and let $\mathcal{W}$ be a Hilbert space. Let $\ell_t : \mathcal{W} \to \mathbb{R}$ be a differentiable function and for any $W \in L(\mathcal{H}, \mathcal{W})$ let $\widetilde{\ell}_t(W) = \ell_t(W\phi(t))$. Then for any $W \in L(\mathcal{H}, \mathcal{W})$, $g_t \in \partial \ell_t(W\phi(t))$, and $G_t = g_t \otimes \phi(t) \in \partial \widetilde{\ell}_t(W)$,*

$$
\|G_t\|_{\mathrm{HS}}^2 = \|g_t\|_{\mathcal{W}, *}^2 \, k(t, t),
$$

*where $k(s, t) = \langle \phi(s), \phi(t) \rangle_{\mathcal{H}}$ is the kernel associated with $\mathcal{H}$ and $\|\cdot\|_{\mathcal{W}, *}$ is the dual norm of $\|\cdot\|_{\mathcal{W}}$.*

*Proof.* We have via Lemma 11 that

$$
G_t := g_t \otimes \phi(t) \in \partial \widetilde{\ell}_t(W) \subseteq L(\mathcal{H}, \mathcal{W})^*,
$$

where $g_t \in \partial \ell_t(W\phi(t)) \subseteq \mathcal{W}^*$. By Riesz representation theorem, we can identify a $\widehat{g}_t \in \mathcal{W}$ such that for any $w \in \mathcal{W}$, $g_t(w) = \langle \widehat{g}_t, w \rangle_{\mathcal{W}}$, and likewise we can identify $G_t \in L(\mathcal{H}, \mathcal{W})^*$ with a rank-one operator $\widehat{G}_t \in L(\mathcal{H}, \mathcal{W})$ with action $\widehat{G}_t(h) = \langle \phi(t), h \rangle_{\mathcal{H}} \widehat{g}_t$. Hence, we have by definition of the Hilbert-Schmidt norm that for any orthonormal basis $\{h_i\}_i$ of $\mathcal{H}$,

$$\|G_t\|_{\mathrm{HS}}^2 = \sum_i \|G_t h_i\|_{\mathcal{W}}^2$$
$$= \sum_i \langle \phi(t), h_i \rangle_{\mathcal{H}}^2 \|\widehat{g}_t\|_{\mathcal{W}}^2$$
$$= \|g_t\|_{\mathcal{W},*} \|\phi(t)\|_{\mathcal{H}}^2,$$

where the last line again uses Riesz representation theorem to write $\|\widehat{g}_t\|_{\mathcal{W}} = \|g_t\|_{\mathcal{W},*}$ and then uses $\sum_i \langle \phi(t), h_i \rangle_{\mathcal{H}}^2 = \|\phi(t)\|_{\mathcal{H}}^2$ by Parseval's identity. Moreover, since $\phi(t)$ are the features of an RKHS with kernel $k$, we have

$$\|\phi(t)\|_{\mathcal{H}}^2 = \langle \phi(t), \phi(t) \rangle_{\mathcal{H}} = k(t,t) . \qquad \square$$

**Lemma 11.** *Let $\mathcal{H}$ be a RKHS with feature map $\phi(t) \in \mathcal{H}$, and let $\mathcal{W}$ be a Hilbert space. Let $\ell_t : \mathcal{W} \to \mathbb{R}$ be a convex function and let $\widetilde{\ell}_t(W) = \ell_t(W\phi(t))$ for $W \in L(\mathcal{H}, \mathcal{W})$. Then for any $W \in L(\mathcal{H}, \mathcal{W})$ and any $g_t \in \partial \ell_t(W\phi(t)) \subseteq \mathcal{W}^*$,*

$$G_t = g_t \otimes \phi(t) \in \partial \widetilde{\ell}_t(W) \in L(\mathcal{H}, \mathcal{W})^*,$$

*where $G_t \in L(\mathcal{H}, \mathcal{W})^*$ is the functional with action $G_t(W) = \langle g_t, W\phi(t) \rangle_{\mathcal{W}}$ for all $W \in L(\mathcal{H}, \mathcal{W})$.*

*Proof.* Let $W \in L(\mathcal{H}, \mathcal{W})$, $w_t = W\phi(t) \in \mathcal{W}$, and let $g_t \in \partial \ell_t(w_t) \subseteq \mathcal{W}^*$. Define $G_t = g_t \otimes \phi(t) \in L(\mathcal{H}, \mathcal{W})^*$ the functional on $L(\mathcal{H}, \mathcal{W})$ with action

$$G_t(W) = \langle g_t, W\phi(t) \rangle_{\mathcal{W}}, \quad \forall W \in L(\mathcal{H}, \mathcal{W}).$$

Now observe that for $g_t \in \partial \ell_t(w_t)$, for any $w \in \mathcal{W}$ we have

$$\ell_t(w) \geq \ell_t(w_t) + \langle g_t, w - w_t \rangle_{\mathcal{W}} = \ell_t(W\phi(t)) + \langle g_t, w - W\phi(t) \rangle_{\mathcal{W}},$$

hence for any $V \in L(\mathcal{H}, \mathcal{W})$ we can take $w = V\phi(t)$ to get

$$\ell_t(V\phi(t)) \geq \ell_t(W\phi(t)) + \langle g_t, V\phi(t) - W\phi(t) \rangle_{\mathcal{W}} = \ell_t(W\phi(t)) + \langle g_t, (V - W)\phi(t) \rangle_{\mathcal{W}}$$

that is,

$$\widetilde{\ell}_t(V) \geq \widetilde{\ell}_t(W) + G_t(V - W).$$

so $G_t = g_t \otimes \phi(t) \in \partial \widetilde{\ell}_t(W) \subseteq L(\mathcal{H}, \mathcal{W})^*$. $\qquad \square$

For completeness, the following lemma provides the inverse of a matrix with entries $K_{ij} = \min(i,j)$. A similar result can be seen in the proof of Jacobsen and Orabona [26, Lemma 4], where a variant of the matrix $K$ appears as an intermediate calculation.

**Lemma 12.** *Let $K \in \mathbb{R}^{T \times T}$ be a matrix with entries $K_{i,j} = \min(i,j)$. Then $K^{-1}$ is a tri-diagonal matrix of the form*

$$K^{-1} = \begin{pmatrix} 2 & -1 & 0 & 0 & \ldots & 0 & 0 \\ -1 & 2 & -1 & 0 & \ldots & 0 & 0 \\ 0 & -1 & 2 & -1 & \ldots & 0 & 0 \\ 0 & 0 & -1 & 2 & \ldots & 0 & 0 \\ \vdots & & & & \ddots & & \\ 0 & 0 & 0 & 0 & \ldots & 2 & -1 \\ 0 & 0 & 0 & 0 & \ldots & -1 & 1 \end{pmatrix}.$$

*Proof.* It can easily be checked that $K$ has Cholesky decomposition $K = U^\top U$ where $U$ is the upper-triangular matrix of $1's$. Hence, $K^{-1} = U^{-1}(U^\top)^{-1}$. Moreover, the inverse of $U$ is the first-order finite-differences operator with entries

$$\Sigma_{ij} = \begin{cases} 1 & \text{if } i = j \\ -1 & \text{if } j = i+1 \\ 0 & \text{otherwise} \end{cases}.$$

Indeed, $(U\Sigma)_{ij} = \sum_{k=1}^{T} U_{ik}\Sigma_{kj} = -U_{i,j-1} + U_{ij} = 1$ for $i = j$ and zero otherwise. Computing $K^{-1} = \Sigma\Sigma^\top$ yields the tri-diagonal matrix of the stated form. □

