# OpenReview forum: "Dynamic Regret Reduces to Kernelized Static Regret"
_NeurIPS.cc/2025/Conference — NeurIPS 2025 poster_

### Official Review · Reviewer_viCE · 2025-07-02

**Clarity:** 3
**Significance:** 3
**Originality:** 4
**Rating:** 5
**Confidence:** 3

**Summary:**

The authors study dynamic regret in online convex optimization, where the learner competes against a sequence of comparators in a convex set $\mathcal{W}$. Their key idea is to reinterpret the comparator sequence as a fixed _function_ from $[1,T]$ to $\mathcal{W}$, thereby reformulating the dynamic regret problem as a static regret problem in a _function space_. Specifically, they consider this function space to be a Reproducing Kernel Hilbert Space (RKHS) associated with a kernel $k(\cdot, \cdot)$. Under this framework, the $t$-th step of the online learning algorithm can be implemented in time $O(t)$ via the kernel trick, and a static regret bound in the RKHS yields a dynamic regret bound for the original problem. By choosing $k$ to be a carefully designed translation-invariant kernel, the algorithm achieves optimal path-length dependence without requiring knowledge of the time horizon. Additionally, they show that kernelized ONS with the spline kernel $\min\\{s,t\\}$ achieves the optimal regret rate for exp-concave online learning.

**Questions:**

## Clarifying Questions
- In the introduction of RKHS in Section 2, the authors denote the domain of functions in $\mathcal{H}$ as $\mathcal{X}$. If I understand correctly, in the context of this paper, $\mathcal{X}$ corresponds to the interval $[1,T]$. it would be helpful to make this more explicit. I also suggest using a more consistent notation such at $\mathcal{T}$.
- In Example 1, the FTRL update involves the function $\Psi_t(\cdot; V_t)$, but the argument is given as the scalar $\|W\|_{HS}$ rather than the Hilbert space element $W$ itself. Could the authors clarify why this is the case?
- In Section 4.1, the authors focus on translation-invariant kernels. My understanding is that in such cases, $k(t,t) = k(0,0)$, which simplifies the tradeoff between $\|u\|_{\mathcal{H}^d}$ and $k(t,t)$. However, is this restriction strictly necessary for the analysis, or could other kernel families also be used?
- The construction of $Q$ in Proposition 2 appears somewhat mysterious. Could the authors provide more intuition or motivation behind this particular choice?
-In Sections 5 and 6, the online learning algorithms in the RKHS do not appear to fall under the FTRL framework described in Example 1, as they seem to depend on the full gradient outer products rather than just the scalar quantity $\sum_{s=1}^{t-1} \\|G_t\\|^2_{\mathrm{HS}}$. Could the authors clarify how these algorithms are implemented in practice?

## Typos
- In several places, $\\|u\\|\_{\mathcal{H}}$ should be corrected to $\\|u\\|\_{\mathcal{H}^d}$.

**Ethical Concerns:**

["NO or VERY MINOR ethics concerns only"]

**Final Justification:**

The authors have clarified my technical questions, and thus I decided to maintain my rating and recommend acceptance.

**Limitations:**

yes

**Paper Formatting Concerns:**

The formatting of the paper appears correct to me.

**Quality:**

4

**Strengths And Weaknesses:**

## Strengths
- The core idea of the paper is highly original and conceptually elegant. It extends the dynamic-to-static reduction framework of Jacobsen and Orabona (2024) by lifting the problem to an RKHS, enabling the use of advanced tools from static regret minimization and RKHS theory to establish state-of-the-art dynamic regret bounds in a unified way.
- The paper is generally well written and easy to follow. While the RKHS framework can be abstract and mathematically dense, the authors provide a reasonable introduction and maintain clean, consistent notation throughout.

## Weaknesses
- A practical limitation, acknowledged by the authors, is that the current implementation requires $O(t)$ time and memory at round $t$, whereas existing dynamic regret algorithms often admit $O(\ln T)$ time and memory. time and space implementations. Although the authors mention the possibility of using approximate kernel methods to improve efficiency, it remains unclear whether these approximations can preserve the same regret guarantees.
- Conceptually, while the paper successfully recovers state-of-the-art dynamic regret bounds in several settings, it falls short of exceeding them or establishing genuinely new regret guarantees. The only exception appears to be Proposition 5, but the regret bound is not fully instantiated—specifically, the kernel function is left abstract, and the connection to path-length-based guarantees is not made explicit.

---

> ### Author Rebuttal · Authors · 2025-07-30
>
> > In the introduction of RKHS in Section 2, the authors denote the domain of functions in H as X. If I understand correctly, in the context of this paper, X corresponds to the interval $[1,T]$. it would be helpful to make this more explicit. I also suggest using a more consistent notation such at
>
> Yes you are correct, you can think of $\mathcal{X}$ as being $[1,T]$.
> More generally $[1,T]\subseteq\mathcal{X}$ since we often want to define
> the kernel outside of this interval for technical reasons. We will make
> this and the notation more clear.
>
> > In Example 1, the FTRL update involves the function $\Psi_t(\cdot;V_t)$, but the argument is given as the scalar rather than the Hilbert space element itself. Could the authors clarify why this is the case?
>
> The form in this example is given as $\psi(\\|w\\|)$ simply because most
> parameter-free algorithms can be characterized as an FTRL algorithm with
> a regularizer of this form (see e.g. Jacobsen & Cutkosky 2022). We use
> this same type of algorithm to derive the guarantee in Proposition 1, by
> applying it in our kernelized reduction, so the example was meant to
> illustrate how such an algorithm would be implemented concretely. We
> will highlight this connection more clearly.
>
> > In Section 4.1, the authors focus on translation-invariant kernels. My understanding is that in such cases, $k(t,t)=k(0,0)$ , which simplifies the tradeoff between $\\|u\\|\_{H}$ and $k(t,t)$ . However, is this restriction strictly necessary for the analysis, or could other kernel families also be used?
>
> Translation invariant kernels were chosen as an example to show that you
> can recover the usual $\sqrt{P\_{T}T}$ bound in this framework, and in
> particular because there is a natural connection between the RKHS norm
> and path-length when using a translation-invariant kernel (see our
> response [to Reviewer HHCS](https://openreview.net/forum?id=4LSulRbbeL&noteId=5BLJtt4cvd) for details).
>
> However, as you point out, the true potential of the framework is that
> it opens the door to studying new and interesting ways to characterize
> the comparator's complexity by measuring it in terms of the rich class
> of RKHS norms, and indeed this norm need not be associated with a
> translation-invariant kernel. One example can be seen in Example 3 of
> Section 5, where the norm is measured using the linear spline kernel and
> yields a *squared* path-length $S\_{T}=\sum\_{t}\\|u\_{t}-u\_{t-1}\\|^{2}$
> bound rather than the usual $\max\_{t}\\|u\_{t}\\|P\_{T}$.
>
> As another example, suppose you have prior knowledge that the comparator
> is periodic (e.g. you're forecasting energy demand, which naturally
> rises and falls during a 24h cycle, for instance). It is possible to
> account for this prior-knowledge via the RKHS norm $\\|u\\|\_{H}$ and pay
> only the path-length of a *single period*, rather than over the whole
> horizon (comprised of *many* periods). As a simple illustrative example,
> consider an alternating comparator sequence
> $(u\_{t})\_{t}=(0,1,0,1,0,1,\ldots)$. The sequence has path-length $T$ but
> is interpolated by a triangle function over $[0,2]$ extended
> periodically, which can be understood as a function in the periodic
> Sobolev space $H=H^{1}\_{per}([0,2])$ and has
> $\\|u\\|\_{H}^{2}=\int\_{0}^{2}u(t)^{2}dt + \int\_{0}^{2}u'(t)^{2}dt = O(1)$
> and $k(t,t)=O(1)$.
>
>
> > -In Sections 5 and 6, the online learning algorithms in the RKHS do not appear to fall under the FTRL framework described in Example 1, as they seem to depend on the full gradient outer products rather than just the scalar quantity . Could the authors clarify how these algorithms are implemented in practice?
>
> The algorithm in section 6 can also be understood as an FTRL algorithm
> similar to the one in Example 1. The outer products come into the mix
> because they are used to define a weighted norm. The implementation is
> analogous, except you replace $||W||^{2}\_{HS}$ with a weighted version,
> $\\|W\\|\_{t}^{2}=\langle W,\Sigma\_{t} W\rangle\_{HS}$ where
> $\Sigma\_{t}\approx\sum\_{s=1}^{t}G\_{t}\otimes G\_{t}$ (analogous to the
> $\sum\_{s=1}^{t}g\_{t}g\_{t}^{\top}$ weighted norm used in "full-matrix
> AdaGrad").
>
> The kernelized versions of both ONS and the VAW forecaster in Section 5
> have known closed-form updates; we defer to Calandriello et al. (2017b, page 3) and Jézéquel et al. (2019,
> page 4) for concrete implementation details, but note here that they share the same crucial property as Example 1: the updates can be computed without explicit evaluation of the feature map $\phi(t)$.
>
>
> > The construction of in Proposition 2 appears somewhat mysterious. Could the authors provide more intuition or motivation behind this particular choice?
>
> The basic idea is that if we could set $Q(w)=1/|w|$, then using
> properties of Fourier transforms (namely, that
> $\int \hat f(w)\overline{\hat g(w)}dw=\int f(t)\overline{g(t)}dt$ and
> $|F^{-1}\[w f(w)\](t)|\le|\nabla f(t)|$) we would naturally relate
> $$\begin{aligned}
>   \\|u\\|\_{H}^{2}=\int\frac{ |\hat u(w)|^{2}}{Q(w)}dw= \int |w|  \hat u(w) \overline{\hat u(w)}dw \le \int |\nabla u(t)| |u(t)|dt \le \max\_{t}|u(t)| \\|\nabla u\\|\_{L^{1}},\end{aligned}$$
> which is the continuous-time version of the $\max\_{t}\\|u\_{t}\\|P\_{T}$
> dependence that we're after. However, this doesn't quite work because
> $Q(w)=1/|w|$ isn't integrable (the integral blows up at 0 and at
> $\infty$). To fix this, we add extra terms to regularize $Q(w)$ when
> $w\to 0$ and as $w\to\infty$. See our [response to Reviewer HHCS](https://openreview.net/forum?id=4LSulRbbeL&noteId=5BLJtt4cvd) for more
> details.

---

> > ### Author Response · Authors · 2025-08-05
> >
> > Thank you for taking the time to carefully review our submission. As the discussion deadline is approaching, please let us know if we have addressed your concerns or if have additional questions/comments, which we would be happy to answer

---

> > ### Comment · Reviewer_viCE · 2025-08-05
> >
> > I thank the authors for their detailed responses and for addressing my questions. I would like to keep my score and encourage the authors to incorporate these clarifications into the paper.

---

### Official Review · Reviewer_5L4F · 2025-07-02

**Clarity:** 3
**Significance:** 3
**Originality:** 2
**Rating:** 4
**Confidence:** 4

**Summary:**

This paper studies dynamic regret in online convex optimization, where the objective is to achieve low cumulative loss relative to an arbitrary benchmark sequence. The authors frame dynamic regret minimization as a static regret problem in a function space. By carefully constructing a suitable function space in the form of a Reproducing Kernel Hilbert Space (RKHS), their reduction enables us to recover the optimal dynamic regret guarantee in the setting of linear losses. The main contribution of this work lies in the fact that, unlike previous dynamic-to-static reductions which are limited to linear losses, the proposed reduction is applicable to any sequence of losses. As a result, it delivers theoretical result in exp-concave and improper linear regression settings.

**Questions:**

Overall, the main contribution of this paper lies in extending the reduction technique to support any sequence of losses, rather than being limited to linear losses. I view this as an important step forward. However, the following concerns arise regarding the proposed method and its theoretical implications:

1. As shown in Algorithm 1, compared to standard OCO algorithms, the kernelized online learning procedure requires maintaining decision variables and gradients in a higher-dimensional space, which results in increased space complexity. How do the authors assess the practical value of this reduction in light of this cost?
2. For exp-concave functions, the proposed reduction yields an $O(d_{\text{eff}}(\lambda)\ln T)$ regret bound that includes a term $d_{\text{eff}}(\lambda)$, which represents the complexity of the RKHS. Could the authors clarify how large this term might be, and whether it dominates the regret bound?
3. The authors argue that their reduction holds for any sequence of losses. If that is the case, is it possible to obtain a dynamic regret bound that depends on the path length for strongly convex functions? Prior dynamic-to-static reduction [1] could not leverage strong convexity due to their restriction to linear losses.

[1] Jacobsen et al. An equivalence between static and dynamic regret minimization. NeurIPS, 2024.

**Ethical Concerns:**

["NO or VERY MINOR ethics concerns only"]

**Final Justification:**

The authors have acknowledged the computational and memory drawbacks introduced by their algorithm. Nonetheless, I also recognize the significance of the proposed reduction technique. Therefore, following the rebuttal discussion, I am inclined to recommend acceptance of this paper.

**Limitations:**

This dynamic-to-static reduction increases the space complexity, which may limit its practicality in real-world applications.

**Paper Formatting Concerns:**

no major formatting issues

**Quality:**

3

**Strengths And Weaknesses:**

Strengths: The paper is clearly written and makes a significant step forward in reducing dynamic regret to static regret.

Weaknesses: Existing reduction techniques that transform dynamic regret into static regret generally rely on an expanded function space. As a result, these approaches tend to incur high space complexity. Specifically, while the original OCO problem operates in a $d$-dimensional space, the current reduction effectively lifts the problem to a $d\times T$-dimensional space. It is unclear whether this space complexity could become a practical concern (as discussed in Questions).

---

> ### Author Rebuttal · Authors · 2025-07-30
>
> > As shown in Algorithm 1, compared to standard OCO algorithms, the kernelized online learning procedure requires maintaining decision variables and gradients in a higher-dimensional space, which results in increased space complexity. How do the authors assess the practical value of this reduction in light of this cost?
>
> We acknowledge that the naive implementation of kernel methods could
> lead to undesirable overhead; in this work we have made no attempt to
> optimize computation and memory overhead so as to focus on developing
> the base results of the framework, and we are up-front about this
> limitation in the main text. However, as noted in the main text there is
> a large body of work studying efficient implementions of kernel methods
> while incuring acceptable performance trade-offs. Indeed, both the
> kernelized ONS and kernelized VAW forecaster discussed in section 5 have
> efficient approximations with well-studied guarantees (Calandriello et
> al. (2017a,2017b), Jezequel et al. (2019)). We are confident that the
> overhead can be improved in the OLO setting as well, but given the
> non-trivial nature of the topic we believe this issue is beyond the
> scope of the current paper.
>
> > For exp-concave functions, the proposed reduction yields an regret bound that includes a term $d\_{eff}$, which represents the complexity of the RKHS. Could the authors clarify how large this term might be, and whether it dominates the regret bound?
>
> The effective dimension is the main penalty that trades-off with
> $\\|u\\|\_{H}^{2}$ in the curved losses setting; it is necessary to keep
> both terms in mind when designing the kernel, because lowering one tends
> to raise the other. How large it can be is tied to the specific kernel.
> We provide a simple example in Example 3 of how $\lambda\\|u\\|^{2}\_{H}$
> and $d\_{\text{eff}}(\lambda)$ can be traded-off to give
> $S\_{T}^{2/3}T^{2/3}$ bounds, where $S\_{T}=\sum\_{t}\\|u\_{t}-u\_{t-1}\\|^{2}$
> is the *squared* path-length. See also our [response to Reviewer qB8Z](https://openreview.net/forum?id=4LSulRbbeL&noteId=6C3wGyBMTE) for
> additional discussion and intuitions about the effective dimension.
>
> > The authors argue that their reduction holds for any sequence of losses. If that is the case, is it possible to obtain a dynamic regret bound that depends on the path length for strongly convex functions? Prior dynamic-to-static reduction \[1\] could not leverage strong convexity due to their restriction to linear losses.
>
> Yes, the argument is essentially the same as it is for exp-concave
> losses. To see why, observe that under strong convexity what you get is
> $$\begin{aligned}
>   \ell\_{t}(W\_t\phi(t))-\ell\_{t}(U\phi(t)) &\le \langle g\_{t},W\_{t}\phi(t)-U\phi(t)\rangle_W - \frac{\alpha}{2}\\|(W\_{t}-U)\phi(t)\\|^{2}\_{W}\\\\
>   &= \langle g\_{t}\otimes\phi(t), W\_{t}-U\rangle\_{HS}-\frac{\alpha}{2}\langle(W\_{t}-U)(\phi(t)\otimes\phi(t)),W\_{t}-U\rangle\_{HS}.\end{aligned}$$
> Thus, following a standard mirror-descent-based argument you would want
> to set divergence $D\_{\psi\_{t}}(X|Y)=\frac{\alpha}{2}\\|X-Y\\|^{2}\_{t}$,
> where
> $\\|W\\|\_{t}^2=\langle W(\lambda I +\sum\_{s=1}^{t}\phi(s)\otimes\phi(s)), W\rangle\_{HS}$,
> since the negative term above will lead to the divergences telescoping
> to $\frac{\alpha\lambda}{2}\\|U\\|^{2}\_{HS}$, and the remaining stability
> terms get bounded as $O(d\_{eff}(\lambda)\log{T})$ using a generalization
> of the log-determinant lemma (Jezequel et al., 2019). Note that in the
> standard static regret setting, it is possible to remove the the leading
> comparator-norm penalty and pay only a $O(\log{T})$ stability penalty;
> such improvements are not possible here, as removing the $\\|U\\|^2\_{HS}$
> term would imply algorithms which can violate existing
> $R\_{T}(u\_{1},\ldots,u\_{T})\ge \Omega(P\_{T})$ lower bounds for
> strongly-convex losses (Yang et al. 2016).
>
> From the above, we see that the strongly-convex case is essentially the
> same as the exp-concave case in terms of both the approach and the
> guarantee. The main difference from applying K-ONS is that K-ONS uses
> the covariance operator of a product kernel which additionally
> incorporates the gradients
> $\bar{k}(i,j)=\langle g\_{i},g\_{j}\rangle k(i,j)$ (Calandriello et al.
> 2017). Thank you for the insightful question!

---

> > ### Comment · Reviewer_5L4F · 2025-08-03
> >
> > Thank you for your detailed response. I will maintain my score.

---

### Official Review · Reviewer_qB8Z · 2025-07-03

**Clarity:** 2
**Significance:** 3
**Originality:** 3
**Rating:** 4
**Confidence:** 4

**Summary:**

This paper proposes a reduction that frames dynamic regret minimization as a static regret problem in a function space, specifically a RKHS, allowing the application of many well-established static regret minimization results. Moreover, by combining parameter-free methods for static regret minimization with a carefully designed translation-invariant kernel, this paper achieves a dynamic regret of $\tilde{O}(\sqrt{P_T T})$ up to poly-logarithmic terms. Furthermore, this paper demonstrates the applicability of the reduction by introducing several results in static regret minimization, such as scale-free and directionally adaptive guarantees, as well as curved losses, thereby achieving the first corresponding bounds.

**Questions:**

-	In Section 4.1, line 220, the author introduces translation-invariant and Fourier transforms to achieve the trade-off. I find myself puzzled about the reasoning behind presenting the trade-off in this manner. It would be better if the authors could elaborate further on the intuition behind this approach.

-	In Theorem 6, the $d_{eff}$ terms might be somewhat challenging for readers to fully grasp. It would be helpful if the authors could provide additional explanation or offer more physical intuition behind these terms.

**Ethical Concerns:**

["NO or VERY MINOR ethics concerns only"]

**Final Justification:**

I recommend accepting this paper.

**Limitations:**

See Weakness above

**Quality:**

2

**Strengths And Weaknesses:**

**Strengths**

-	Using reduction to apply static regret results to dynamic regret appears quite promising, and this paper demonstrates its feasibility.

-	This paper achieves $T$-free dynamic regret (without requiring the doubling trick) as well as scale-free dynamic regret.

-	Compared to existing works [1, 2] that are limited to linear losses, this paper further extends to curved functions and explores the dynamic regret in this context.


[1] Andrew Jacobsen and Francesco Orabona. An equivalence between static and dynamic regret minimization. In The Thirty-eighth Annual Conference on Neural Information Processing Systems, 2024.

[2] Zhiyu Zhang, Ashok Cutkosky, and Yannis Paschalidis. Unconstrained dynamic regret via sparse coding. Advances in Neural Information Processing Systems, 36, 2024

**Weaknesses**

-	In the abstract, the dynamic regret of $O(\sqrt{P_T T})$ actually includes some poly-logarithmic terms, as presented in Theorem 5. Moreover, since it is obtained using a parameter-free approach, the reader may find it difficult to remove the corresponding poly-logarithmic factors due to the lower bounds of parameter-free methods. It would be greatly appreciated if the authors could clarify the poly-logarithmic dependency in the abstract to avoid unnecessary misunderstandings.

-	The current version of the paper does not appear to provide an analysis of the efficiency issues that may arise when implementing kernel methods in practice, nor does it discuss the potential impact of employing approximate calculations on the results. Some readers might be concerned about whether these factors represent additional costs introduced by the dynamic-to-static reduction. It would be helpful if the authors could include some discussion regarding the complexity of the problem and the practical implications involved.

---

> ### Author Rebuttal · Authors · 2025-07-30
>
> > In Theorem 6, the $d\_{eff}$ terms might be somewhat challenging for readers to fully grasp. It would be helpful if the authors could provide additional explanation or offer more physical intuition behind these terms.
>
> The effective dimension $d\_{eff}(\lambda)$ is a generalization of the
> usual notion of dimension. Consider first a finite-dimensional RKHS $H$:
> if $H$ has dimension $d$, it means that there is an orthonormal basis
> $(\psi\_{i})\_{i=1}^{d}$ such that any $f\in H$ can be written
> $f(x)=\sum\_{i=1}^{d}\alpha\_{i}\psi\_{i}(x)$. Here the notion of dimension
> is the same as the familiar one from linear algebra: the dimension is
> the number of "directions" you can move in, with the directions
> characterized by the basis elements $\psi\_{i}$.
>
> In general an RKHS can be infinite dimensional, so there may be
> infinitely many directions we can move in. However, we can still
> characterize how many of these directions are "significant". One
> important basis of an RKHS is obtained as the eigenfunctions of an
> integral operator related to the kernel, and the effective dimension
> measures how many of these eigenfunctions are associated with a
> "non-neglible" eigenvalue $\sigma\_{1}\ge\sigma\_{2}\ge\ldots\ge 0$. That
> is, let $t\_{0}$ be the first $t$ such that $\sigma\_{t}< \lambda$, then
> $$\begin{aligned}
>   d\_{\text{eff}}(\lambda)=\sum\_{n}\frac{\sigma\_{n}}{\sigma\_{n}+\lambda} = \sum\_{n < t\_{0}}\frac{\sigma\_{n}}{\sigma\_{n}+\lambda}+\sum\_{n\ge t\_{0}}\frac{\sigma\_{n}}{\sigma\_{n}+\lambda} \le t\_{0}+\sum\_{n\ge t\_{0}}\frac{\sigma\_{n}}{\lambda}.\end{aligned}$$
> The first term counts how many $\sigma\_{n}$ are non-negligable relative
> to the regularization parameter $\lambda$, and represents the number
> "important" directions we can move in. The regularization parameter
> $\lambda$ controls the trade-off between the remainder term
> $\sum\_{n\ge t\_{0}}\sigma\_{n}/\lambda$ (the remaining small
> $\sigma\_{n}$'s) and the variability penalty, $\lambda\\|u\\|^{2}\_{H}$.
> Hence, one should aim to design a kernel such that the spectrum is
> controlled and the remainder term can be suitably traded-off against
> $\\|u\\|\_{H}^{2}$. We will add additional discussions to aid the reader's
> intuitions.
>
> > In Section 4.1, line 220, the author introduces translation-invariant and Fourier transforms to achieve the trade-off. I find myself puzzled about the reasoning behind presenting the trade-off in this manner. It would be better if the authors could elaborate further on the intuition behind this approach.
>
> The basic idea is that if we could set $Q(w)=1/|w|$, then using
> properties of Fourier transforms (namely, that
> $\int \hat f(w)\overline{\hat g(w)}dw=\int f(t)\overline{g(t)}dt$ and
> $|F^{-1}\[w f(w)\](t)|\le\|\nabla f(t)|$ ) we would naturally relate the
> RKHS norm associated with a translation-invariant kernel to the
> path-length: $$\begin{aligned}
>   \\|u\\|\_{H}^{2}=\int\frac{ |\hat u(w)|^{2}}{Q(w)}dw= \int |w|  \hat u(w) \overline{\hat u(w)}dw \le \int |\nabla u(t)| |u(t)|dt \le \max\_{t}|u(t)| \\|\nabla u\\|\_{L^{1}},\end{aligned}$$
> which is the continuous-time version of the $\max\_{t}\\|u\_{t}\\|P\_{T}$
> dependence that we want to show. However, this doesn't quite work
> because $Q(w)=1/|w|$ isn't integrable (the integral blows up at 0 and at
> $\infty$). To fix this, we add extra terms to regularize $Q(w)$ when
> $w\to 0$ and as $w\to\infty$. See our [response to Reviewer HHCS](https://openreview.net/forum?id=4LSulRbbeL&noteId=5BLJtt4cvd) for details.

---

> ### Comment · Reviewer_qB8Z · 2025-08-03
>
> I thank the authors for their response and clarification, and I have decided to keep my score.

---

### Official Review · Reviewer_HHCS · 2025-07-13

**Clarity:** 3
**Significance:** 3
**Originality:** 3
**Rating:** 5
**Confidence:** 3

**Summary:**

The paper proposes a clean reduction that turns dynamic-regret minimization in online convex optimization into a static-regret problem in an RKHS. By lifting each comparator sequence into an operator and pairing it with an auxiliary loss, the authors show that the learner’s dynamic regret equals the static regret. Leveraging this view, they (i) match the optimal rate (up to poly-logarithmic terms) in the linear case without horizon knowledge and (ii) demonstrate that the framework extends to exp-concave losses and yields directionally adaptive guarantees with little additional effort.

**Questions:**

1. What intuition drives the logarithmic spectral density in Proposition 2, and how sensitive are the regret bounds to alternative kernels?
2. Given poly-log terms hidden in the bounds, can you provide explicit constants with a short discussion?

**Ethical Concerns:**

["NO or VERY MINOR ethics concerns only"]

**Quality:**

3

**Strengths And Weaknesses:**

The work is principled and technically solid, and the manuscript is generally well written, though clarity suffers in places and empirical evidence is missing.

---

### Strengths

- Neat idea: treat the discrete comparator sequence as a finite-sample realization of an underlying static function. This approach is sensible, though not entirely new given decades of work on kernel embeddings.
- The technique reduces the original problem to static online operator learning in an RKHS, subsuming prior “stacking’’ tricks and allowing infinite-dimensional features. In doing so, it achieves optimal regret without prior knowledge of the horizon, which is implicitly captured by the RKHS construction.
- The construction naturally extends beyond linear loss functions to tackle second-order problems such as online Newton step or directional covariance.

### Weaknesses

- Although well written overall, several improvements would enhance readability:
    - Because the paper is dense, a high-level overview of the results and their significance, with a concise summary of the proof techniques, would help.
    - Given the notation- and equation-heavy presentation, more intuition for key terms (e.g., the spectral density $Q(w)$ in Proposition 2) would be desirable.
    - Minor issues: $g_t$ (line 60) and ${y}^\circ$ (line 90) appear without proper introduction; inconsistency between $\mathcal{H}^d$ and $\mathcal{H}$ in line 60.
- The paper discusses practical implications but provides no empirical study to show whether the kernel construction preserves the theoretical edge. Even a simple numerical experiment, however naïvely implemented, would significantly strengthen the manuscript.

---

> ### Author Rebuttal · Authors · 2025-07-30
>
> > What intuition drives the logarithmic spectral density in Proposition 2
>
> The key intuition is to observe that the ideal choice of density would
> be $Q(w)=1/|w|$. Indeed, we would have (loosely speaking)
> $$\begin{aligned}
>  \\|u\\|\_{H}^{2}=\int\frac{ |\hat u(w)|^{2}}{Q(w)}dw= \int |w|  \hat u(w) \overline{\hat u(w)}dw \le \int |\nabla u(t)| |u(t)|dt \le \max\_{t}|u(t)| \\|\nabla u\\|\_{L^{1}}, \end{aligned}$$
> where we've used Plancherel theorem
> $\int \hat f(w)\overline{\hat g(w)}dw=\int f(t)\overline{g(t)}dt$ and
> the fact that $|F^{-1}\[w \hat u(w)\](t)|\le |\nabla u(t)|$. Notice that
> the final bound is precisely the continuous-time analogue of what we're
> looking for: $\max\_{t}\\|u\_{t}\\|P\_{T}$.
>
> However, $Q(w)=1/|w|$ is not a valid choice because it is not
> integrable: indeed, $\int 1/|w|\ dw$ diverges at 0 and $\infty$. To fix
> this, we instead set $Q(w)=R\_{\infty}(w)R\_{0}(w)/|w|$, where
> $R\_{\infty}(w)=1/(1+w^{p})^{s}$ is a "tapering function", which helps
> control $Q(w)$ in the asymptotic regime, and
> $R\_{0}(w)=1/\log(1+w^{-1/2})\log\log(1+w^{-1/2})$ which ensures that $Q$
> is well-behaved near zero. The intuition behind $R\_{0}(w)$ is that near
> $0$ we have
> $Q(w)\approx R\_{0}(w)/w \approx \frac{1}{w\log(w^{-1/2})\log^{2}\log(w^{-1/2})}$,
> which after a change of variables $t=\log(w^{-1/2})$ integrates near
> zero as
> $\int\_{0}^{\epsilon}Q(w)dw \approx \int\_{\log(1/\epsilon)}^{\infty}\frac{1}{t\log^{2}(t)}dt= O(1)$.
> We can add a discussion of these intuitions to the appendix.
>
> Together these two additional bits of regularization make $Q$
> integrable, yet only increase the associated norm $\\|u\\|\_{H}$ by a small
> amount, letting us achieve almost the same result as the ideal
> $Q(w)=1/|w|$.
>
> > how sensitive are the regret bounds to alternative kernels?
>
> It is possible to design other kernels which achieve $\sqrt{P\_{T}T}$,
> though generally alternative kernels will fail to be horizon
> independent. For instance, it can be shown using our general
> characterization in Section 4.1 that setting
> $Q(w)\approx|w|^{p}/\sqrt{1+w^{2}}$ can also guarantee $\sqrt{P\_{T}T}$
> dynamic regret, so long as you set the exponent $p$ correctly. The
> correct value of $p$ is a function of $T$, making the result horizon
> dependent. We believe the ability of our proposed kernel to avoid prior
> knowledge of T represents a significant and new insight to the
> foundations of dynamic regret.
>
> More generally, in the OLO setting there is a very explicit trade-off in
> terms of the RKHS norm $\\|u\\|\_{H}$ and the variance penalties
> $\sum\_{t}\\|g\_{t}\\|^{2}k(t,t)$, both of which depend on the choice of
> kernel. Kernels with small $k(t,t)$ will tend to make $\\|u\\|\_{H}$ large
> and vice-versa. It is important to carefully balance these two terms
> when designing the kernel. An analogous trade-off occurs between
> $\\|u\\|\_{H}^{2}$ and $d\_{\text{eff}}(\lambda)$ in the curved losses
> setting.
>
> > Given poly-log terms hidden in the bounds, can you provide explicit constants with a short discussion?
>
> A more complete statement of the bound in Proposition 1 would be
> $R\_{T}(u)\le O(\\|u\\|\_{H^{d}}\sqrt{\sum\_{t} \\|g\_{t}\\|^{2}k(t,t)\Lambda\_{T}(u)})$,
> where e.g.
> $\Lambda\_{T}(u)=\log\left(\frac{\\|u\\|\_{H^{d}}\sqrt{\sum\_{t} \\|g\_{t}\\|^{2}k(t,t)}\log^{2}(T)}{\epsilon}+1\right)$
> (The precise form of $\Lambda\_T(u)$ depends on the choice of base
> algorithm; see Proposition 6 in Appendix C.2 for a concrete example).
> The penalty $\Lambda\_{T}(u)$ is seen as the price of adaptivity to the
> comparator sequence, and is incurred by all prior works which minimize
> dynamic regret without prior knowledge of $M\ge \max\_{t}\\|u\_{t}\\|$ (see,
> e.g., Jacobsen & Cutkosky (2022), Zhang et. al. (2023), Jacobsen &
> Orabona (2024)).
>
> Aside from this, additional logarithmic factors can appear due to the
> choice of kernel. The algorithm in Section 4 is significantly more
> adaptive than prior works since it does not require the horizon $T$, and
> this comes at the cost of a multiplicative $\log(T)\log\log(T)$ penalty
> in the bound (see Proposition 3). We conjecture that it is possible to
> incur only the $\Lambda\_{T}(u)$ penalty above when leveraging
> prior-knowledge of $T$, but leave this question for future work.

---

> > ### Comment · Reviewer_HHCS · 2025-08-02
> >
> > Thank the authors for the detailed response, I'd like to keep my score.

---

### Decision · Program_Chairs · 2025-09-17

**Decision:**

Accept (poster)

**Comment:**

This work proposes a novel way to tackle dynamic regret in OCO by reducing it to static regret for an online game in an RKHS. The reduction extends prior work by going beyond linear losses. The resulting algorithms are made computationally tractable by using a kernel trick. The main technical novelty seems to be in designing the right type of shift-invariant kernel. This allows the authors to also take advantage of the curvature of losses, e.g., in special cases the authors are able to preserve exp-concavity of losses and apply an Online Newton-step algorithm in the designed RKHS.

Main concerns raised by reviewers were on the practicality of the algorithm given that the computational and memory complexity will scale with the time-horizon, which is expected due to the kernelization, lack of empirical evaluation and clarity in the writing. This is a theoretical work and it's main contribution is in the reduction to static regret in an RKHS, as such the reviewers have agreed that the practicality issues and lack of empirical evaluation is not a significant enough drawback. I recommend this paper for acceptance and urge the authors to improve the writing as suggested during the reviewing process